# A Mathematical Exploration of Why Language Models Help Solve Downstream Tasks

**Nikunj Saunshi, Sadhika Malladi & Sanjeev Arora**
Princeton University
{`nsaunshi,smalladi,arora`}@cs.princeton.edu

## Abstract

Autoregressive language models, pretrained using large text corpora to do well on next word prediction, have been successful at solving many downstream tasks, even with zero-shot usage. However, there is little theoretical understanding of this success. This paper initiates a mathematical study of this phenomenon for the downstream task of text classification by considering the following questions: (1) What is the intuitive connection between the pretraining task of next word prediction and text classification? (2) How can we mathematically formalize this connection and quantify the benefit of language modeling? For (1), we hypothesize, and verify empirically, that classification tasks of interest can be reformulated as sentence completion tasks, thus making language modeling a meaningful pretraining task. With a mathematical formalization of this hypothesis, we make progress towards (2) and show that language models that are $\epsilon$-optimal in cross-entropy (log-perplexity) learn features that can *linearly solve* such classification tasks with $\mathcal{O}(\sqrt{\epsilon})$ error, thus demonstrating that doing well on language modeling can be beneficial for downstream tasks. We experimentally verify various assumptions and theoretical findings, and also use insights from the analysis to design a new objective function that performs well on some classification tasks.

## 1 Introduction

The construction of increasingly powerful language models has revolutionized natural language processing (NLP). Using gigantic text corpora and a cross-entropy objective, language models are trained to predict a distribution over the *next word* to follow a given context (piece of text). Pretrained language models are useful for many downstream NLP tasks, either as initializations (Ramachandran et al., 2017; Howard & Ruder, 2018) or as a source of contextual word embeddings (McCann et al., 2017; Peters et al., 2018). Recent models (Radford et al., 2019; Brown et al., 2020) have even bypassed the need for careful fine-tuning and have demonstrated strong performance on downstream tasks without fine-tuning. This work aims to understand this incredible success of language models.

Since next word prediction is a powerful test of language understanding, at an intuitive level it is believable that doing well on language modeling can help with many diverse NLP tasks. At the same time, it is quite intriguing how improvements in the test perplexity of language models translate to better downstream performance. Attempting to understand this phenomenon naturally raises the following questions: *(a) why should training on the next-word prediction task, with the cross-entropy objective, result in useful features for downstream tasks? (b) what role do inductive biases of the model architecture and training algorithms play in this empirical success?* Given the nascency of deep learning theory, it is very challenging to say anything mathematically precise about (b) for deep networks. Given these difficulties, this paper focusses on the mathematical study of (a) by exploring if and how quantitative improvements on downstream NLP tasks can be *mathematically guaranteed* for language models that do well on the cross-entropy objective. As a first cut analysis, we restrict attention to *text classification tasks* and the striking observation that they can be solved fairly well with *linear classifiers* on top of fixed language models features, i.e. without finetuning (Table 1). Although we treat models as black boxes, just first-order optimality conditions of the cross-entropy objective reveal interesting properties of learned features, leading to an understanding of their success on classification tasks. Insights from the analysis help us construct a simple objective

(Quad), that provably learns useful features for classification tasks, as also verified empirically. We summarize our contributions along with an overview of the paper below.

In Section 2, we set up notation and formally describe language modeling and the ubiquitous low-dimensional softmax parametrization, along with a description of the cross-entropy objective and properties of its optimal solutions. We then describe the observation, in Section 3.1, that text classification tasks of interest can be reformulated as sentence completion tasks. Amenability to such a reformulation is mathematically formalized (Section 3.2) as the classification task being a *natural task*: tasks that can be solved *linearly* using conditional distribution over words following an input text. Section 4 presents our main results, theorems 4.1 and 4.2, that use the above formalization to mathematically quantify the utility of language model features on natural tasks: $\epsilon$-optimal language model (in cross-entropy) will do $\mathcal{O}(\sqrt{\epsilon})$-well on such tasks. Theorem 4.2 shows a stronger result for low-dimensional softmax models by leveraging a new tool, *conditional mean features* (Definition 4.1), which we show (Section 6) to be effective in practice. The usefulness of the language model features themselves is demonstrated by arguing a weak linear relationship between them and conditional mean features. In Section 5.2, we present a new mathematically motivated objective (*Quad*) that has formal guarantees. Experiments in Section 6 verify the sentence completion reformulation idea and the good performance of conditional mean features on standard benchmarks.

## 1.1 RELATED WORK

**Text embedding methods:** Prior to language models, large text corpora like Wikipedia (Merity et al., 2016) were used to learn low-dimensional embeddings for words (Mikolov et al., 2013b;a; Pennington et al., 2014) and subsequently for sentences (Kiros et al., 2015; Arora et al., 2017; Pagliardini et al., 2018; Logeswaran & Lee, 2018) for downstream task usage. These methods were inspired by the distributional hypothesis (Firth, 1957; Harris, 1954), which posits that meaning of text is determined in part by the surrounding context. Recent methods like BERT (Devlin et al., 2018) and variants (Lan et al., 2019; Yang et al., 2019; Liu et al., 2019) learn models from auxiliary tasks, such as sentence completion, and are among the top performers on downstream tasks. In this work we consider autoregressive models and make a distinction from masked language models like BERT; Table 2 shows that language model and BERT features have comparable performances.

**Language models for downstream tasks:** We are interested in language models (Chen & Goodman, 1999), especially those that use neural networks to compute low-dimensional features for contexts and parametrize the next word distribution using softmax (Xu & Rudnicky, 2000; Bengio et al., 2003). Language models have shown to be useful for downstream tasks as initializations (Ramachandran et al., 2017; Howard & Ruder, 2018) or as learned feature maps (Radford et al., 2017; McCann et al., 2017; Peters et al., 2018). The idea of phrasing classification tasks as sentence completion problems to use language models is motivated by recent works (Radford et al., 2019; Puri & Catanzaro, 2019; Schick & Schütze, 2020) that show that many downstream tasks can be solved by next word prediction for an appropriately conditioned language model. This idea also shares similarities with work that phrase a suite of downstream tasks as question-answering tasks (McCann et al., 2018) or text-to-text tasks (Raffel et al., 2019) and symbolic reasoning as fill-in-the-blank tasks (Talmor et al., 2019). Our work exploits this prevalent idea of task rephrasing to theoretically analyze why language models succeed on downstream tasks.

**Relevant theory:** Since the success of early word embedding algorithms like word2vec (Mikolov et al., 2013a) and GloVe (Pennington et al., 2014), there have been attempts to understand them theoretically. Levy & Goldberg (2014) argue that word2vec algorithm implicitly factorizes the PMI matrix. Noise Contrastive Estimation (NCE) theory is used to understand word embeddings (Dyer, 2014) and to show parameter recovery for negative sampling based conditional models (Ma & Collins, 2018). A latent variable model (Arora et al., 2016) is used to explain and unify various word embedding algorithms. Theoretical justification is provided for sentence embedding methods either by using a latent variable model (Arora et al., 2017) or through the lens of compressed sensing (Arora et al., 2018). Also relevant is recent work on theory for contrastive learning (Arora et al., 2019; Tosh et al., 2020b;a; Wang & Isola, 2020) and reconstruction-based methods (Lee et al., 2020), which analyze the utility of self-supervised representations learned for downstream tasks. Our work is the first to analyze the efficacy of language model features on downstream tasks.

## 2  LANGUAGE MODELING AND OPTIMAL SOLUTIONS

We use $\mathcal{S}$ to denote the discrete set of all contexts, i.e. complete or partial sentences (prefixes), $\mathcal{W}$ to denote the vocabulary of words, with $V = |\mathcal{W}|$ being the vocabulary size. For a discrete set $A$, let $\Delta_A$ denote the set of distributions on $A$. We use $p, p_L \in \Delta_{\mathcal{S}}$ to denote probability distributions over $\mathcal{S}$, and $p_{\cdot|s}, p^*_{\cdot|s} \in \Delta_{\mathcal{W}}$ to denote conditional distributions, where $p_{\cdot|s}(w)$ is the predicted probability of word $w$ following context $s$ and $p^*_{\cdot|s}(w)$ denotes the true conditional probability. Boldface $\boldsymbol{p}_{\cdot|s}, \boldsymbol{p}^*_{\cdot|s} \in \mathbb{R}^V$ denote vectors of probabilities for $p_{\cdot|s}, p^*_{\cdot|s} \in \Delta_{\mathcal{W}}$. For $\boldsymbol{v} \in \mathbb{R}^V$, $\boldsymbol{v}(w)$ indexes the coordinate for $w \in \mathcal{W}$; $\boldsymbol{p}_{\cdot|s}(w)$ is the probability of $w$ according to $p_{\cdot|s}$. We use $\phi_w \in \mathbb{R}^d$ to denote a $d$-dimensional embedding for word $w$; word embeddings are stacked into the columns $\Phi \in \mathbb{R}^{d \times V}$. We use $f : \mathcal{S} \to \mathbb{R}^d$ for a feature map from contexts to $d$-dimensional embeddings, e.g. $f(s)$ can be the output of a Transformer model for input context $s \in \mathcal{S}$. For embeddings $\{\theta_s\}_{s \in \mathcal{S}}$ with $\theta_s \in \mathbb{R}^D$ (any $D$), we use $\{\theta_s\}$ to denote $g : \mathcal{S} \to \mathbb{R}^D$ such that $g(s) = \theta_s$.

### 2.1  LANGUAGE MODELING USING CROSS-ENTROPY

Language model aims to learn the true distribution of a text corpus and a popular approach to do so is through next word prediction. Given a context (e.g., a sentence $s \in \mathcal{S}$), it predicts a distribution $p_{\cdot|s}$ over the word to follow, e.g. for the context "The food was ", the model could place high probabilities on words "delicious", "expensive", "bland", etc. We use $p_L$ to denote the true distribution over the context set $\mathcal{S}$ in the language modeling corpus. A standard approach is to minimize the expected cross-entropy loss between the true distribution $p^*_{\cdot|s}$ and the model prediction $p_{\cdot|s}$. We define the cross-entropy loss for a language model with output vector of probabilities $\{\boldsymbol{p}_{\cdot|s}\}_{s \in \mathcal{S}}$ as

$$\ell_{\text{xent}}(\{\boldsymbol{p}_{\cdot|s}\}) = \mathop{\mathbb{E}}_{s \sim p_L} \mathop{\mathbb{E}}_{w \sim p^*_{\cdot|s}} \left[ -\log(\boldsymbol{p}_{\cdot|s}(w)) \right] = \mathop{\mathbb{E}}_{s \sim p_L} \left[ \ell_{\text{xent},s}(\boldsymbol{p}_{\cdot|s}) \right] \tag{1}$$

To understand what language models learn, we look at the optimal solution of the cross-entropy objective. While one cannot practically hope to learn the optimal solution due to optimization, statistical and expressivity limitations, the optimal solution at least tells us the best that language modeling can hope to do. A well-known property of cross-entropy objective is that its optimal solution is $\boldsymbol{p}^*_{\cdot|s}$, which can be proved by noting that $\ell_{\text{xent},s}(\boldsymbol{p}_{\cdot|s}) = D_{\text{KL}}(p^*_{\cdot|s}, p_{\cdot|s}) + C$.

**Proposition 2.1** (Cross-entropy recovers $\boldsymbol{p}^*_{\cdot|s}$). *The unique minimizer of $\ell_{\text{xent}}(\{\boldsymbol{p}_{\cdot|s}\})$ is $\boldsymbol{p}_{\cdot|s} = \boldsymbol{p}^*_{\cdot|s}$ for every $s \in \text{support}(p_L)$.*

### 2.2  SOFTMAX PARAMETRIZED LANGUAGE MODELING

Unlike traditional language models like $n$-gram models, neural language models parametrize the conditional distribution $p_{\cdot|s}$ as a softmax computed using *low dimensional* embeddings. For an embedding $\theta \in \mathbb{R}^d$, the softmax distribution over $\mathcal{W}$ using word embeddings $\Phi \in \mathbb{R}^{d \times V}$ is $p_{\theta,\Phi}(w) = e^{\theta^\top \phi_w} / Z_\theta$, where $Z_\theta = \sum_{w' \in \mathcal{W}} e^{\theta^\top \phi_{w'}}$ is the partition function. While $p_{\theta,\Phi}$ depends on $\Phi$, we will use $p_\theta$ instead whenever $\Phi$ is clear from context. Just like $\boldsymbol{p}^*_{\cdot|s}$, we can interpret $\boldsymbol{p}_\theta \in \mathbb{R}^V$ as a vector of probabilities for the distribution $p_\theta$.

We now describe the abstraction for softmax models that is applicable to most neural models. A language model first embeds a context $s$ into $f(s) \in \mathbb{R}^d$ using a feature map $f : \mathcal{S} \to \mathbb{R}^d$ that is parametrized by an architecture of choice (e.g. Transformer (Vaswani et al., 2017)). The output conditional distribution is set to be the softmax distribution induced by the context embedding $f(s)$ and word embeddings $\Phi$, i.e. $p_{\cdot|s} = p_{f(s)}$. The cross-entropy in its familiar form is presented below

$$\ell_{\text{xent}}(f, \Phi) = \mathop{\mathbb{E}}_{s \sim p_L} \mathop{\mathbb{E}}_{w \sim p^*_{\cdot|s}} \left[ -\log(\boldsymbol{p}_{f(s)}(w)) \right] = \mathop{\mathbb{E}}_{s \sim p_L} \left[ \mathop{\mathbb{E}}_{w \sim p^*_{\cdot|s}} \left[ -f(s)^\top \phi_w \right] + \log(Z_{f(s)}) \right] \tag{2}$$

We rewrite it as $\ell_{\text{xent}}(f, \Phi) = \mathop{\mathbb{E}}_{s \sim p_L} [\ell_{\text{xent},s}(f(s), \Phi)]$, where $\ell_{\text{xent},s}(\theta, \Phi) = \ell_{\text{xent},s}(\boldsymbol{p}_{\theta,\Phi})$ is the cross-entropy loss for a context $s$ that uses embedding $\theta$. Analogous to Proposition 2.1, we would like to know the optimal $d$-dimensional feature map $f^*$ and the induced conditional distribution $\boldsymbol{p}_{f^*(s)}$[1].

---

[1] A finite minimizer may not always exist. This is handled in Section 4 that deals with $\epsilon$-optimal solutions.

**Proposition 2.2** (Softmax models recover $\boldsymbol{p}^*_{\cdot|s}$ on a subspace). *Fix a fixed $\Phi$, if $f^* \in \arg\min_{f:\mathcal{S}\to\mathbb{R}^d} \ell_{xent}(f, \Phi)$ exists, then $\Phi\boldsymbol{p}_{f^*(s)} = \Phi\boldsymbol{p}^*_{\cdot|s}$ for every $s \in support(p_L)$.*

Unlike Proposition 2.1, $\boldsymbol{p}_{f^*(s)} \in \mathbb{R}^V$ is only guaranteed to be equal to $\boldsymbol{p}^*_{\cdot|s} \in \mathbb{R}^V$ on the $d$-dimensional subspace spanned by rows of $\Phi \in \mathbb{R}^{d \times V}$. We may not learn $\boldsymbol{p}^*_{\cdot|s}$ exactly when $d < V$, but this result at least guarantees learning $\boldsymbol{p}^*_{\cdot|s}$ on a *linear subspace* determined by word embeddings $\Phi$. This forms the basis for our main results later and is proved by using the first-order optimality condition, i.e. $\nabla_\theta \ell_{\text{xent},s}(f^*(s)) = 0, \forall s \in \mathcal{S}$. The gradient of cross-entropy is $\nabla_\theta \ell_{\text{xent},s}(\theta) = -\Phi\boldsymbol{p}^*_{\cdot|s} + \nabla_\theta Z_\theta / Z_\theta = -\Phi\boldsymbol{p}^*_{\cdot|s} + \Phi\boldsymbol{p}_\theta$. Setting it to 0 completes the proof. We use the properties of optimal solutions to understand why language models help with classification tasks.

# 3 USING LANGUAGE MODELS FOR CLASSIFICATION TASKS

Sections 2.1 and 2.2 suggest that language models aim to learn $\boldsymbol{p}^*_{\cdot|s}$, or a low-dimensional projection $\Phi\boldsymbol{p}^*_{\cdot|s}$. Thus to understand why language models help with downstream tasks, a natural starting point is to understand how access to $\boldsymbol{p}^*_{\cdot|s}$ can help with downstream tasks. In a thought experiment, we use oracle access to $\boldsymbol{p}^*_{\cdot|s}$ for any $s$ and demonstrate that sentence classification task can be solved by reformulating it as a sentence completion problem and using $\boldsymbol{p}^*_{\cdot|s}$ to get completions to predict the label. This sentence completion reformulation is mathematically formalized as *natural tasks*.

## 3.1 SENTENCE COMPLETION REFORMULATION

For exposition, we consider the sentence classification task of sentiment analysis, where the inputs are movie reviews (subset of $\mathcal{S}$) and labels belongs to $\{\pm 1\}$, denoting positive and negative reviews.

**Classification task as sentence completion:** Can we predict the label for a movie review $s$ by using $\boldsymbol{p}^*_{\cdot|s}$? One way is to use $\boldsymbol{p}^*_{\cdot|s}$ to compare probabilities of ":)" and ":(" following a movie review and to predict sentiment based on which is higher. This seems like a reasonable strategy, since ":)" is likelier than ":(" to follow a positive movie review. One issue, however, is that $\boldsymbol{p}^*_{\cdot|s}$ will place much higher probability on words that start sentences, like "The", rather than discriminative words useful for the task. To allow a larger set of grammatically correct completions, we can append a prompt like "This movie is " at the end of all movie reviews and query probabilities of indicative adjectives like good, bad, interesting, boring etc. that are better indicators of sentiment. This approach of adding a prompt can also work for other classification tasks. For the AG news dataset (Zhang et al., 2015) containing news articles from 4 categories (world, science/tech., sports, business), a prompt like "This article is about " can help solve the task. The theoretical and practical relevance of prompts is discussed in Theorem 4.1, and Section 6 respectively. We note that the choice of prompts and completion words is less important than the underlying idea of sentence completion reformulation and its formalization.

**Solving tasks using a linear function of $\boldsymbol{p}^*_{\cdot|s}$:** The above process is actually a sub-case of using a linear classifier on top of $\boldsymbol{p}^*_{\cdot|s} \in \mathbb{R}^V$. For sentiment analysis, if $w_+ = ":)"$ and $w_- = ":("$, then the sign of $\boldsymbol{p}^*_{\cdot|s}(w_+) - \boldsymbol{p}^*_{\cdot|s}(w_-)$ can predict the sentiment. This strategy can be expressed as $\boldsymbol{v}^\top \boldsymbol{p}^*_{\cdot|s}$, where the linear classifier $\boldsymbol{v} \in \mathbb{R}^V$ has $\boldsymbol{v}(w_+) = 1$, $\boldsymbol{v}(w_-) = -1$ and $\boldsymbol{v}(w') = 0$ for $w' \in \mathcal{W}\backslash\{w_+, w_-\}$. Similarly with the prompt, we can assign positive weights in $\boldsymbol{v}$ to adjectives like "good" and negative weights to adjectives like "boring". Strength of sentiment in different adjectives (e.g., "good" vs "amazing") can be captured through different weights. This equivalence between sentence completion reformulation and linear classifier on $\boldsymbol{p}^*_{\cdot|s}$ is further explored in Section D.1. Other tasks can be similarly solved with a different set of words for each class. We verify experimentally that SST and AG news tasks can be solved by a linear function of probabilities of just a small subset of words in Section 6 and for many other classification tasks in Section F.1, thus lending credibility to the sentence completion view.

### 3.2 NATURAL CLASSIFICATION TASKS

We now translate the above sentence completion reformulation into a reasonable mathematical characterization for classification tasks of interest. Firstly we formally define text classification tasks and the standard metric for performance of linear classification on fixed features. A binary classification task[2] $\mathcal{T}$ is characterized by a distribution $p_{\mathcal{T}}$ over $\mathcal{S} \times \{\pm 1\}$, where the input $s$ is a piece of text from $\mathcal{S}$ and the label $y$ is in $\{\pm 1\}$. Given a feature map $g : \mathcal{S} \to \mathbb{R}^D$ (arbitrary $D$), $\mathcal{T}$ is solved by fitting a linear classifier $\boldsymbol{v} \in \mathbb{R}^D$ on top of $g(s)$ and the metric of classification loss is

$$\ell_{\mathcal{T}}(g, \boldsymbol{v}) = \mathbb{E}_{(s,y) \sim p_{\mathcal{T}}} \left[ \ell(\boldsymbol{v}^\top g(s), y) \right]; \;\; \ell_{\mathcal{T}}(g) = \inf_{\boldsymbol{v} \in \mathbb{R}^D} \ell_{\mathcal{T}}(g, \boldsymbol{v}) \tag{3}$$

where $\ell$ is a 1-Lipschitz surrogate to the 0-1 loss, like the hinge loss $\ell(\hat{y}, y) = (1 - y\hat{y})_+$ or the logistic loss $\ell(\hat{y}, y) = \log(1 + e^{-y\hat{y}})$. For given embeddings $\{\theta_s\}_{s \in \mathcal{S}}$, the classification loss is written as $\ell_{\mathcal{T}}(\{\theta_s\}, \boldsymbol{v}) = \mathbb{E}_{(s,y) \sim p_{\mathcal{T}}}[\ell(\boldsymbol{v}^\top \theta_s, y)]$.

We now formalize classification tasks amenable to sentence completion reformulation, from Section 3.1), as $(\tau, B)$-natural tasks, i.e. tasks that achieve a small classification loss of $\tau$ by using a linear classifier with $\ell_\infty$-norm bounded[3] by $B$ on top of features $\boldsymbol{p}^*_{\cdot|s} \in \mathbb{R}^V$.

**Definition 3.1.** *A classification task $\mathcal{T}$ is $(\tau, B)$-natural if* $\min_{\boldsymbol{v} \in \mathbb{R}^V, \|\boldsymbol{v}\|_\infty \leq B} \ell_{\mathcal{T}}(\{\boldsymbol{p}^*_{\cdot|s}\}, \boldsymbol{v}) \leq \tau$.

While we motivated this formalization of linear classification over $\boldsymbol{p}^*_{\cdot|s}$ in Section 3.1, we provide a mathematical justification in Section D.1, along with interpretations for $\tau$ and $B$ that relate them to the Bayes optimal predictor and probability mass of indicative words respectively. Low dimensional softmax models, however, only learn $\boldsymbol{p}^*_{\cdot|s}$ in the subspace of $\Phi$, per Proposition 2.2. Thus we are also interested in subset of tasks that this subspace can solve.

**Definition 3.2.** *Task $\mathcal{T}$ is $(\tau, B)$-natural w.r.t. $\Phi \in \mathbb{R}^{d \times V}$ if* $\min_{\boldsymbol{v} \in \text{row-span}(\Phi), \|\boldsymbol{v}\|_\infty \leq B} \ell_{\mathcal{T}}(\{\boldsymbol{p}^*_{\cdot|s}\}, \boldsymbol{v}) \leq \tau$.

Note that every $(\tau, B)$-natural task w.r.t. $\Phi$ is trivially $(\tau, B)$-natural, though the converse may not hold. However it can be argued that if $\Phi$ has some "nice properties", then $(\tau, B)$-natural tasks of interest will roughly also be $(\tau, B)$-natural w.r.t. $\Phi$. Capturing the synonym structure of words can be such a nice property, as discussed in Section D.2. A better understanding of these properties of word embeddings $\Phi$ can potentially enable better performance of language models on downstream tasks. In fact, Section 5.2 describes a carefully designed objective that can learn word embeddings with desirable properties like synonyms having similar embeddings. In the subsequent sections, we use the above formalization to show guarantees for language models on natural tasks.

## 4 GUARANTEES FOR LANGUAGE MODELS ON NATURAL TASKS

We now show guarantees for features from language models on natural tasks in two cases: 1) for an arbitrary language model $\{p_{\cdot|s}\}$ where we use $V$-dimensional features $\boldsymbol{p}_{\cdot|s} \in \mathbb{R}^V$ for downstream tasks and 2) for softmax language model $(f, \Phi)$ where we use new $d$-dimensional features $\Phi \boldsymbol{p}_{f(s)} \in \mathbb{R}^d$. Since we cannot practically hope to learn the optimal solutions described in propositions 2.1 and 2.2, we only assume that the language models are $\epsilon$-optimal in cross-entropy. We first define $\ell^*_{\text{xent}}$ to be the minimum achievable cross-entropy and $\ell^*_{\text{xent}}(\Phi)$ to be the minimum achievable cross-entropy by a $d$-dimensional softmax language model using $\Phi$; clearly $\ell^*_{\text{xent}} \leq \ell^*_{\text{xent}}(\Phi)$.

$$\ell^*_{\text{xent}} = \ell_{\text{xent}}(\{\boldsymbol{p}^*_{\cdot|s}\}), \;\; \ell^*_{\text{xent}}(\Phi) = \mathbb{E}_{s \sim p_L} \left[ \inf_{\theta \in \mathbb{R}^d} \ell_{\text{xent},s}(\theta, \Phi) \right] \tag{4}$$

We first present the results for arbitrary language models with a proof sketch that describes the main ideas, following which we present our main results for softmax language models.

### 4.1 ARBITARY LANGUAGE MODELS

We show guarantees for a language model that is $\epsilon$-optimal, i.e. $\ell_{\text{xent}}(\{\boldsymbol{p}_{\cdot|s}\}) - \ell^*_{\text{xent}} \leq \epsilon$, on $(\tau, B)$-natural tasks. An important consideration is that the language model distribution $p_L$ of contexts is

---

[2]Extending to $k$-way tasks is straightforward.

[3]$\ell_\infty$ makes sense since $\|\boldsymbol{p}^*_{\cdot|s}\|_1 = 1$ & $\|\cdot\|_\infty$ is dual norm of $\|\cdot\|_1$.

often a diverse superset of the downstream distribution $p_{\mathcal{T}}$ (defined in Section 2.2) over sentences, thus requiring us to show how guarantees of $\boldsymbol{p}_{\cdot|s} \approx \boldsymbol{p}^*_{\cdot|s}$ *on average* over the distribution $s \sim p_L$ transfer to guarantees on a subset $p_{\mathcal{T}}$. In the worst case, all of the $\epsilon$ error in cross-entropy by $\{\boldsymbol{p}_{\cdot|s}\}$ is incurred on sentences from the subset $p_{\mathcal{T}}$, leading to pessimistic bounds[4]. In practice, however, the errors might be more evenly distributed across $p_L$, thus bypassing this worst case bound. As a first step, we present the worst case bound here; stronger guarantees are in Section 5.1. The worst-case coefficient $\gamma(p_{\mathcal{T}})$, defined below, captures that $p_{\mathcal{T}}$ is a $\gamma(p_{\mathcal{T}})$-fraction of $p_L$.

$$\gamma(p_{\mathcal{T}}) = \sup\{\gamma \in (0,1] : p_L(s) \geq \gamma p_{\mathcal{T}}(s) \ \forall s \in \mathcal{S}\} \tag{5}$$

We now present our results that applies to any language model, regardless of the parametrization (e.g., $n$-gram models, softmax models). The result suggests that small test cross-entropy (hence test perplexity) is desirable to guarantee good classification performance, thus formalizing the intuition that better language models will be more useful for downstream tasks.

**Theorem 4.1.** *Let $\{\boldsymbol{p}_{\cdot|s}\}$ be a language model that is $\epsilon$-optimal, i.e. $\ell_{xent}(\{\boldsymbol{p}_{\cdot|s}\}) - \ell^*_{xent} \leq \epsilon$, for some $\epsilon > 0$. For a classification task $\mathcal{T}$ that is $(\tau, B)$-natural, we have*

$$\ell_{\mathcal{T}}\left(\{\boldsymbol{p}_{\cdot|s}\}\right) \leq \tau + \sqrt{2B^2 \epsilon \left(\gamma(p_{\mathcal{T}})\right)^{-1}}$$

This upper bounds classification loss on task $\mathcal{T}$ for $V$-dimensional features $\{\boldsymbol{p}_{\cdot|s}\}$ from an $\epsilon$-optimal language model. We discuss factors that lead to small upper bound and corresponding intuitions.
- $\epsilon$ is small: learned language model has smaller cross-entropy (log-perplexity)
- $\tau$ is small: task can be solved well through a sentence completion reformulation with a set of indicative words as completions, as in Section 3.1, and has small Bayes error (cf. Section D.1)
- $B$ is small: set of indicative words has high probability mass in $\boldsymbol{p}^*_{\cdot|s}$ (cf. Section D.1). This could potentially explain the superior performance when prompts are added (Section 6).
- $\gamma(p_{\mathcal{T}})$ is large: $p_{\mathcal{T}}$ is closer to $p_L$; note that $\gamma(p_{\mathcal{T}}) \leq 1$ with equality if and only if $p_{\mathcal{T}} = p_L$

Thus the bound captures meaningful intuitions about good performance of language models on downstream tasks. We provide a detailed proof sketch in Section E.1 and a strengthened version of this (Theorem B.1) is presented in Section E.6. Proving this result requires connecting the classification loss with language modeling cross-entropy loss and dealing with distribution mismatch; we present a rough outline to do so below. Since $\mathcal{T}$ is $(\tau, B)$-natural, let $\boldsymbol{v}^*$ be the classifier with $\|\boldsymbol{v}^*\|_\infty \leq B$ and $\ell_{\mathcal{T}}(\{\boldsymbol{p}^*_{\cdot|s}\}, \boldsymbol{v}^*) \leq \tau$. The result follows from the following 3 inequalities:

$$\ell_{\mathcal{T}}\left(\{\boldsymbol{p}_{\cdot|s}\}, \boldsymbol{v}^*\right) - \ell_{\mathcal{T}}(\{\boldsymbol{p}^*_{\cdot|s}\}, \boldsymbol{v}^*) \leq \sqrt{\mathop{\mathbb{E}}_{s \sim p_{\mathcal{T}}}[(\boldsymbol{v}^{*\top}(\boldsymbol{p}_{\cdot|s} - \boldsymbol{p}^*_{\cdot|s}))^2]} \qquad \text{... Lipschitzness + Jensen's}$$

$$\mathop{\mathbb{E}}_{s \sim p_{\mathcal{T}}}[(\boldsymbol{v}^{*\top}(\boldsymbol{p}_{\cdot|s} - \boldsymbol{p}^*_{\cdot|s}))^2] \leq \gamma(p_{\mathcal{T}})^{-1} \mathop{\mathbb{E}}_{s \sim p_L}[(\boldsymbol{v}^{*\top}(\boldsymbol{p}_{\cdot|s} - \boldsymbol{p}^*_{\cdot|s}))^2] \qquad \text{... Transfer } p_{\mathcal{T}} \text{ to } p_L$$

$$\forall \boldsymbol{v} \in \mathbb{R}^V, \ (\boldsymbol{v}^\top(\boldsymbol{p}_{\cdot|s} - \boldsymbol{p}^*_{\cdot|s}))^2 \leq 2\|\boldsymbol{v}\|^2_\infty(\ell_{\text{xent},s}(\boldsymbol{p}_{\cdot|s}) - \ell_{\text{xent},s}(\boldsymbol{p}^*_{\cdot|s})) \quad \text{... Pinsker's inequality}$$

The first and third inequalities (Lemma E.8 and Lemma E.3) connect the classification loss to the cross-entropy loss in language modeling, while the second inequality deals with distribution mismatch between $p_L$ and $p_{\mathcal{T}}$. We now present a stronger result for softmax models.

## 4.2 SOFTMAX LANGUAGE MODEL WITH CONDITIONAL MEAN FEATURES

We now consider a softmax language model with feature map $f$ that satisfies $\ell_{\text{xent}}(f, \Phi) - \ell^*_{\text{xent}}(\Phi) \leq \epsilon$; suboptimality is measured w.r.t. the best $d$-dimensional model, unlike Theorem 4.1,. Note that Theorem 4.1 can be invoked here to give a bound of $\ell_{\mathcal{T}}(\{\boldsymbol{p}_{f(s)}\}) \leq \tau + \mathcal{O}(B\sqrt{\epsilon + \epsilon^*_\Phi})$ on $(\tau, B)$-natural tasks, where $\epsilon^*_\Phi = \ell^*_{\text{xent}}(\Phi) - \ell^*_{\text{xent}}$ is the suboptimality of the best $d$-dimensional model. The fixed error of $\mathcal{O}(B\sqrt{\epsilon^*_\Phi})$ (even when $\epsilon = 0$), however, is undesirable. We improve on this by proving a stronger result specifically for softmax models. Inspired by Proposition 2.2, our guarantees are for features $\Phi\boldsymbol{p}_{f(s)} \in \mathbb{R}^d$ called conditional mean features.

**Definition 4.1** (Conditional Mean Features)**.** *For a feature map $f : \mathcal{S} \to \mathbb{R}^d$ and $\Phi \in \mathbb{R}^{d \times V}$, we define conditional mean features $\Phi p_f : \mathcal{S} \to \mathbb{R}^d$, where $\Phi p_f(s) = \Phi\boldsymbol{p}_{f(s)}$, where $\boldsymbol{p}_{f(s)} \in \mathbb{R}^V$.*

---

[4]For instance if $p_{\mathcal{T}}$ is 0.001 fraction of $p_L$, $\{p_{\cdot|s}\}$ could have $1000\epsilon$ error on $p_{\mathcal{T}}$ and 0 error on rest of $p_L$.

We now present the result for softmax language models that has similar implications as Theorem 4.1, but with above-mentioned subtle differences.

**Theorem 4.2.** *For a fixed $\Phi$, let $f$ be features from an $\epsilon$-optimal $d$-dimensional softmax language model, i.e. $\ell_{xent}(f, \Phi) - \ell^*_{xent}(\Phi) \leq \epsilon$. For a classification task $\mathcal{T}$ that is $(\tau, B)$-natural w.r.t. $\Phi$,*

$$\ell_{\mathcal{T}}(\Phi p_f) \leq \tau + \sqrt{2B^2 \epsilon \left( \gamma(p_{\mathcal{T}}) \right)^{-1}}$$

This result guarantees good performance of conditional mean features $\Phi p_f$ on some natural tasks, thereby suggesting a novel way to extract features for downstream tasks. We empirically verify the good performance of $\Phi p_f(s)$ on classifications tasks (Section 6) and also find a $\mathcal{O}(\sqrt{\epsilon})$-like behavior (Section F.5). The proof (Section E.3) is similar to that of Theorem 4.1, the main difference being the use of the following inequality, proved using a *softmax variant of Pinsker's inequality* (Lemma E.4).

$$\forall \boldsymbol{v} \in \text{row-span}(\Phi), \ (\boldsymbol{v}^\top (\boldsymbol{p}_{f(s)} - \boldsymbol{p}^*_{\cdot|s}))^2 \leq 2\|\boldsymbol{v}\|_\infty^2 (\ell_{\text{xent},s}(\boldsymbol{p}_{f(s)}) - \inf_{f^*(s) \in \mathbb{R}^d} \ell_{\text{xent},s}(\boldsymbol{p}_{f^*(s)}))$$

The more general result (Theorem 5.1) replaces $\gamma(p_{\mathcal{T}})$ with a more refined coefficient (Section 5.1). While guarantees are only for natural tasks w.r.t. $\Phi$, Section D.2 discusses why this might be enough for *tasks of interest* if word embeddings $\Phi$ satisfy *nice properties*.

### 4.3 $\Phi p_f(s)$ IS A LINEAR FUNCTION OF $f(s)$

Theorem 4.2 shows that $\Phi p_f$ is useful for linear classification. However, using feature map $f$ directly is more standard and performs better in practice (Section 6). Here we argue that there is a linear relation between $f$ and $\Phi p_f$ if word embeddings $\Phi$ satisfy a certain Gaussian-like property, which we show implies that tasks solvable linearly with $\Phi p_f$ are also solvable linearly using $f$.

**Assumption 4.1.** *There exists a symmetric positive semidefinite matrix $\boldsymbol{A} \in \mathbb{R}^{d \times d}$, a vector $\boldsymbol{b} \in \mathbb{R}^d$ and a constant $c \in \mathbb{R}$ such that $\log(Z_\theta) = \frac{1}{2}\theta^\top \boldsymbol{A}\theta + \theta^\top \boldsymbol{b} + c$ for any $\theta \in \mathbb{R}^d$.*

If word embeddings were distributed as Gaussians, i.e. $V$ columns of $\Phi$ are sampled from $\mathcal{N}(\mu, \Sigma)$ independently, it is not hard to show (Lemma E.1) that $\log(Z_\theta) \approx \frac{1}{2}\theta^\top \Sigma \theta + \theta^\top \mu + \log(V)$. While some papers (Arora et al., 2016; Mu & Viswanath, 2018) have noted that word embeddings are fairly random-like in the bulk to argue that the log partition function is constant for $\|\theta\|_2 = 1$, our quadratic assumption is a bit stronger. However, empirically we find the fit to be very good, as evident in Figure 1. Under the above assumption, we can show a linear relation between $f$ and $\Phi p_f$.

**Lemma 4.3.** *Under Assumption 4.1, feature map $f$ satisfies $\Phi p_f(s) = \boldsymbol{A}f(s) + \boldsymbol{b}, \forall s \in \mathcal{S}$.*

**Corollary 4.1.** *Under same setting as Lemma 4.3 and Theorem 4.2, $\ell_{\mathcal{T}}(f) \leq \tau + \mathcal{O}(B\sqrt{\epsilon})$.*

This shows that $f$ itself is good for natural classification tasks. However, in practice, the linearity between $f$ and $\Phi p_f$ only weakly holds on features from pretrained GPT-2 (Radford et al., 2018). The fractional residual norm of the best linear fit, i.e. $r = \frac{\mathbb{E}_{s \sim p} \|\Phi p_f(s) - \boldsymbol{A}f(s) - \boldsymbol{b}\|^2}{\mathbb{E}_{s \sim p} \|\Phi p_f(s)\|^2}$, measured for different distributions ($r = 0$ is perfect fit) are 0.28 for SST, 0.39 for AG News, and 0.18 for IMDb contexts. This non-trivial linear relationship, although surprising, might not completely explain the success of $f$, which usually performs better than $\Phi p_f$; we leave exploring this to future work.

## 5 EXTENSIONS

### 5.1 BETTER HANDLING OF DISTRIBUTIONAL SHIFT

The bounds in the previous section use the coefficient $\gamma(p_{\mathcal{T}})$ to transfer guarantees from $p_L$ to $p_{\mathcal{T}}$ and we define a more refined notion of transferability here. The coefficient $\gamma(p_{\mathcal{T}})$ is independent of the learned model and assumes a worst case distribution of errors. For the refined coefficient, we first define the error made in predicted probabilities by a softmax language model $f$ as $\Delta_{\{\boldsymbol{p}_{f(s)}\}}(s) = \boldsymbol{p}_{f(s)} - \boldsymbol{p}^*_{\cdot|s}$. For any distribution $p \in \Delta_{\mathcal{S}}$, we define uncentered covariance of a function $g : \mathcal{S} \to \mathbb{R}^D$ as $\Sigma_p(g) = \mathbb{E}_{s \sim p} \left[ g(s)g(s)^\top \right]$. The refined transferability coefficient is then defined as

$$\gamma(p; \Phi p_f) := \left( \left\| \Sigma_{p_L}(\Phi \Delta_{\{\boldsymbol{p}_{f(s)}\}})^{-\frac{1}{2}} \Sigma_p(\Phi \Delta_{\{\boldsymbol{p}_{f(s)}\}}) \Sigma_{p_L}(\Phi \Delta_{\{\boldsymbol{p}_{f(s)}\}})^{-\frac{1}{2}} \right\|_2 \right)^{-1}$$

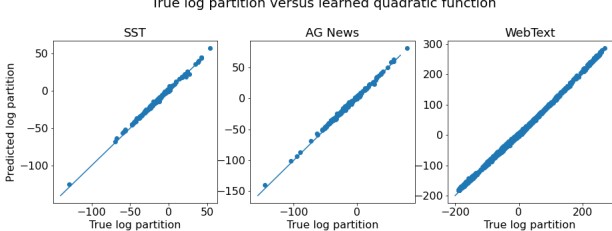

Figure 1: Learned quadratic function v/s log partition function on various datasets for features computed from pre-trained GPT-2 to verify Assumption 4.1. We also plot the $y = x$ line for reference.

We state the refined result for softmax language models; detailed results are deferred to Section B.

**Theorem 5.1** (Simplified). *In the same setting as Theorem 4.2, $\ell_{\mathcal{T}}(\Phi p_f) \leq \tau + \sqrt{\frac{2B^2\epsilon}{\gamma(p_{\mathcal{T}};\Phi p_f)}}$*

It is easy show that $\gamma(p_{\mathcal{T}};\Phi p_f) \geq \gamma(p_{\mathcal{T}})$, so this is indeed a stronger bound. The coefficient $\gamma(p_{\mathcal{T}};\Phi p_f)$ measures how average error on $f$ on $p_L$ can propagate to $p_{\mathcal{T}}$. This can potentially be much smaller than $\gamma(p_{\mathcal{T}})$ due to some inductive biases of $f$. For instance, if errors made by the model are random-like, i.e. $\Delta_{\{\boldsymbol{p}_{f(s)}\}}(s) \sim \rho$, *independently of $s$*, then $\Sigma_{p_L}(\Phi\Delta_{\{\boldsymbol{p}_{f(s)}\}}) \approx \Sigma_p(\Phi\Delta_{\{\boldsymbol{p}_{f(s)}\}}) \approx \mathbb{E}_{\eta\sim\rho}[\eta\eta^\top]$, making $\gamma(p;\Phi p_f) \approx 1$. Independence prevents accumulation of language modeling error on contexts from $p_{\mathcal{T}}$, bypassing the worst case transfer of $\gamma(p_{\mathcal{T}})$.

## 5.2 QUAD: A NEW OBJECTIVE FUNCTION

In Definition 3.2 we discuss how low dimensional softmax language models learn a linear projection of $\boldsymbol{p}^*_{\cdot|s}$, only solving tasks that lie in the row span of word embeddings $\Phi$. Although $\Phi$ defines tasks that language model features can solve, the standard cross-entropy objective does not lend a simple closed form expression for optimal $\Phi$. This motivates the construction of our Quad objective, that has two nice properties: (1) the optimal feature map $f^*$ is a linear function of $\boldsymbol{p}^*_{\cdot|s}$ and thus can solve some natural tasks, and (2) the optimal $\Phi^*$ has an intuitively meaningful closed-form solution.

$$\ell_{quad}(f,\Phi) = \mathop{\mathbb{E}}_{s\sim p_L}\left[\mathop{\mathbb{E}}_{w\sim p^*_{\cdot|s}}[-f(s)^\top\phi_w] + \frac{1}{2}\|\Phi^\top f(s)\|^2\right] \tag{6}$$

The Quad objective is very similar to the cross-entropy objective from Equation (2), with the log partition function replaced by a quadratic function, inspired in part by Assumption 4.1. We can derive the optimal solution $\Phi^*$ that depends on the eigen-decomposition of a *substitutability matrix*.

**Definition 5.1.** *The substitutability matrix is defined to be $\Omega^* := \mathop{\mathbb{E}}_{s\sim p_L}\left[\boldsymbol{p}^*_{\cdot|s}\boldsymbol{p}^{*\top}_{\cdot|s}\right] \in \mathbb{R}^{V\times V}$. If $\Omega^* = \boldsymbol{U}\boldsymbol{S}\boldsymbol{U}^\top$ is the eigendecomposition, then $\boldsymbol{U}_d \in \mathbb{R}^{V\times d}$ is matrix of top $d$ eigenvectors of $\Omega^*$.*

The matrix $\Omega^*$ captures substitutability between pairs of words. Words $w$ and $w'$ are substitutable if they have identical conditional probabilities for every context $s \in \mathcal{S}$ and thus can replace occurrences of each other while still providing meaningful completions. By definition, these words satisfy $\Omega^*[w] = \Omega^*[w']$. Such pairs of words were called "free variants" in the work on distributional semantics (Harris, 1954), and capture the notion of synonyms; more in Section D.2.

**Theorem 5.2.** *Let $f^*, \Phi^* = \arg\min_{f,\Phi} \ell_{quad}(f,\Phi)$. Then $\Phi^* = \boldsymbol{B}\boldsymbol{U}_d^\top$, for full rank $\boldsymbol{B} \in \mathbb{R}^{d\times d}$. Also, for a classification task $\mathcal{T}$ that is $(\tau, B)$-natural w.r.t. $\Phi^*$, we have $\ell_{\mathcal{T}}(f^*) \leq \tau$.*

Thus $f^*$ excels on natural tasks w.r.t. $\Phi^*$, which in turn, is the best $d$-dimensional projection of $\Omega^*$. Thus words $w, w' \in \mathcal{W}$ that are synonyms (hence substitutable) will satisfy $\phi^*_w = \phi^*_{w'}$, fulfilling the desired property for word embeddings discussed in Definition 3.2.

We train using the Quad objective and compare its performance to a similarly trained GPT-2 language model. The results in Table 3 suggest that Quad performs comparably to $\Phi p_f$ from the cross-entropy objective, which fits our theory since both are linear functions of $\boldsymbol{p}^*_{\cdot|s}$. Section F.3 has more details and experiments. The goal of testing Quad is to demonstrate that theoretical insights can aid the design of provably effective algorithms. Refer to Section C for more details on Quad.

Table 1: Accuracy (%) on $k$-way *linear classification* using fixed GPT-2 features. Good performance of features $f(s)$, conditional mean features $\Phi p_f(s)$ and meaningful subset of $\leq 30$ (and $\leq 2k$) coordinates of $\boldsymbol{p}_{f(s)}$ verify the sentence completion reformulation and main results. The numbers right below the features denote dimensionality of the features. An asterisk indicates that we added a task-specific prompt. Other baselines are fine-tuning (FT, Section F.2) and random projection of $\boldsymbol{p}_{f(s)}$ (rand. proj.). Sentence version of SST (train/test: 6.9K/1.8K) is used.

| Task | $k$ | $f(s)$ 768 | $\Phi p_f(s)$ 768 | $\boldsymbol{p}_{f(s)}$ (subset) $\leq 30$ | $\boldsymbol{p}_{f(s)}$ (class words) $\leq 2k$ | $\boldsymbol{p}_{f(s)}$ (rand. proj.) 768 | FT |
|---|---|---|---|---|---|---|---|
| SST | 2 | 87.5 | 83.3 | 82.6 | 78.7 | 67.5 | 91.4 |
| SST* | 2 | 89.4 | 87.3 | 85.4 | 79.1 | 76.4 | 92.3 |
| SST fine | 5 | 49.2 | 43.5 | 44.0 | 39.2 | 23.1 | 50.2 |
| SST fine* | 5 | 49.4 | 48.6 | 47.6 | 40.3 | 28.8 | 53.5 |
| AG | 4 | 90.7 | 84.6 | 83.8 | 75.4 | 58.5 | 94.5 |
| AG* | 4 | 91.1 | 88.2 | 86.1 | 75.1 | 63.7 | 94.4 |

## 6 EXPERIMENTS

We use experiments to verify (1) linear classification on fixed language model features does comparably to fine-tuning the features, (2) sentence completion reformulation (Section 3.1), i.e. tasks can be solved using probabilities for indicative words, (3) conditional mean features are effective.

**Tasks using linear function of $\boldsymbol{p}^*_{\cdot|s}$:** We validate our claims from Section 3 that classification tasks can be solved by linear functions of $\boldsymbol{p}^*_{\cdot|s}$. Since $\boldsymbol{p}^*_{\cdot|s}$ is never available, we instead use the output features $f(s)$ and probabilities $\boldsymbol{p}_{\cdot|s} := \boldsymbol{p}_{f(s)}$ from a small pretrained GPT-2 model (Radford et al., 2019). Table 1 demonstrates that on binary and fine-grained Stanford Sentiment Treebank (SST) (Socher et al., 2013) and AG News (Zhang et al., 2015) tasks, probabilities $\boldsymbol{p}_{f(s)}$ of just 30 or so task-relevant tokens (see Section F.1) can solve the tasks. Even just one/two token per class ("class words") yields non-trivial performance. Furthermore, we validate the sentence completion reformulation in Section 3.1 by using the probabilities $\boldsymbol{p}_{f(s)}$ after adding a task specific prompt and consistently observing improved performance, including for fine-tuning (FT) with small datasets.

**$\Phi p_f$ and $f$ are good features:** We first note that linear classification over fixed features $f(s)$ from the pretrained model performs comparably to the FT baseline. We further validate Theorem 4.2 by verifying that the conditional mean features $\Phi p_f(s)$ also linearly solve downstream tasks fairly well. This performance is comparable to, but always worse than $f(s)$, as seen in columns 3 and 4 of Table 1. We again find that adding a prompt improves performance. Note that a random projection of $\boldsymbol{p}_{f(s)}$ to same dimensions as $\Phi p_f(s)$ has very poor performance. Section E.5 has results for a wider range of classification tasks. Evidence for Assumption 4.1 is provided by learning a quadratic function to fit the log partition function of features from pretrained GPT-2 model (see Section F.4). Figure 1 demonstrates that the fit holds for its training and unseen data (e.g., WebText (Radford et al., 2019)).

## 7 CONCLUSIONS AND FUTURE WORK

We provide intuitive and mathematical explanations for the success of language model features on classification tasks by reformulating them as sentence completion problems. This reformulation is formalized as *natural tasks*: those that can be solved linearly using the conditional probability distribution $\boldsymbol{p}^*_{\cdot|s}$. Insights from our analysis help design the Quad objective that provably learns good features for these natural tasks. We hope our analysis will inspire other mathematical insights into language models. While Section 4.3 argues linearity between conditional mean features $\Phi p_f$ and $f$, it is insufficient to explain the observed superiority of $f$ over $\Phi p_f$. We leave exploring this limitation of our analysis to future work. Guarantees for softmax models are for natural tasks w.r.t. $\Phi$, thus knowing the optimal $d$-dimensional word embeddings $\Phi^*$ for $\ell_{\text{xent}}(f, \Phi)$ is also important. Other meaningful directions include providing guarantees for other successful models like BERT (Devlin et al., 2018) and more diverse downstream tasks. Although we would like to show stronger guarantees by exploiting model and algorithmic inductive biases, as well as study the setting of fine-tuning language model features, lack of a good theory of deep learning is the current bottleneck.

**Acknowledgments:** Sanjeev Arora, Sadhika Malladi and Nikunj Saunshi are supported by NSF, ONR, Simons Foundation, Amazon Research, DARPA and SRC.

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

# A  OVERVIEW

Section B is a more detailed version of Section 5.1 and Section C is a detailed version of Section 5.2. Section D.1 has a discussion about why *natural tasks* are a reasonable formalization for the sentence completion reformulation and also interpretations for $\tau$ and $B$ in the definition of natural tasks. Section D.2 discusses desirable properties of word embeddings $\Phi$ like capturing synonym structure in words. Section E contains proofs for all results, including proof sketches for the main results in Section E.1. Lemma E.4 is the softmax variant of Pinsker's inequality that we prove and use for our main results.

Section F contains many more experimental findings that consolidate many of our theoretical results. Section F.1 provides the information about subsets of words used for results in Table 1 and also additional experiments to test the performance of pretrained language model embeddings $f$ on more downstream tasks and also verifying that conditional mean embeddings $\Phi p_f$ do well on these tasks. In Section F.3, we present additional results for Quad objective trained on a larger corpus and tested on SST. Section F.4 provides additional details on how $\boldsymbol{A}$, $\boldsymbol{b}$ and $c$ from Assumption 4.1 are learned and also further verification of the assumption on more datasets. Finally, Section F.5 experimentally verifies the $\mathcal{O}(\sqrt{\epsilon})$ dependence from Theorem 4.2.

# B  BETTER HANDLING OF DISTRIBUTIONAL SHIFT

While the bounds above used $\gamma(p_\mathcal{T})$ to transfer from the distribution $p_L$ to $p_\mathcal{T}$, we define a more refined notion of transferability here. While $\gamma(p_\mathcal{T})$ only depends on $p_L$ and $p_\mathcal{T}$, the more refined notions depend also on the learned language model, thus potentially exploiting some inductive biases. We first define the notion of error made in the predicted probabilities by any predictor $\boldsymbol{p}_{\cdot|s}$ as $\Delta_{\{\boldsymbol{p}_{\cdot|s}\}}(s) = \boldsymbol{p}_{\cdot|s} - \boldsymbol{p}^*_{\cdot|s}$. Thus for any softmax language model $f$ we have $\Delta_{\{\boldsymbol{p}_{f(s)}\}}(s) = \boldsymbol{p}_{f(s)} - \boldsymbol{p}^*_{\cdot|s}$. For any distribution $p \in \Delta_S$, we define the covariance[5] of a function $g : \mathcal{S} \to \mathbb{R}^D$ as $\Sigma_p(g) = \underset{s \sim p}{\mathbb{E}}\left[g(s)g(s)^\top\right]$. We define 3 coefficients for the results to follow

**Definition B.1.** *For any distribution $p \in \Delta_S$, we define the following*

$$\gamma(p; \{\boldsymbol{p}_{\cdot|s}\}) := \left(\left\|\Sigma_{p_L}(\Delta_{\{\boldsymbol{p}_{\cdot|s}\}})^{-\frac{1}{2}} \Sigma_p(\Delta_{\{\boldsymbol{p}_{\cdot|s}\}})\Sigma_{p_L}(\Delta_{\{\boldsymbol{p}_{\cdot|s}\}})^{-\frac{1}{2}}\right\|_2\right)^{-1} \tag{7}$$

$$\gamma_\Phi(p; \{\boldsymbol{p}_{\cdot|s}\}) := \left(\left\|\Sigma_{p_L}(\Phi\Delta_{\{\boldsymbol{p}_{\cdot|s}\}})^{-\frac{1}{2}} \Sigma_p(\Phi\Delta_{\{\boldsymbol{p}_{\cdot|s}\}})\Sigma_{p_L}(\Phi\Delta_{\{\boldsymbol{p}_{\cdot|s}\}})^{-\frac{1}{2}}\right\|_2\right)^{-1} \tag{8}$$

$$\gamma(p; \Phi p_f) := \gamma_\Phi(p; \{\boldsymbol{p}_{f(s)}\}) \tag{9}$$

We notice that $\Sigma_p(\Delta_{\{\boldsymbol{p}_{\cdot|s}\}}) = \underset{s \sim p}{\mathbb{E}}\left[(\boldsymbol{p}_{\cdot|s} - \boldsymbol{p}^*_{\cdot|s})(\boldsymbol{p}_{\cdot|s} - \boldsymbol{p}^*_{\cdot|s})^\top\right]$, $\Sigma_p(\Phi\Delta_{\{\boldsymbol{p}_{\cdot|s}\}}) = \Phi\Sigma_p(\Delta_{\{\boldsymbol{p}_{\cdot|s}\}})\Phi^\top$. We are now ready to state the most general results.

**Theorem B.1** (Strengthened Theorem 4.1). *Let $\{\boldsymbol{p}_{\cdot|s}\}$ be a language model that is $\epsilon$-optimal, i.e. $\ell_{xent}(\{\boldsymbol{p}_{\cdot|s}\}) - \ell^*_{xent} \leq \epsilon$ for some $\epsilon > 0$. For a classification task $\mathcal{T}$ that is $(\tau, B)$-natural, we have*

$$\ell_\mathcal{T}\left(\{\boldsymbol{p}_{\cdot|s}\}\right) \leq \tau + \sqrt{\frac{2B^2\epsilon}{\gamma(p_\mathcal{T}; \{\boldsymbol{p}_{\cdot|s}\})}}$$

*For a classification task $\mathcal{T}$ that is $(\tau, B)$-natural w.r.t. $\Phi$, we have*

$$\ell_\mathcal{T}\left(\{\boldsymbol{p}_{\cdot|s}\}\right) \leq \ell_\mathcal{T}\left(\{\Phi\boldsymbol{p}_{\cdot|s}\}\right) \leq \tau + \sqrt{\frac{2B^2\epsilon}{\gamma_\Phi(p_\mathcal{T}; \{\boldsymbol{p}_{\cdot|s}\})}}$$

**Theorem 5.1** (Strengthened Theorem 4.2). *For a fixed $\Phi$, let $f$ be features from an $\epsilon$-optimal $d$-dimensional softmax language model, i.e. $\ell_{xent}(f, \Phi) - \ell^*_{xent}(\Phi) \leq \epsilon$, where $\ell^*_{xent}(\Phi)$ is defined in Equation (4). For a classification task $\mathcal{T}$ that is $(\tau, B)$-natural w.r.t. $\Phi$, we have*

$$\ell_\mathcal{T}\left(\{\boldsymbol{p}_{f(s)}\}\right) \leq \ell_\mathcal{T}(\Phi p_f) \leq \tau + \sqrt{\frac{2B^2\epsilon}{\gamma(p_\mathcal{T}; \Phi p_f)}}$$

---

[5]This is not exactly the covariance since the mean is not subtracted, all results hold even for the usual covariance.

**Discussions:** It is not hard to show that the coefficients satisfy $\gamma_\Phi(p_\mathcal{T}; \{\boldsymbol{p}_{\cdot|s}\}) \geq \gamma(p_\mathcal{T}; \{\boldsymbol{p}_{\cdot|s}\}) \geq \gamma(p_\mathcal{T})$ and $\gamma(p_\mathcal{T}; \Phi p_f) \geq \gamma(p_\mathcal{T})$, thus showing that these results are strictly stronger than the ones from the previous section. The transferability coefficient is a measure of how guarantees on $p_L$ using a language model can be transferred to another distribution of contexts and it only depends on the distribution of contexts and not the labels. Unlike $\gamma(p_\mathcal{T})$, the coefficients in Definition B.1 depend on the learned models, either $\{\boldsymbol{p}_{\cdot|s}\}$ or $\{\boldsymbol{p}_{f(s)}\}$, and can be potentially much smaller due to the inductive bias of the learned models. For instance, if errors made by the model are random-like, i.e. $\Delta_{\{\boldsymbol{p}_{\cdot|s}\}}(s) \sim \rho$, independently of $s$, then $\Sigma_{p_L}(\Delta_{\{\boldsymbol{p}_{\cdot|s}\}}) \approx \Sigma_p(\Delta_{\{\boldsymbol{p}_{\cdot|s}\}}) \approx \mathbb{E}_{\eta \sim \rho}[\eta\eta^\top]$, making $\gamma(p; \{\boldsymbol{p}_{\cdot|s}\}) \approx 1$. Independence prevents language modeling error from accumulating on contexts from $p_\mathcal{T}$, bypassing the worst case transfer of $\gamma(p_\mathcal{T})$.

## C   QUAD: A NEW OBJECTIVE FUNCTION

In Definition 3.2 we discuss how low dimensional softmax language models learn a linear projection of $\boldsymbol{p}^*_{\cdot|s}$, only solving tasks that lie in the row span of word embeddings $\Phi$. Although $\Phi$ defines tasks that language model features can solve, the standard cross-entropy objective does not lend a simple closed form expression for optimal $\Phi$. This motivates the construction of our Quad objective, that has two nice properties: (1) the optimal feature map $f^*$ is a linear function of $\boldsymbol{p}^*_{\cdot|s}$ and thus can solve some natural tasks, and (2) the optimal $\Phi^*$ has an intuitively meaningful closed-form solution.

$$\ell_{quad,s}(\theta, \Phi) = \mathbb{E}_{w \sim p^*_{\cdot|s}} [-\theta^\top \phi_w] + \frac{1}{2}\|\Phi^\top \theta\|^2 = -\theta^\top \Phi \boldsymbol{p}^*_{\cdot|s} + \frac{1}{2}\|\Phi^\top \theta\|^2 \tag{10}$$

$$\ell_{quad}(f, \Phi) = \mathbb{E}_{s \sim p_L} [\ell_{quad,s}(f(s), \Phi)] \tag{11}$$

The Quad objective is very similar to the cross-entropy objective from Equation (2), with the log partition function replaced by a quadratic function, inspired in part by Assumption 4.1. We can derive the optimal solution $\Phi^*$ that depends on the eigen-decomposition of a *substitutability matrix*.

**Definition 5.1.** *The substitutability matrix is defined to be* $\Omega^* \coloneqq \mathbb{E}_{s \sim p_L}\left[\boldsymbol{p}^*_{\cdot|s} {\boldsymbol{p}^*_{\cdot|s}}^\top\right] \in \mathbb{R}^{V \times V}$. *If* $\Omega^* = \boldsymbol{U}\boldsymbol{S}\boldsymbol{U}^\top$ *is the eigendecomposition, then* $\boldsymbol{U}_d \in \mathbb{R}^{V \times d}$ *is matrix of top $d$ eigenvectors of* $\Omega^*$.

The matrix $\Omega^*$ captures substitutability between pairs of words. Words $w$ and $w'$ are substitutable if they have identical conditional probabilities for every context $s \in \mathcal{S}$ and thus can replace occurrences of each other while still providing meaningful completions. By definition, these words satisfy $\Omega^*[w] = \Omega^*[w']$. Such pairs of words were called "free variants" in the work on distributional semantics (Harris, 1954), and capture the notion of synonyms in the distributional hypothesis. We now derive expressions for the optimal solution of the Quad objective described in Equation (11). The proof of all results from this section are in Section E.5.

**Theorem C.1.** *The optimal solution* $f^*, \Phi^* = \arg\min_{f,\Phi} \ell_{quad}(f, \Phi)$ *satisfies*

$$\Phi^* = \boldsymbol{B}\boldsymbol{U}_d^\top, \text{ for full rank } \boldsymbol{B} \in \mathbb{R}^{d \times d}$$
$$f^*(s) = (\Phi^* {\Phi^*}^\top)^{-1/2}\Phi^* \boldsymbol{p}^*_{\cdot|s} = \boldsymbol{C}\boldsymbol{U}_d^\top \boldsymbol{p}^*_{\cdot|s}, \text{ for full rank } \boldsymbol{C} \in \mathbb{R}^{d \times d}$$

*If $\Phi$ is fixed, then the optimal solution is* $f^*(s) = (\Phi\Phi^\top)^{-1/2}\Phi \boldsymbol{p}^*_{\cdot|s}$.

**Theorem 5.2.** *Let* $f^*, \Phi^* = \arg\min_{f,\Phi} \ell_{quad}(f, \Phi)$. *Then* $\Phi^* = \boldsymbol{B}\boldsymbol{U}_d^\top$, *for full rank* $\boldsymbol{B} \in \mathbb{R}^{d \times d}$. *Also, for a classification task $\mathcal{T}$ that is $(\tau, B)$-natural w.r.t. $\Phi^*$, we have $\ell_\mathcal{T}(f^*) \leq \tau$.*

Thus $f^*$ excels on natural tasks w.r.t. $\Phi^*$, which in turn, is the best $d$-dimensional projection of $\Omega^*$. Thus words $w, w' \in \mathcal{W}$ that are synonyms (hence substitutable) will satisfy $\phi^*_w = \phi^*_{w'}$, fulfilling the desired property for word embeddings discussed in Definition 3.2. We train using the Quad objective and compare its performance to a similarly trained language model, finding Quad to be reasonably effective. The goal of testing Quad is not to obtain state-of-the-art results, but to demonstrate that theoretical insights can aid the design of provably effective algorithms.

# D    MORE ON NATURAL TASKS

The discussions in this section may not be formal and precise in places, they are meant to provide more intuition for some of the definitions and results.

## D.1    SENTENCE COMPLETION REFORMULATION ≡ NATURAL TASK

We provide informal justification for why the sentence completion reformulation can be formalized as being able to solve using a linear classifier over $\boldsymbol{p}^*_{\cdot|s} \in \mathbb{R}^V$. The analysis will also end up providing some intuitions for $\tau$ and $B$ in Definition 3.1 and Theorem 4.1. In particular, we will show that a task that is amenable to the sentence completion reformulation will be $(\tau, B)$-natural, with $\tau = \mathcal{O}(\textbf{Bayes-Error}(\mathcal{T}))$, i.e. $\tau$ is small if the Bayes error for the task error, and $B = \mathcal{O}(\alpha(\mathcal{W}_{\text{indicative}})^{-1})$ is inversely proportional to the probability mass of the set of indicative words for the task. This is formalized in Proposition D.2.

**Linear classifier over $\boldsymbol{p}^*_{\cdot|s}$**

Consider a binary classification task $\mathcal{T}$ and that can be solved with a sentence completion reformulation after adding a prompt as in Section 3.1, for e.g. sentiment classification can be solved by adding a prompt "This movie is" at the end of every movie review and use the completions to solve the task. Recall that $p_{\mathcal{T}}$ is the distribution over $\mathcal{S} \times \{\pm 1\}$ for the task $\mathcal{T}$. We abuse notation and use $p_{\mathcal{T}}$ to denote the distribution over inputs where a prompt is added to each to input, for e.g. "I loved the movie." is transformed to "I loved the movie. This movie is". For any $s \sim p_{\mathcal{T}}$, let $p_{\mathcal{T}}(y = 1|s)$ and $p_{\mathcal{T}}(y = -1|s)$ denote the conditional probabilities of the sentiment of review $s$ (with an added prompt) being positive and negative respectively. By law of total probability we can write this conditional probability as

$$p_{\mathcal{T}}(y = 1|s) = \sum_{w \in \mathcal{W}} \Pr(y = 1|(s, w)) \Pr(w|s) = \sum_{w \in \mathcal{W}} \Pr(y = 1|(s, w)) \, p^*_{\cdot|s}(w) \qquad (12)$$

For any task $\mathcal{T}$ we can roughly partition the vocabulary set $\mathcal{W}$ into the following

**Indicative words $\mathcal{W}_{\text{indicative}}$:** $w$ can be an *indicative completion* for the task, like "good", "boring", "trash" etc, after a movie review like $s =$"I loved the movie. This movie is". In this case the sentence completion reformulation can be interpreted as the following: the completion $w$ after a review $s$ is sufficient to determine the sentiment of the review, i.e. we do not need to know the content of the review $s$ to predict the label if we know the completion $w$. This can be formalized as $\Pr(y = 1|(s, w)) \approx P(y = 1|w)$ for some fixed distribution $P$ for indicative completions $w$.

**Irrelevant words $\mathcal{W}_{\text{irrelevant}}$:** $w$ can be an *irrelevant completion* for the task, like "a", "very", "not". In this case the completions, on the other hand, do not reveal anything more about the sentiment for the review than $s$ itself, i.e. $\Pr(y = 1|(s, w)) \approx p_{\mathcal{T}}(y = 1|s)$ for irrelevant completions $w$.

Thus from Equation (12) we get

$$\begin{aligned}
p_{\mathcal{T}}(y = 1|s) &= \sum_{w \in \mathcal{W}_{\text{indicative}}} \Pr(y = 1|(s, w)) \, p^*_{\cdot|s}(w) + \sum_{w \in \mathcal{W}_{\text{irrelevant}}} \Pr(y = 1|(s, w)) \, p^*_{\cdot|s}(w) \\
&\approx \sum_{w \in \mathcal{W}_{\text{indicative}}} P(y = 1|w) \, p^*_{\cdot|s}(w) + \sum_{w \in \mathcal{W}_{\text{irrelevant}}} p_{\mathcal{T}}(y = 1|s) \, p^*_{\cdot|s}(w) \\
&= \sum_{w \in \mathcal{W}_{\text{indicative}}} \boldsymbol{v}_1(w) \, p^*_{\cdot|s}(w) + p_{\mathcal{T}}(y = 1|s) \sum_{w \in \mathcal{W}_{\text{irrelevant}}} p^*_{\cdot|s}(w) \\
&= \boldsymbol{v}_1^\top \boldsymbol{p}^*_{\cdot|s} + p_{\mathcal{T}}(y = 1|s) p^*_{\cdot|s}(\mathcal{W}_{\text{irrelevant}})
\end{aligned}$$

where $\boldsymbol{v}_1 \in \mathbb{R}^V$ is defined as $\boldsymbol{v}_1(w) = P(y = 1|w)$ for $w \in \mathcal{W}_{\text{indicative}}$ and $\boldsymbol{v}_1(w) = 0$ for $w \in \mathcal{W}_{\text{irrelevant}}$. Similarly we can define $\boldsymbol{v}_{-1} \in \mathbb{R}^V$ with $\boldsymbol{v}_{-1}(w) = P(y = -1|w)$ for $w \in \mathcal{W}_{\text{indicative}}$, $\boldsymbol{v}_{-1}(w) = 0$ for $w \in \mathcal{W}_{\text{irrelevant}}$. From the earlier calculation, and a similar one for $y = -1$, we get

$$p_{\mathcal{T}}(y = b|s) \approx \frac{1}{1 - p^*_{\cdot|s}(\mathcal{W}_{\text{irrelevant}})} \boldsymbol{v}_b^\top \boldsymbol{p}^*_{\cdot|s} = \frac{1}{p^*_{\cdot|s}(\mathcal{W}_{\text{indicative}})} \boldsymbol{v}_b^\top \boldsymbol{p}^*_{\cdot|s}, \text{ for } b \in \{\pm 1\}$$

If we assume $p^*_{\cdot|s}(\mathcal{W}_{\text{indicative}}) \approx \alpha(\mathcal{W}_{\text{indicative}})$ is roughly the same for all $s$, i.e. probability mass of indicative words following a modified review is approximately the same, then we get

$$p_{\mathcal{T}}(y=1|s) - p_{\mathcal{T}}(y=-1|s) \approx \boldsymbol{v}_{\mathcal{T}}^\top \boldsymbol{p}^*_{\cdot|s} \text{ , where } \boldsymbol{v}_{\mathcal{T}} = \frac{1}{\alpha(\mathcal{W}_{\text{indicative}})}(\boldsymbol{v}_1 - \boldsymbol{v}_{-1}) \qquad (13)$$

Thus we can approximately express the difference in conditional probabilities of the 2 classes as a linear function of $\boldsymbol{p}^*_{\cdot|s}$. While it is intuitively clear why knowing $p_{\mathcal{T}}(y=1|s) - p_{\mathcal{T}}(y=-1|s)$ is useful for solving the task, we show precisely why in the next part.

**Interpretation for $\tau$ and $B$**

Based on the above discussed, we will show that the task $\mathcal{T}$ from earlier is $(\tau, B)$-natural according to the Definition 3.1 and will also give us an interpretation for $\tau$ and $B$. First we show that the following predictor from Equation (13) is effective for task $\mathcal{T}$

$$g_{\mathcal{T}}(s) = p_{\mathcal{T}}(y=1|s) - p_{\mathcal{T}}(y=-1|s) \approx \boldsymbol{v}_{\mathcal{T}}^\top \boldsymbol{p}^*_{\cdot|s} \qquad (14)$$

We reuse the notation from Equation (3) and define the task loss for any predictor $g : \mathcal{S} \to \mathbb{R}$ as

$$\ell_{\mathcal{T}}(g) = \mathbb{E}_{(s,y)\sim p_{\mathcal{T}}}\left[\ell(g(s), y)\right] \qquad (15)$$

Furthermore let **Bayes-Error**$(\mathcal{T}) \coloneqq \inf_{g:\mathcal{S}\to\mathbb{R}} \mathbb{E}_{(s,y)\sim p_{\mathcal{T}}}[\mathbf{1}\{g(s) \neq y\}]$ denote the Bayes error of the task $\mathcal{T}$, i.e. the optimal $0-1$ error achievable on the task.

**Proposition D.1.** *For any task $\mathcal{T}$ and for the hinge loss $\ell$, $\ell_{\mathcal{T}}(g_{\mathcal{T}}) \leq 4$ **Bayes-Error**$(\mathcal{T})$, where $g_{\mathcal{T}}(s) = p_{\mathcal{T}}(y=1|s) - p_{\mathcal{T}}(y=-1|s)$.*

Thus if a task is easily solvable, i.e. has small Bayes error, then it will be solvable by the predictor $g_{\mathcal{T}}(s)$. Since we argued above that sentence reformulation implies that $g_{\mathcal{T}}(s)$ is a linear function of $\boldsymbol{p}^*_{\cdot|s}$, we can now show that $\mathcal{T}$ is a *natural task* as formalized in Definition 3.1.

**Proposition D.2** (Informal). *Task $\mathcal{T}$ that can be reformulated as a sentence completion task (described above) is a $(\tau, B)$-natural task w.r.t. the hinge loss, with the follow parameters*

$$\tau \leq 4 \text{ **Bayes-Error**}(\mathcal{T}) \text{ and } B = \alpha(\mathcal{W}_{\text{indicative}})^{-1}$$

*Here **Bayes-Error**$(\mathcal{T})$ is the Bayes error of task $\mathcal{T}$ and $\alpha(\mathcal{W}_{indicative})$ is the total mass of the indicative words for the task.*

If the task $\mathcal{T}$ can be reformulated as sentence completion, then $\mathcal{T}$ is $(\tau, B)$-natural where

- $\tau$ is small if the task is unambiguous, i.e. it has small Bayes error
- $B$ is small if the probability mass of the set of indicative words $\mathcal{W}_{\text{indicative}}$ is large, i.e. the task depends on a large set of frequent words

Thus the upper bound in Theorem 4.1 is smaller if the task can be reformulated as sentence completion task with a large and frequent set of completions, and we can ever hope to solve it well (Bayes error is small). The proofs for the above propositions are in Section D.1.

### D.2 NICE PROPERTIES OF WORD EMBEDDINGS $\Phi$

We argue here that if the word embeddings $\Phi$ satisfy certain *nice properties*, then $(\tau, B)$-natural *tasks of interest* will be $(\tau', B')$-natural w.r.t. $\Phi$, where we will provide informal quantifications for the *nice properties* and *tasks of interest* that lead to a small value for $\tau'$ and $B'$. The *nice property* will be related to $\Phi$ capturing the semantic meaning (synonym structure) of words and *tasks of interest* will be those that try to distinguish word completion (in the sentence completion reformulation) with very different meanings, i.e. tries to distinguish more coarse-grained semantic notions rather than very fine-grained ones. Note that the results here are informal and qualitative, rather than quantitative.

Consider a task $\mathcal{T}$ that is $(\tau, B)$-natural task and let $\boldsymbol{v}^* \in \mathbb{R}^V$ be the classifier such that $\ell_{\mathcal{T}}(\{\boldsymbol{p}^*_{\cdot|s}\}, \boldsymbol{v}^*) \leq \tau$ and $\|\boldsymbol{v}^*\|_\infty \leq B$. We want to find properties of $\Phi$ and $\boldsymbol{v}^*$ that will make $\mathcal{T}$ to be $(\tau', B')$-natural w.r.t. $\Phi$ such that $\tau'$ and $B'$ are not too large.[6]

---

[6]Note that the converse is trivially true, i.e. a $(\tau, B)$-natural task w.r.t. $\Phi$ is also $(\tau, B)$-natural.

We will show that $\mathcal{T}$ is $(\tau', B')$-natural w.r.t. $\Phi$ by finding a classifier $\boldsymbol{v}$ such that $\boldsymbol{v} = \Phi^\top \lambda \in \mathbb{R}^V$, $\|\boldsymbol{v}\|_\infty \le B'$ and $\ell_{\mathcal{T}}(\{\boldsymbol{p}^*_{\cdot|s}\}, \boldsymbol{v}) \le \tau'$. First we define $P_\Phi := \Phi^\dagger \Phi \in \mathbb{R}^{V \times V}$ to be the projection matrix for the row-span of $\Phi$ and $P_\Phi^\perp := I_V - P_\Phi$ to be orthogonal projection matrix. We will show that the classifier $\boldsymbol{v} = P_\Phi \boldsymbol{v}^*$ suffices for our case, under some intuitive conditions on $\boldsymbol{v}^*$ and $\Phi$.

To compute $B'$, we first look at the $\ell_\infty$ norm of $\boldsymbol{v} = P_\Phi \boldsymbol{v}^*$

$$B' = \|\boldsymbol{v}\|_\infty = \|P_\Phi \boldsymbol{v}^*\|_\infty = \|\boldsymbol{v}^* - P_\Phi^\perp \boldsymbol{v}^*\|_\infty \le \|\boldsymbol{v}^*\|_\infty + \|P_\Phi^\perp \boldsymbol{v}^*\|_\infty \le B + \|P_\Phi^\perp \boldsymbol{v}^*\|_2$$

To find the upper bound $\tau'$, we upper bound the classification loss of $\boldsymbol{v} = P_\Phi \boldsymbol{v}^*$. We first define the substitutability matrix $\Omega_p^* = \mathbb{E}_{s \sim p}\left[\boldsymbol{p}^*_{\cdot|s} \boldsymbol{p}^{*\top}_{\cdot|s}\right]$, similar to the one in Definition 5.1. Then

$$
\begin{aligned}
\ell_{\mathcal{T}}(\{\boldsymbol{p}^*_{\cdot|s}\}, \boldsymbol{v}) &= \mathbb{E}_{(s,y) \sim p_{\mathcal{T}}}\left[\ell(\boldsymbol{v}^\top \boldsymbol{p}^*_{\cdot|s}, y)\right] = \mathbb{E}_{(s,y) \sim p_{\mathcal{T}}}\left[\ell((P_\Phi \boldsymbol{v}^*)^\top \boldsymbol{p}^*_{\cdot|s}, y)\right] \\
&\le^{(a)} \mathbb{E}_{(s,y) \sim p_{\mathcal{T}}}\left[\ell(\boldsymbol{v}^{*\top} \boldsymbol{p}^*_{\cdot|s}, y)\right] + \mathbb{E}_{s \sim p_{\mathcal{T}}}[|(\boldsymbol{v}^* - P_\Phi \boldsymbol{v}^*)^\top \boldsymbol{p}^*_{\cdot|s}|] \\
&= \ell_{\mathcal{T}}(\{\boldsymbol{p}^*_{\cdot|s}\}, \boldsymbol{v}^*) + \mathbb{E}_{s \sim p_{\mathcal{T}}}\left[|\boldsymbol{v}^{*\top} P_\Phi^\perp \boldsymbol{p}^*_{\cdot|s}|\right] \\
&\le^{(b)} \tau + \sqrt{\mathbb{E}_{s \sim p_{\mathcal{T}}}\left[(\boldsymbol{v}^{*\top} P_\Phi^\perp \boldsymbol{p}^*_{\cdot|s})^2\right]} = \tau + \sqrt{\mathbb{E}_{s \sim p_{\mathcal{T}}}\left[\boldsymbol{v}^{*\top} P_\Phi^\perp \boldsymbol{p}^*_{\cdot|s} \boldsymbol{p}^{*\top}_{\cdot|s} P_\Phi^\perp \boldsymbol{v}^*\right]} \\
&=^{(c)} \tau + \sqrt{\boldsymbol{v}^{*\top} P_\Phi^\perp \Omega^*_{p_{\mathcal{T}}} P_\Phi^\perp \boldsymbol{v}^*} \le^{(d)} \tau + \|P_\Phi^\perp \boldsymbol{v}^*\|_2 \sqrt{\left\|P_\Phi^\perp \Omega^*_{p_{\mathcal{T}}} P_\Phi^\perp\right\|_2}
\end{aligned}
$$

where $(a)$ follows from 1-Lipschitz property of $\ell$, $(b)$ from Jensen's inequality and that $\ell_{\mathcal{T}}(\{\boldsymbol{p}^*_{\cdot|s}\}, \boldsymbol{v}^*) \le \tau$, $(c)$ from the definition of substitutability matrix $\Omega^*_{p_{\mathcal{T}}}$ and $(d)$ by definition of spectral norm of a symmetric PSD matrix.

Thus we have shown that $\mathcal{T}$ is $(\tau', B')$-natural w.r.t. $\Phi$, where

$$\tau' = \tau + \|P_\Phi^\perp \boldsymbol{v}^*\|_2 \sqrt{\left\|P_\Phi^\perp \Omega^*_{p_{\mathcal{T}}} P_\Phi^\perp\right\|_2}, \; B' = B + \|P_\Phi^\perp \boldsymbol{v}^*\|_2 \tag{16}$$

We will now show that if $\Phi$ captures the notion of synonyms, then $\left\|P_\Phi^\perp \Omega^*_{p_{\mathcal{T}}} P_\Phi^\perp\right\|_2$ will be small leading to $\tau'$ being small. Furthermore we also shed some light on what it means for $\|P_\Phi^\perp \boldsymbol{v}^*\|_2$ to be small, which will in turn make $B'$ small and $\tau'$ smaller. We do so with the following arguments, 1) $\Omega^*_{p_{\mathcal{T}}}$ captures semantic meaning of words and thus its top eigen-directions will capture more dominant semantic concepts, 2) if $\Phi$ captures the "top-$d$" directions of meaning, i.e. the top-$d$ eigen-directions of $\Omega^*_{p_{\mathcal{T}}}$, then $\left\|P_\Phi^\perp \Omega^*_{p_{\mathcal{T}}} P_\Phi^\perp\right\|_2 = \mathcal{O}(1/d)$, 3) if additionally $\boldsymbol{v}^*$ cares about the "top-$d$" directions of meaning, i.e. top-$d$ eigen-directions of $\Omega^*_{p_{\mathcal{T}}}$ then $\|P_\Phi^\perp \boldsymbol{v}^*\|_2$ will be small. We expand on these points below

1. **Substitutability matrix ($\Omega^*_{p_{\mathcal{T}}}$) captures semantic meaning:** We use a similar argument to the one in Section 5.2 right after Definition 5.1 that is based on distributional semantics (Harris, 1954). Harris (1954) posits that meaning for elements (words) can be derived from the environments (contexts) in which they occur. Thus Harris (1954) argues that words that occur in almost identical set of contexts have the same meaning, i.e. are synonyms. On the other hand, if two words share some contexts but not all, then they have different meanings and the amount of difference in meaning roughly corresponds to amount of difference in contexts. In our setting, the similarity of words $w$ and $w'$ can then be determined by the probabilities assigned to them by different contexts $s$. In particular, if $\boldsymbol{p}^*_{\cdot|s}(w) = \boldsymbol{p}^*_{\cdot|s}(w')$ for all or most $s \in \text{supp}(p_{\mathcal{T}})$, then $w$ and $w'$ have essentially the same meaning w.r.t. the distribution of contexts $p_{\mathcal{T}}$ and the closer $[\boldsymbol{p}^*_{\cdot|s}(w)]_{s \in \text{supp}(p_{\mathcal{T}})}$ and $[\boldsymbol{p}^*_{\cdot|s}(w')]_{s \in \text{supp}(p_{\mathcal{T}})}$ are, the closer the meaning of $w$ and $w'$ are. For the substitutability matrix $\Omega^*_{p_{\mathcal{T}}} = \mathbb{E}_{s \sim p_{\mathcal{T}}}[\boldsymbol{p}^*_{\cdot|s} \boldsymbol{p}^{*\top}_{\cdot|s}] \in \mathbb{R}^{V \times V}$, it is not hard to show that $\Omega^*_{p_{\mathcal{T}}}(w) = \Omega^*_{p_{\mathcal{T}}}(w')$ is equivalent to $\boldsymbol{p}^*_{\cdot|s}(w) = \boldsymbol{p}^*_{\cdot|s}(w') \; \forall s \sim p_{\mathcal{T}}$, where $\Omega^*_{p_{\mathcal{T}}}(w)$ is the row of $\Omega^*_{p_{\mathcal{T}}}$ corresponding to word $w$. To show this, we can define $\boldsymbol{\beta}_w \in \mathbb{R}^{|\text{supp}(p_{\mathcal{T}})|}$ to be an embedding of $w$ that looks like $\boldsymbol{\beta}_w = [\boldsymbol{p}^*_{\cdot|s}(w) \sqrt{p_{\mathcal{T}}(s)}]_{s \in \text{supp}(p_{\mathcal{T}})}$. It is easy to see that $\boldsymbol{\beta}_{w_1}^\top \boldsymbol{\beta}_{w_2} = \mathbb{E}_{s \sim p_{\mathcal{T}}}\left[\boldsymbol{p}^*_{\cdot|s}(w_1) \boldsymbol{p}^*_{\cdot|s}(w_2)\right] = \Omega^*_{p_{\mathcal{T}}}(w_1, w_2)$. Thus $\boldsymbol{\beta}_w = \boldsymbol{\beta}_{w'} \implies \Omega^*_{p_{\mathcal{T}}}(w) = $

$\Omega^*_{p_\mathcal{T}}(w')$ is straightforward to see. For the converse,

$$\Omega^*_{p_\mathcal{T}}(w) = \Omega^*_{p_\mathcal{T}}(w') \implies \Omega^*_{p_\mathcal{T}}(w, w) = \Omega^*_{p_\mathcal{T}}(w', w) = \Omega^*_{p_\mathcal{T}}(w, w') = \Omega^*_{p_\mathcal{T}}(w', w') \tag{17}$$

$$\implies \boldsymbol{\beta}_w^\top \boldsymbol{\beta}_w = \boldsymbol{\beta}_w^\top \boldsymbol{\beta}_{w'} = \boldsymbol{\beta}_{w'}^\top \boldsymbol{\beta}_{w'} \implies \boldsymbol{\beta}_w = \boldsymbol{\beta}_{w'} \tag{18}$$

Thus $\Omega^*_{p_\mathcal{T}}$ indeed does capture the synonyms structure between words, and the top eigen-directions of it capture the most significant "semantic meaning" directions.

2. $\Phi$ **has nice properties**: if $\Phi$ roughly respects this synonym structure by aligning with the top-$d$ eigen-directions of $\Omega^*_{p_\mathcal{T}}$, we have

$$\left\| P_\Phi^\perp \Omega^*_{p_\mathcal{T}} P_\Phi^\perp \right\|_2 \le \lambda_{d+1}(\Omega^*_{p_\mathcal{T}}) \le \frac{1}{d+1} \sum_{i=1}^{d+1} \lambda_i(\Omega^*_{p_\mathcal{T}}) \le \frac{1}{d+1} \mathrm{tr}(\Omega^*_{p_\mathcal{T}}) \tag{19}$$

$$\le \frac{1}{d+1} \mathop{\mathbb{E}}_{s \sim p_\mathcal{T}} \mathrm{tr}(\boldsymbol{p}^*_{\cdot|s} \boldsymbol{p}^{*\top}_{\cdot|s}) \le \frac{1}{d+1} \tag{20}$$

From Equation (16), we then have $\tau' \le \tau + \frac{\|P_\Phi^\perp \boldsymbol{v}^*\|_2}{\sqrt{d}}$

3. **Tasks of interest**: It is more likely for a classifier $\boldsymbol{v}^*$ to separate words with big differences in meaning rather than small differences. For e.g., it is more likely for a task to separate word completions "good" and "bad" rather than "good" and "nice". Since top eigen-directions of $\Omega^*_{p_\mathcal{T}}$ capture more dominant semantic meanings, this could correspond to $\boldsymbol{v}^*$ aligning with the top eigen-directions of $\Omega^*_{p_\mathcal{T}}$. In combination with the above property about $\Phi$, this could suggest that $\|P_\Phi^\perp \boldsymbol{v}^*\|_2$ is small, thus leading to $\tau'$ and $B'$ being small.

Note that they above arguments are informal and qualitative, and we leave exploring desirable properties of $\Phi$ more formally to future work.

### D.3 Proofs for Section D.1

*Proposition D.1.* Let $p_b(s) = p_\mathcal{T}(y = b|s)$ for $b \in \{\pm 1\}$, $p_{min}(s) = \min_{b \in \{\pm 1\}} p_b(s)$, $p_{max}(s) = \max_{b \in \{\pm 1\}} p_b(s)$ and $g^*(s) = \arg\max_{b \in \{\pm 1\}} p_b(s)$ denote the Bayes optimal predictor. We first notice that there is a simple well-known closed form expression for the Bayes risk

$$\textbf{Bayes-Error}(\mathcal{T}) = \mathop{\mathbb{E}}_{(s,y) \sim p_\mathcal{T}} [\mathbf{1}\{g^*(s) \ne y\}]$$

$$= \mathop{\mathbb{E}}_{(s,y) \sim p_\mathcal{T}} \left[ \mathbf{1}\left\{ \arg\max_{b \in \{\pm 1\}} p_b(s) \ne y \right\} \right] = \mathop{\mathbb{E}}_{s \sim p_\mathcal{T}} [p_{min}(s)]$$

$\square$

We now analyze the hinge loss of the predictor $g_{p_\mathcal{T}}$ defined in Equation (14). Note that since $g_{p_\mathcal{T}}(s) \le 1$, the hinge loss $\ell(g_{p_\mathcal{T}}(s), y) = (1 - y g_{p_\mathcal{T}}(s))_+ = 1 - y g_{p_\mathcal{T}}(s)$ for every $s, y$. Thus the total loss is

$$g_{p_\mathcal{T}}(s) = \mathop{\mathbb{E}}_{(s,y) \sim p_\mathcal{T}} [(1 - y g_{p_\mathcal{T}}(s))_+] = \mathop{\mathbb{E}}_{(s,y) \sim p_\mathcal{T}} [(1 - y g_{p_\mathcal{T}}(s))]$$

$$\overset{(a)}{=} \mathop{\mathbb{E}}_{s \sim p_\mathcal{T}} [p_1(s)(1 - g_{p_\mathcal{T}}(s)) + p_{-1}(s)(1 + g_{p_\mathcal{T}}(s))] = \mathop{\mathbb{E}}_{s \sim p_\mathcal{T}} [1 - (p_1(s) - p_{-1}(s)) g_{p_\mathcal{T}}(s)]$$

$$\overset{(b)}{=} \mathop{\mathbb{E}}_{s \sim p_\mathcal{T}} [1 - (p_1(s) - p_{-1}(s))^2] = \mathop{\mathbb{E}}_{s \sim p_\mathcal{T}} [(p_1(s) + p_{-1}(s))^2 - (p_1(s) - p_{-1}(s))^2]$$

$$= \mathop{\mathbb{E}}_{s \sim p_\mathcal{T}} [4 p_1(s) p_{-1}(s)] = 4 \mathop{\mathbb{E}}_{s \sim p_\mathcal{T}} [p_{min}(s) p_{max}(s)]$$

$$\overset{(c)}{\le} 4 \mathop{\mathbb{E}}_{s \sim p_\mathcal{T}} [p_{min}(s)] = 4\, \textbf{Bayes-Error}(\mathcal{T})$$

where $(a)$ follows by splitting the expectation over $y|s$, $(b)$ follows from the definition of $g_{p_\mathcal{T}}(s)$ in Equation (14) and $(c)$ follows from $p_{max}(s) \le 1$. This completes the proof.

*Proposition D.2.* Let $B = \alpha(\mathcal{W}_{\text{indicative}})^{-1}$. We first note the following using the definition of $\boldsymbol{v}$ from Equation (13).

$$\|\boldsymbol{v}_{\mathcal{T}}\|_{\infty} = \alpha(\mathcal{W}_{\text{indicative}})^{-1} \max_{w \in \mathcal{W}} |\boldsymbol{v}_1(w) - \boldsymbol{v}_{-1}(w)| = B \max_{w \in \mathcal{W}} |P(y = 1|w) - P(y = -1|w)| \leq B \tag{21}$$

To find the value of $\tau$ that makes the task $(\tau, B)$-natural (Definition 3.1), we observe the following

$$\min_{\boldsymbol{v} \in \mathbb{R}^V, \|\boldsymbol{v}\| \leq B} \ell_{\mathcal{T}}(\{\boldsymbol{p}^*_{\cdot|s}\}, \boldsymbol{v}) =^{(a)} \ell_{\mathcal{T}}(\{\boldsymbol{p}^*_{\cdot|s}\}, \boldsymbol{v}_{\mathcal{T}}) = \mathop{\mathbb{E}}_{(s,y) \sim p_{\mathcal{T}}} [\ell(\boldsymbol{v}_{\mathcal{T}}^{\top} \boldsymbol{p}^*_{\cdot|s}, y)]$$

$$=^{(b)} \mathop{\mathbb{E}}_{(s,y) \sim p_{\mathcal{T}}} [\ell(g_{\mathcal{T}}(s), y)] = \ell_{\mathcal{T}}(g_{\mathcal{T}})$$

$$\leq^{(c)} 4 \text{ } \mathbf{Bayes\text{-}Error}(\mathcal{T})$$

where $(a)$ follows from the calculation in Equation (21), $(b)$ follows from Equation (13) and $(c)$ follows from Proposition D.1. □

# E   PROOFS

## E.1   PROOF SKETCH

We first present a sketch of the arguments that help us show our main results, theorems 4.1 and 4.2. The subsections after the next one contain the full proofs for strengthened versions of these results.

### E.1.1   PROOF SKETCH FOR ARBITRARY LANGUAGE MODELS: THEOREM 4.1

Here we want to show guarantees for features $\{\boldsymbol{p}_{\cdot|s}\}$ on a $(\tau, B)$-natural task $\mathcal{T}$. From the definition of natural tasks, we know

$$\exists \boldsymbol{v}^* \in \mathbb{R}^V, \|\boldsymbol{v}^*\|_{\infty} \leq B \text{ s.t. } \ell_{\mathcal{T}}(\{\boldsymbol{p}^*_{\cdot|s}\}, \boldsymbol{v}^*) \leq \tau \tag{22}$$

We wish to upper bound the classification error $\ell_{\mathcal{T}}(\{\boldsymbol{p}_{\cdot|s}\})$ and do so using the following sequence of inequalities.

$$\ell_{\mathcal{T}}(\{\boldsymbol{p}_{\cdot|s}\}) - \tau = \inf_{\boldsymbol{v} \in \mathbb{R}^V} \ell_{\mathcal{T}}(\{\boldsymbol{p}_{\cdot|s}\}, \boldsymbol{v}) - \tau \leq \ell_{\mathcal{T}}(\{\boldsymbol{p}_{\cdot|s}\}, \boldsymbol{v}^*) - \ell_{\mathcal{T}}(\{\boldsymbol{p}^*_{\cdot|s}\}, \boldsymbol{v}^*)$$

$$= \frac{\ell_{\mathcal{T}}(\{\boldsymbol{p}_{\cdot|s}\}, \boldsymbol{v}^*) - \ell_{\mathcal{T}}(\{\boldsymbol{p}^*_{\cdot|s}\}, \boldsymbol{v}^*)}{\sqrt{\mathop{\mathbb{E}}_{s \sim p_{\mathcal{T}}} [(\boldsymbol{v}^{*\top}(\boldsymbol{p}_{\cdot|s} - \boldsymbol{p}^*_{\cdot|s}))^2]}} \cdot \sqrt{\frac{\mathop{\mathbb{E}}_{s \sim p_{\mathcal{T}}} [(\boldsymbol{v}^{*\top}(\boldsymbol{p}_{\cdot|s} - \boldsymbol{p}^*_{\cdot|s}))^2]}{\mathop{\mathbb{E}}_{s \sim p_L} [(\boldsymbol{v}^{*\top}(\boldsymbol{p}_{\cdot|s} - \boldsymbol{p}^*_{\cdot|s}))^2]}} \cdot \sqrt{\mathop{\mathbb{E}}_{s \sim p_L} [(\boldsymbol{v}^{*\top}(\boldsymbol{p}_{\cdot|s} - \boldsymbol{p}^*_{\cdot|s}))^2]}$$

$$= \underbrace{\frac{\ell_{\mathcal{T}}(\{\boldsymbol{p}_{\cdot|s}\}, \boldsymbol{v}^*) - \ell_{\mathcal{T}}(\{\boldsymbol{p}^*_{\cdot|s}\}, \boldsymbol{v}^*)}{\sqrt{\boldsymbol{v}^{*\top} \Sigma_{p_{\mathcal{T}}}(\Delta_{\{\boldsymbol{p}_{\cdot|s}\}}) \boldsymbol{v}^*}}}_{\boldsymbol{\alpha_1}(\boldsymbol{v}^*)} \cdot \underbrace{\sqrt{\frac{\boldsymbol{v}^{*\top} \Sigma_{p_{\mathcal{T}}}(\Delta_{\{\boldsymbol{p}_{\cdot|s}\}}) \boldsymbol{v}^*}{\boldsymbol{v}^{*\top} \Sigma_{p_L}(\Delta_{\{\boldsymbol{p}_{\cdot|s}\}}) \boldsymbol{v}^*}}}_{\boldsymbol{\alpha_2}(\boldsymbol{v}^*)} \cdot \underbrace{\sqrt{\mathop{\mathbb{E}}_{s \sim p_L} [(\boldsymbol{v}^{*\top}(\boldsymbol{p}_{\cdot|s} - \boldsymbol{p}^*_{\cdot|s}))^2]}}_{\boldsymbol{\alpha_3}(\boldsymbol{v}^*)} \tag{23}$$

$\boldsymbol{\alpha_1}(\boldsymbol{v}^*)$: Classification loss → error covariance on $p_{\mathcal{T}}$. Use Lipschitzness of $\ell$ and Jensen's inequality

$\boldsymbol{\alpha_2}(\boldsymbol{v}^*)$: Error covariance from $p_{\mathcal{T}} \to p_L$. Use transferability coefficient

$\boldsymbol{\alpha_3}(\boldsymbol{v}^*)$: Error covariance → cross-entropy loss. Use (modified) Pinsker's inequality

where $\Sigma_p(g) \coloneqq \mathop{\mathbb{E}}_{s \sim p} [g(s)g(s)^{\top}]$ is the uncentered covariance of $g$ w.r.t. distribution $p \in \Delta_{\mathcal{S}}$, as defined in Section 5.1. We upper bound $\ell_{\mathcal{T}}(\{\boldsymbol{p}_{\cdot|s}\}) - \tau$ by upper bounding each of $\boldsymbol{\alpha_1}(\boldsymbol{v}^*), \boldsymbol{\alpha_2}(\boldsymbol{v}^*), \boldsymbol{\alpha_3}(\boldsymbol{v}^*)$ as follows

- **Classification loss → prediction error covariance:** $\boldsymbol{\alpha_1}(\boldsymbol{v}^*)$ is upper bounded by using Lipschitzness of the loss $\ell$ used in the definition of $\ell_{\mathcal{T}}$, e.g. hinge loss or logistic loss, and then followed by an application of Jensen's inequality

$$\text{Lemma E.8} \implies \boldsymbol{\alpha_1}(\boldsymbol{v}) \leq 1 \text{ for all } \boldsymbol{v} \in \mathbb{R}^V$$

- **Error covariance from $p_{\mathcal{T}} \to p_L$:** $\boldsymbol{\alpha_2}(\boldsymbol{v}^*)$ handles the mismatch in distributions $p_{\mathcal{T}}$ and $p_L$ over which the classification loss and cross-entropy losses are measured respectively. It is upper bounded by the transferability coefficient

$$\text{Lemma E.10 and Lemma E.9} \implies \boldsymbol{\alpha_2}(\boldsymbol{v}) \le \sqrt{\gamma(p_\mathcal{T})^{-1}} \text{ for all } \boldsymbol{v} \in \mathbb{R}^V$$

- **Error covariance → cross-entropy loss (arbitrary language models):** This is arguably the most important step that connects the error in prediction to the cross-entropy loss. For the arbitrary language model case, this is proved using Pinsker's inequality and taking expectation over the distribution $p_L$.

$$\text{Lemma E.3} \implies \boldsymbol{\alpha_3}(\boldsymbol{v}) \le \sqrt{2\|\boldsymbol{v}\|_\infty^2 (\ell_{\text{xent}}(\{\boldsymbol{p}_{\cdot|s}\}) - \ell_{\text{xent}}(\boldsymbol{p}_{\cdot|s}^*))} \text{ for all } \boldsymbol{v} \in \mathbb{R}^V$$

### E.1.2 Proof sketch for softmax language models: Theorem 4.2

Here we want to show guarantees for features $\Phi p_f = \{\Phi \boldsymbol{p}_{f(s)}\}$ on a $(\tau, B)$-natural task $\mathcal{T}$ w.r.t $\Phi$. From the definition of natural tasks w.r.t. $\Phi$, we know

$$\exists \boldsymbol{v}^* = \Phi^\top \lambda \in \mathbb{R}^V, \|\boldsymbol{v}^*\|_\infty \le B \text{ s.t. } \ell_\mathcal{T}(\{\boldsymbol{p}_{\cdot|s}^*\}, \boldsymbol{v}^*) \le \tau \tag{24}$$

Note that the difference here is that $\boldsymbol{v}^*$ is in the span of $\Phi$ rather than an arbitrary vector in $\mathbb{R}^V$. We wish to upper bound the classification error $\ell_\mathcal{T}(\{\Phi \boldsymbol{p}_{f(s)}\})$ and do so using the following sequence of inequalities.

$$\begin{aligned}
\ell_\mathcal{T}(\{\Phi \boldsymbol{p}_{f(s)}\}) - \tau &= \inf_{\lambda \in \mathbb{R}^d} \ell_\mathcal{T}(\{\Phi \boldsymbol{p}_{f(s)}\}, \lambda) - \tau \\
&= \inf_{\boldsymbol{v} = \Phi^\top \lambda \in \mathbb{R}^V} \ell_\mathcal{T}(\{\boldsymbol{p}_{f(s)}\}, \boldsymbol{v}) - \tau \\
&\le \ell_\mathcal{T}(\{\boldsymbol{p}_{f(s)}\}, \boldsymbol{v}^*) - \ell_\mathcal{T}(\{\boldsymbol{p}_{\cdot|s}^*\}, \boldsymbol{v}^*) \\
&\le \boldsymbol{\alpha_1}(\boldsymbol{v}^*) \cdot \boldsymbol{\alpha_2}(\boldsymbol{v}^*) \cdot \boldsymbol{\alpha_3}(\boldsymbol{v}^*)
\end{aligned} \tag{25}$$

where the first inequality follows because $\boldsymbol{v}^*$ is in the span of $\Phi$ and second inequality follows from Equation (23). The bounds for $\boldsymbol{\alpha_1}(\boldsymbol{v}^*)$ and $\boldsymbol{\alpha_2}(\boldsymbol{v}^*)$ are the same as arbitrary language models. The main difference is the bound on $\boldsymbol{\alpha_3}(\boldsymbol{v}^*)$ which will be a stronger bound for softmax models.

- **Error covariance → cross-entropy loss (softmax language models):** For softmax language models, we need to prove a modified version of Pinsker's inequality specifically for softmax models. This version will show a bound that only works when $\boldsymbol{v}^*$ is in the span of $\Phi$ and if the evaluated model $\boldsymbol{p}_{f(s)}$ computes softmax using $\Phi$ as well.

$$\text{Lemma E.4} \implies \boldsymbol{\alpha_3}(\boldsymbol{v}) \le \sqrt{2\|\boldsymbol{v}\|_\infty^2 (\ell_{\text{xent}}(\{\boldsymbol{p}_{f(s)}\}) - \inf_{f^*}(\{\boldsymbol{p}_{f^*(s)}\}))} \ \forall \boldsymbol{v} = \Phi^\top \lambda \in \mathbb{R}^V$$

Thus we suffer the suboptimality of the language model $\{\boldsymbol{p}_{f(s)}\}$ w.r.t. the best softmax model $\{\boldsymbol{p}_{f^*(s)}\}$ rather than the absolute best language model $\{\boldsymbol{p}_{\cdot|s}^*\}$. This is done using the softmax variant of Pinsker's inequality in Lemma E.4. We now present the detailed proofs for all results.

### E.2 Proofs for arbitrary language models

**Theorem B.1** (Strengthened Theorem 4.1). *Let $\{\boldsymbol{p}_{\cdot|s}\}$ be a language model that is $\epsilon$-optimal, i.e. $\ell_{\text{xent}}(\{\boldsymbol{p}_{\cdot|s}\}) - \ell_{\text{xent}}^* \le \epsilon$ for some $\epsilon > 0$. For a classification task $\mathcal{T}$ that is $(\tau, B)$-natural, we have*

$$\ell_\mathcal{T}(\{\boldsymbol{p}_{\cdot|s}\}) \le \tau + \sqrt{\frac{2B^2\epsilon}{\gamma(p_\mathcal{T}; \{\boldsymbol{p}_{\cdot|s}\})}}$$

*For a classification task $\mathcal{T}$ that is $(\tau, B)$-natural w.r.t. $\Phi$, we have*

$$\ell_\mathcal{T}(\{\boldsymbol{p}_{\cdot|s}\}) \le \ell_\mathcal{T}(\{\Phi \boldsymbol{p}_{\cdot|s}\}) \le \tau + \sqrt{\frac{2B^2\epsilon}{\gamma_\Phi(p_\mathcal{T}; \{\boldsymbol{p}_{\cdot|s}\})}}$$

*Proof.* The proof has two main steps that we summarize by the following two lemmas. The first one upper bounds the downstream performance on natural tasks with the covariance of errors.

**Lemma E.2.** *For a language model* $\{\boldsymbol{p}_{\cdot|s}\}$*, if* $\mathcal{T}$ *is* $(\tau, B)$*-natural,*

$$\ell_{\mathcal{T}}(\{\boldsymbol{p}_{\cdot|s}\}) \leq \tau + \sup_{\boldsymbol{v} \in \mathbb{R}^V, \|\boldsymbol{v}\|_\infty \leq B} \sqrt{\frac{\boldsymbol{v}^\top \Sigma_{p_L}(\Delta_{\{\boldsymbol{p}_{\cdot|s}\}})\boldsymbol{v}}{\gamma(p_{\mathcal{T}}; \{\boldsymbol{p}_{\cdot|s}\})}}$$

*If* $\mathcal{T}$ *is* $(\tau, B)$*-natural w.r.t.* $\Phi \in \mathbb{R}^{d \times V}$*,*

$$\ell_{\mathcal{T}}(\{\Phi\boldsymbol{p}_{\cdot|s}\}) \leq \tau + \sup_{\substack{\boldsymbol{v} = \Phi^\top \lambda \in \mathbb{R}^V, \\ \|\boldsymbol{v}\|_\infty \leq B}} \sqrt{\frac{\boldsymbol{v}^\top \Sigma_{p_L}(\Delta_{\{\boldsymbol{p}_{\cdot|s}\}})\boldsymbol{v}}{\gamma_\Phi(p_{\mathcal{T}}; \{\boldsymbol{p}_{\cdot|s}\})}}$$

*where* $\gamma(\cdot)$ *and* $\gamma_\Phi(\cdot)$ *are from Definition B.1.*

The second lemma upper bounds the covariance of error with the suboptimality of the language model.

**Lemma E.6.** *For a language model* $\{\boldsymbol{p}_{\cdot|s}\}$ *and classifier* $\boldsymbol{v} \in \mathbb{R}^V$*,*

$$\boldsymbol{v}^\top \Sigma_{p_L}(\Delta_{\{\boldsymbol{p}_{\cdot|s}\}})\boldsymbol{v} \leq 2\|\boldsymbol{v}\|_\infty^2 \left(\ell_{xent}(\{\boldsymbol{p}_{\cdot|s}\}) - \ell_{xent}^*\right)$$

*where* $\Sigma_{p_L}(\Delta_{\{\boldsymbol{p}_{\cdot|s}\}}) = \mathbb{E}_{s \sim p_L}\left[(\boldsymbol{p}_{\cdot|s} - \boldsymbol{p}_{\cdot|s}^*)(\boldsymbol{p}_{\cdot|s} - \boldsymbol{p}_{\cdot|s}^*)^\top\right]$ *as defined in Section B.*

We prove both the above lemmas in Section E.6. We first use these to prove the main result.

Combining the two lemmas, we get the following inequality

$$\begin{aligned}
\ell_{\mathcal{T}}(\{\boldsymbol{p}_{\cdot|s}\}) &\leq^{(a)} \tau + \sup_{\boldsymbol{v} \in \mathbb{R}^V, \|\boldsymbol{v}\|_\infty \leq B} \sqrt{\frac{\boldsymbol{v}^\top \Sigma_{p_L}(\Delta_{\{\boldsymbol{p}_{\cdot|s}\}})\boldsymbol{v}}{\gamma(p_{\mathcal{T}}; \{\boldsymbol{p}_{\cdot|s}\})}} \\
&\leq^{(b)} \tau + \sup_{\boldsymbol{v} \in \mathbb{R}^V, \|\boldsymbol{v}\|_\infty \leq B} \sqrt{\frac{2\|\boldsymbol{v}\|_\infty^2 \left(\ell_{\text{xent}}(\{\boldsymbol{p}_{\cdot|s}\}) - \ell_{\text{xent}}^*\right)}{\gamma(p_{\mathcal{T}}; \{\boldsymbol{p}_{\cdot|s}\})}} \\
&\leq^{(c)} \tau + \sqrt{\frac{2B^2\epsilon}{\gamma(p_{\mathcal{T}}; \{\boldsymbol{p}_{\cdot|s}\})}}
\end{aligned}$$

where $(a)$ uses first part of Lemma E.2, $(b)$ uses Lemma E.6 and $(c)$ uses the $\epsilon$-optimality of $\{\boldsymbol{p}_{\cdot|s}\}$. This proves the first part of the result. The second part can also be proved similarly.

$$\begin{aligned}
\ell_{\mathcal{T}}(\{\Phi\boldsymbol{p}_{\cdot|s}\}) &\leq^{(a)} \tau + \sup_{\substack{\boldsymbol{v} = \Phi^\top \lambda \in \mathbb{R}^V, \\ \|\boldsymbol{v}\|_\infty \leq B}} \sqrt{\frac{\boldsymbol{v}^\top \Sigma_{p_L}(\Delta_{\{\boldsymbol{p}_{\cdot|s}\}})\boldsymbol{v}}{\gamma_\Phi(p_{\mathcal{T}}; \{\boldsymbol{p}_{\cdot|s}\})}} \\
&\leq^{(b)} \tau + \sup_{\substack{\boldsymbol{v} = \Phi^\top \lambda \in \mathbb{R}^V, \\ \|\boldsymbol{v}\|_\infty \leq B}} \sqrt{\frac{2\|\boldsymbol{v}\|_\infty^2 \left(\ell_{\text{xent}}(\{\boldsymbol{p}_{\cdot|s}\}) - \ell_{\text{xent}}^*\right)}{\gamma_\Phi(p_{\mathcal{T}}; \{\boldsymbol{p}_{\cdot|s}\})}} \\
&\leq \tau + \sup_{\boldsymbol{v} \in \mathbb{R}^V, \|\boldsymbol{v}\|_\infty \leq B} \sqrt{\frac{2\|\boldsymbol{v}\|_\infty^2 \left(\ell_{\text{xent}}(\{\boldsymbol{p}_{\cdot|s}\}) - \ell_{\text{xent}}^*\right)}{\gamma_\Phi(p_{\mathcal{T}}; \{\boldsymbol{p}_{\cdot|s}\})}} \leq^{(c)} \tau + \sqrt{\frac{2B^2\epsilon}{\gamma_\Phi(p_{\mathcal{T}}; \{\boldsymbol{p}_{\cdot|s}\})}}
\end{aligned}$$

where $(a)$ uses second part of Lemma E.2, $(b)$ uses Lemma E.6 and $(c)$ uses the $\epsilon$-optimality of $\{\boldsymbol{p}_{\cdot|s}\}$. The proof of the lemmas can be found in Section E.6. □

**Theorem 4.1.** *Let* $\{\boldsymbol{p}_{\cdot|s}\}$ *be a language model that is* $\epsilon$*-optimal, i.e.* $\ell_{xent}(\{\boldsymbol{p}_{\cdot|s}\}) - \ell_{xent}^* \leq \epsilon$*, for some* $\epsilon > 0$*. For a classification task* $\mathcal{T}$ *that is* $(\tau, B)$*-natural, we have*

$$\ell_{\mathcal{T}}(\{\boldsymbol{p}_{\cdot|s}\}) \leq \tau + \sqrt{\frac{2B^2\epsilon}{\gamma(p_{\mathcal{T}})}}$$

*Proof.* This follows from the first part of Theorem B.1 if we can also show that $\gamma(p_\mathcal{T}; \{\boldsymbol{p}_{\cdot|s}\})^{-1} \leq \gamma(p_\mathcal{T})^{-1}$. For that we use the following lemma that we prove in Section E.6.

**Lemma E.9.** *For any $g : \mathcal{S} \to \mathbb{R}^D$ and $p_\mathcal{T} \in \Delta_\mathcal{S}$, we have $\|\Sigma_{p_L}(g)^{-\frac{1}{2}} \Sigma_{p_\mathcal{T}}(g) \Sigma_{p_L}(g)^{-\frac{1}{2}}\|_2 \leq \gamma(p_\mathcal{T})^{-1}$*

Instantiating this for $g = \Delta_{\{\boldsymbol{p}_{\cdot|s}\}}$ and using Equation (7), we get $\gamma(p_\mathcal{T}; \{\boldsymbol{p}_{\cdot|s}\})^{-1} \leq \gamma(p_\mathcal{T})^{-1}$, which completes the proof. $\square$

### E.3 PROOFS FOR SOFTMAX LANGUAGE MODELS

**Theorem 5.1** (Strengthened Theorem 4.2)**.** *For a fixed $\Phi$, let $f$ be features from an $\epsilon$-optimal $d$-dimensional softmax language model, i.e. $\ell_{xent}(f, \Phi) - \ell^*_{xent}(\Phi) \leq \epsilon$, where $\ell^*_{xent}(\Phi)$ is defined in Equation (4). For a classification task $\mathcal{T}$ that is $(\tau, B)$-natural w.r.t. $\Phi$, we have*

$$\ell_\mathcal{T}\left(\{\boldsymbol{p}_{f(s)}\}\right) \leq \ell_\mathcal{T}(\Phi p_f) \leq \tau + \sqrt{\frac{2B^2\epsilon}{\gamma(p_\mathcal{T}; \Phi p_f)}}$$

*Proof.* Instantiating Lemma E.2 for $\boldsymbol{p}_{\cdot|s} = \boldsymbol{p}_{f(s)}$, we get

$$\ell_\mathcal{T}(\{\Phi\boldsymbol{p}_{f(s)}\}) \leq \tau + \sup_{\substack{\boldsymbol{v}=\Phi^\top\lambda\in\mathbb{R}^V, \\ \|\boldsymbol{v}\|_\infty\leq B}} \sqrt{\frac{\boldsymbol{v}^\top \Sigma_{p_L}(\Delta_{\{\boldsymbol{p}_{f(s)}\}})\boldsymbol{v}}{\gamma_\Phi(p_\mathcal{T}; \{\boldsymbol{p}_{f(s)}\})}}$$

$$=^{(a)} \tau + \sqrt{\frac{\displaystyle\sup_{\|\Phi^\top\lambda\|_\infty\leq B} \lambda^\top \Phi\Sigma_{p_L}(\Delta_{\{\boldsymbol{p}_{f(s)}\}})\Phi^\top\lambda}{\gamma(p_\mathcal{T}; \Phi p_f)}}$$

$$= \tau + \sqrt{\frac{\displaystyle\sup_{\|\Phi^\top\lambda\|_\infty\leq B} \lambda^\top \Sigma_{p_L}(\Phi\Delta_{\{\boldsymbol{p}_{f(s)}\}})\lambda}{\gamma(p_\mathcal{T}; \Phi p_f)}}$$

where $(a)$ follows from Equation (9) that says $\gamma(p_\mathcal{T}; \Phi p_f) = \gamma_\Phi(p_\mathcal{T}; \{\boldsymbol{p}_{f(s)}\})$. We now prove a similar result for the second term in the following lemma that we prove in Section E.6.

**Lemma E.7.** *For a fixed $\Phi$ and a softmax language model with features $f$ and $\lambda \in \mathbb{R}^d$,*

$$\lambda^\top \Sigma_{p_L}(\Phi\Delta_{\{\boldsymbol{p}_{f(s)}\}})\lambda \leq 2\|\Phi^\top\lambda\|_\infty^2 \left(\ell_{xent}(f, \Phi) - \ell^*_{xent}(\Phi)\right)$$

*where $\Sigma_{p_L}(\Phi\Delta_{\{\boldsymbol{p}_{f(s)}\}}) = \mathbb{E}_{s\sim p_L}\left[(\Phi\boldsymbol{p}_{f(s)} - \Phi\boldsymbol{p}^*_{\cdot|s})(\Phi\boldsymbol{p}_{f(s)} - \Phi\boldsymbol{p}^*_{\cdot|s})^\top\right]$ as defined in Section B.*

Using Lemma E.7 directly gives us $\ell_\mathcal{T}(\Phi p_f) = \ell_\mathcal{T}(\{\Phi\boldsymbol{p}_{f(s)}\}) \leq \tau + \sqrt{\frac{B^2(\ell_{xent}(f,\Phi)-\ell^*_{xent}(\Phi))}{\gamma_\Phi(p_\mathcal{T};\Phi p_f)}}$, and the $\epsilon$-optimality almost completes the proof. The only thing remaining to show is that $\ell_\mathcal{T}(\{\boldsymbol{p}_{f(s)}\}) \leq \ell_\mathcal{T}(\Phi p_f)$ which follows from the following sequence.

$$\ell_\mathcal{T}(\{\boldsymbol{p}_{f(s)}\}) = \inf_{\boldsymbol{v}\in\mathbb{R}^V, b\in\mathbb{R}} \ell_\mathcal{T}(\{\boldsymbol{p}_{f(s)}\}, \boldsymbol{v}) \leq \inf_{\Phi^\top\lambda\in\mathbb{R}^V, b\in\mathbb{R}} \ell_\mathcal{T}(\{\boldsymbol{p}_{f(s)}\}, (\Phi^\top\lambda, b))$$

$$= \inf_{\lambda\in\mathbb{R}^d, b\in\mathbb{R}} \ell_\mathcal{T}(\{\Phi\boldsymbol{p}_{f(s)}\}, (\lambda, b)) = \ell_\mathcal{T}(\Phi p_f)$$

$\square$

**Theorem 4.2.** *For a fixed $\Phi$, let $f$ be features from an $\epsilon$-optimal $d$-dimensional softmax language model, i.e. $\ell_{xent}(f, \Phi) - \ell^*_{xent}(\Phi) \leq \epsilon$, where $\ell^*_{xent}(\Phi)$ is defined in Equation (4). For a classification task $\mathcal{T}$ that is $(\tau, B)$-natural w.r.t. $\Phi$, we have*

$$\ell_\mathcal{T}\left(\{\boldsymbol{p}_{f(s)}\}\right) \leq \ell_\mathcal{T}(\Phi p_f) \leq \tau + \sqrt{\frac{2B^2\epsilon}{\gamma(p_\mathcal{T})}}$$

*Proof.* This result follows directly from Theorem 5.1, if we can also show that $\gamma(p_{\mathcal{T}}; \Phi p_f)^{-1} \leq \gamma(p_{\mathcal{T}})^{-1}$ just like in the proof of Theorem 4.1. For that we again use Lemma E.9 with $g = \Phi\Delta_{\{p_{f(s)}\}}$ and Equation (9) and this completes the proof. □

### E.4 PROOFS FOR SECTION 4.3

We first show why Assumption 4.1 is approximately true when word embeddings are gaussian like.

**Lemma E.1.** *Suppose word embeddings $\phi_w$ are independent samples from the distribution $\mathcal{N}(\mu, \Sigma)$. Then for any $\theta \in \mathbb{R}^d$ such that $\lambda^2 = \theta^\top \Sigma \theta = O(1)$ we have that $|\log(Z_\theta) - \frac{1}{2}\theta^\top \Sigma \theta - \theta^\top \mu - \log(V)| \leq \epsilon$ with probability $1 - \delta$ for $\epsilon = \tilde{O}\left(\frac{e^{\lambda^2}}{\sqrt{V}}\right)$ and $\delta = 1 - \exp(-\Omega(\log^2(V)))$.*

*Proof.* We first note that $\log(Z_\theta) = \log\left(\sum_w e^{\theta^\top \phi_w}\right) = \theta^\top \mu + \log\left(\sum_w e^{\theta^\top(\phi_w - \mu)}\right)$, thus we can simply deal with the case where $\phi_w$ are sampled from $\mathcal{N}(0, \Sigma)$. Furthermore the only random variable of interest is $X_w = \theta^\top \phi_w$ which is a gaussian variable $\mathcal{N}(0, \theta^\top \Sigma \theta) = \mathcal{N}(0, \lambda^2)$. Thus the problem reduces to showing that for $V$ samples of $X_w \sim \mathcal{N}(0, \lambda^2)$, $\log(Z)$ is concentrated around $\lambda^2 + \log(V)$ where $Z = \sum_w \exp(X_w)$. This can be proved similarly to the proof of Lemma 2.1 in Arora et al. (2016). It is easy to see that $\mathbb{E}_{X_w \sim \mathcal{N}(0, \lambda^2)}[\exp(X_w)] = e^{\lambda^2}$. However the variable $\exp(X_w)$ is neither sub-gaussian nor sub-exponential and thus standard inequalities cannot be used directly. We use the same technique as Arora et al. (2016) to first observe that $\mathbb{E}[Z] = Ve^{\frac{1}{2}\lambda^2}$ and $\text{Var}[Z] \leq \mathbb{E}[\exp(2X_w)] = Ve^{2\lambda^2}$. After conditioning on the event that $X_w \leq \frac{1}{2}\lambda\log(V)$ and applying Berstein's inequality just like in Arora et al. (2016) completes the proof. □

We next prove Lemma 4.3 that establishes a linear relationship between $\Phi p_f$ and $f$ (under Assumption 4.1) and also the guarantees for $f$ on natural tasks.

**Lemma 4.3.** *Under Assumption 4.1, any feature map $f : \mathcal{S} \to \mathbb{R}^d$ satisfies $\Phi p_f(s) = \mathbf{A}f(s) + \mathbf{b}$, for all $s \in \mathcal{S}$.*

*Proof.* Assumption 4.1 gives us that $\log(Z_\theta) = \frac{1}{2}\theta^\top \mathbf{A}\theta + \theta^\top \mathbf{b} + c$. We prove this lemma by matching the gradients of $\log(Z_\theta)$ and the quadratic function on the R.H.S.

$$\nabla_\theta \log(Z_\theta) = \frac{\nabla_\theta Z_\theta}{Z_\theta} = \frac{\sum_{w \in \mathcal{W}} e^{\phi_w^\top \theta}\phi_w}{Z_\theta} = \sum_{w \in \mathcal{W}} p_\theta(w)\phi_w = \Phi p_\theta$$

Whereas the gradient of the quadratic part is $\nabla_\theta[\frac{1}{2}\theta^\top \mathbf{A}\theta + \theta^\top \mathbf{b} + c] = \mathbf{A}\theta + \mathbf{b}$. Matching the two for $\theta = f(s)$ gives us $\Phi p_f(s) = \Phi p_{f(s)} = \mathbf{A}f(s) + \mathbf{b}$. □

**Corollary 4.1.** *Using Lemma 4.3, for any $\epsilon$-optimal $f$, as defined in Theorem 4.2, for classification tasks that are $(\tau, B)$-natural w.r.t. $\Phi$ we have $\ell_{\mathcal{T}}(f) \leq \tau + \mathcal{O}(\sqrt{\epsilon})$.*

*Proof.* The main idea is that Lemma 4.3 gives us that $\Phi p_f(s) = \mathbf{A}f(s) + \mathbf{b}$ and thus any linear function of $\Phi p_f$ will also be a linear function of $f(s)$. From Theorem 5.1 (or Theorem 4.2), we also know that $\Phi p_f$ will do well on $\mathcal{T}$, i.e. $\ell_{\mathcal{T}}(\Phi p_f) \leq \tau + \mathcal{O}(B\sqrt{\epsilon})$. We formalize[7] the intuition as

$$\ell_{\mathcal{T}}(\Phi p_f) = \inf_{\lambda \in \mathbb{R}^d, b} \ell_{\mathcal{T}}(\Phi p_f, (\lambda, b)) = \inf_{\lambda \in \mathbb{R}^d, b} \ell_{\mathcal{T}}(\mathbf{A}f + \mathbf{b}, (\lambda, b)) = \inf_{\lambda \in \mathbb{R}^d, b} \ell_{\mathcal{T}}(f, (\mathbf{A}^\top \lambda, b + \lambda^\top \mathbf{b}))$$

$$\geq \inf_{\mathbf{v} \in \mathbb{R}^d, b'} \ell_{\mathcal{T}}(f, (\mathbf{v}, b')) = \ell_{\mathcal{T}}(f)$$

This shows that $\ell_{\mathcal{T}}(f) \leq \ell_{\mathcal{T}}(\Phi p_f) \leq \tau + \mathcal{O}(B\sqrt{\epsilon})$ and completes the proof. □

---

[7]Note that here we assume that we learn both a linear classifier and an intercept for a downstream classification task. All results in the paper essentially remain the same with an intercept in the definition of classification loss.

### E.5 PROOFS FOR SECTION C

**Theorem C.1.** *The optimal solution $f^*, \Phi^* = \arg\min_{f,\Phi} \ell_{quad}(f, \Phi)$ satisfies*

$$\Phi^* = \boldsymbol{B}\boldsymbol{U}_d^\top, \text{ for full rank } \boldsymbol{B} \in \mathbb{R}^{d \times d}$$
$$f^*(s) = (\Phi^*\Phi^{*\top})^{-1/2}\Phi^*\boldsymbol{p}^*_{\cdot|s} = \boldsymbol{C}\boldsymbol{U}_d^\top\boldsymbol{p}^*_{\cdot|s}, \text{ for full rank } \boldsymbol{C} \in \mathbb{R}^{d \times d}$$

*If $\Phi$ is fixed, then the optimal solution is $f^*(s) = (\Phi\Phi^\top)^{-1/2}\Phi\boldsymbol{p}^*_{\cdot|s}$.*

*Proof.* From Equations (10) and (11) we know that, $\ell_{quad,s}(\theta, \Phi) = -\theta^\top\Phi\boldsymbol{p}^*_{\cdot|s} + \frac{1}{2}\|\Phi^\top\theta\|^2$ and $\ell_{quad}(f, \Phi) = \mathop{\mathbb{E}}\limits_{s \sim p_L}[\ell_{quad,s}(f(s), \Phi)]$. For a fixed $\Phi$, we define $f_\Phi^*(s) = \arg\min_{\theta \in \mathbb{R}^d} \ell_{quad,s}(\theta, \Phi)$.

We use the first-order optimality condition to get $f_\Phi^*(s)$, by using the fact that $\nabla_\theta\ell_{quad,s}(\theta, \Phi) = -\Phi\boldsymbol{p}^*_{\cdot|s} + \Phi\Phi^\top\theta$. Setting the gradient to zero, we get $f_\Phi^*(s) = (\Phi\Phi^\top)^{-1}\Phi\boldsymbol{p}^*_{\cdot|s}$ [8]. To get the optimal $\Phi^*$ for this objective, we plug in this expression for $f_\Phi^*$ in $\ell_{quad}$ and find $\Phi^* = \arg\min_\Phi \ell_{quad}(f_\Phi^*, \Phi)$.

$$
\begin{aligned}
\ell_{quad}(f_\Phi^*, \Phi) &= \mathop{\mathbb{E}}\limits_{s \sim p^*}[\ell_{quad,s}(f_\Phi^*(s), \Phi)] = \mathop{\mathbb{E}}\limits_{s \sim p^*}\left[-f_\Phi^*(s)^\top\Phi\boldsymbol{p}^*_{\cdot|s} + \frac{1}{2}\|\Phi^\top f_\Phi^*(s)\|^2\right] \\
&= \mathop{\mathbb{E}}\limits_{s \sim p^*}\left[-((\Phi\Phi^\top)^{-1}\Phi\boldsymbol{p}^*_{\cdot|s})^\top\Phi\boldsymbol{p}^*_{\cdot|s} + \frac{1}{2}\|\Phi^\top(\Phi\Phi^\top)^{-1}\Phi\boldsymbol{p}^*_{\cdot|s}\|^2\right] \\
&= \mathop{\mathbb{E}}\limits_{s \sim p^*}\left[-\boldsymbol{p}^*_{\cdot|s}{}^\top\Phi^\top(\Phi\Phi^\top)^{-1}\Phi\boldsymbol{p}^*_{\cdot|s} + \frac{1}{2}\boldsymbol{p}^*_{\cdot|s}{}^\top\Phi^\top(\Phi\Phi^\top)^{-1}\Phi\Phi^\top(\Phi\Phi^\top)^{-1}\Phi\boldsymbol{p}^*_{\cdot|s}\right] \\
&= \mathop{\mathbb{E}}\limits_{s \sim p^*}\left[-\frac{1}{2}\boldsymbol{p}^*_{\cdot|s}{}^\top\Phi^\top(\Phi\Phi^\top)^{-1}\Phi\boldsymbol{p}^*_{\cdot|s}\right] = -\frac{1}{2}\mathop{\mathbb{E}}\limits_{s \sim p^*}\left[\mathrm{tr}\left(\boldsymbol{p}^*_{\cdot|s}{}^\top\Phi^\top(\Phi\Phi^\top)^{-1}\Phi\boldsymbol{p}^*_{\cdot|s}\right)\right] \\
&= -\frac{1}{2}\mathrm{tr}\left(\Phi^\top(\Phi\Phi^\top)^{-1}\Phi\mathop{\mathbb{E}}\limits_{s \sim p^*}\left[\boldsymbol{p}^*_{\cdot|s}\boldsymbol{p}^*_{\cdot|s}{}^\top\right]\right) \\
&= -\frac{1}{2}\left\langle\Phi^\top(\Phi\Phi^\top)^{-1}\Phi, \mathop{\mathbb{E}}\limits_{s \sim p^*}\left[\boldsymbol{p}^*_{\cdot|s}\boldsymbol{p}^*_{\cdot|s}{}^\top\right]\right\rangle = -\frac{1}{2}\left\langle\Phi^\top(\Phi\Phi^\top)^{-1}\Phi, \Omega^*\right\rangle
\end{aligned}
$$

where $\Omega^*$ is the substitutability matrix defined in Definition 5.1. Let $\Phi = \boldsymbol{N}\boldsymbol{T}\boldsymbol{V}^\top$ be the SVD. Then the above objective reduces to $\ell_{quad}(f_\Phi^*, \Phi) = -\frac{1}{2}\left\langle\boldsymbol{V}\boldsymbol{V}^\top, \Omega^*\right\rangle$ And hence learning the optimal $\Phi^*$ reduces to learning an optimal $\boldsymbol{V}^*$ such that

$$\boldsymbol{V}^* = \mathop{\arg\min}\limits_{\boldsymbol{V} \in \mathbb{R}^{V \times d}, \boldsymbol{V}^\top\boldsymbol{V} = I_d} -\langle\boldsymbol{V}\boldsymbol{V}^\top, \Omega^*\rangle$$

We will now show that the best such matrix is the matrix of top $d$ eigenvectors of $\Omega^*$, i.e. $\boldsymbol{V}^* = \boldsymbol{U}_d$ (cf. Definition 5.1). Here we will assume that the eigenvalues of $\Omega^*$ are all distinct for simplicity of presentation. First we note that $\langle\boldsymbol{V}\boldsymbol{V}^\top, \Omega^*\rangle = \|\boldsymbol{V}\boldsymbol{V}^\top\Omega^{*\frac{1}{2}}\|_F^2$, where $\Omega^{*\frac{1}{2}} = \boldsymbol{U}\boldsymbol{S}^{\frac{1}{2}}\boldsymbol{U}^\top$, with $\boldsymbol{U}$, $\boldsymbol{U}_d$ and $\boldsymbol{S}$ define in Definition 5.1. This can be shown by the following sequence of steps

$$
\begin{aligned}
\langle\boldsymbol{V}\boldsymbol{V}^\top, \Omega^*\rangle &= \mathrm{tr}(\boldsymbol{V}\boldsymbol{V}^\top\Omega^*) = \mathrm{tr}(\boldsymbol{V}\boldsymbol{V}^\top\boldsymbol{V}\boldsymbol{V}^\top\Omega^*) = \mathrm{tr}(\boldsymbol{V}\boldsymbol{V}^\top\Omega^*\boldsymbol{V}\boldsymbol{V}^\top) \\
&= \mathrm{tr}(\boldsymbol{V}\boldsymbol{V}^\top\boldsymbol{U}\boldsymbol{S}\boldsymbol{U}^\top\boldsymbol{V}\boldsymbol{V}^\top) = \mathrm{tr}(\boldsymbol{V}\boldsymbol{V}^\top\boldsymbol{U}\boldsymbol{S}^{\frac{1}{2}}\boldsymbol{U}^\top\boldsymbol{U}\boldsymbol{S}^{\frac{1}{2}}\boldsymbol{U}^\top\boldsymbol{V}\boldsymbol{V}^\top) \\
&= \mathrm{tr}(\boldsymbol{V}\boldsymbol{V}^\top\Omega^{*\frac{1}{2}}\Omega^{*\frac{1}{2}}\boldsymbol{V}\boldsymbol{V}^\top) = \langle\boldsymbol{V}\boldsymbol{V}^\top\Omega^{*\frac{1}{2}}, \boldsymbol{V}\boldsymbol{V}^\top\Omega^{*\frac{1}{2}}\rangle \\
&= \|\boldsymbol{V}\boldsymbol{V}^\top\Omega^{*\frac{1}{2}}\|_F^2
\end{aligned}
$$

Furthermore, we notice that $\|\boldsymbol{V}\boldsymbol{V}^\top\Omega^{*\frac{1}{2}}\|_F^2 = \|\Omega^{*\frac{1}{2}}\|_F^2 - \|\Omega^{*\frac{1}{2}} - \boldsymbol{V}\boldsymbol{V}^\top\Omega^{*\frac{1}{2}}\|_F^2$ as shown below

$$\|\Omega^{*\frac{1}{2}} - \boldsymbol{V}\boldsymbol{V}^\top\Omega^{*\frac{1}{2}}\|_F^2 = \|\Omega^{*\frac{1}{2}}\|_F^2 + \|\boldsymbol{V}\boldsymbol{V}^\top\Omega^{*\frac{1}{2}}\|_F^2 - 2\mathrm{tr}(\Omega^{*\frac{1}{2}}\boldsymbol{V}\boldsymbol{V}^\top\Omega^{*\frac{1}{2}})$$

---

[8]It will be clear later that the optimal solution will have as high a rank as possible $\Phi$. All inverses can be replaced by pseudo-inverses for low-rank matrices.

$$\begin{aligned}
&= \|\Omega^{*\frac{1}{2}}\|_F^2 + \|\boldsymbol{V}\boldsymbol{V}^\top\Omega^{*\frac{1}{2}}\|_F^2 - 2\text{tr}(\Omega^{*\frac{1}{2}}\boldsymbol{V}\boldsymbol{V}^\top\boldsymbol{V}\boldsymbol{V}^\top\Omega^{*\frac{1}{2}}) \\
&= \|\Omega^{*\frac{1}{2}}\|_F^2 + \|\boldsymbol{V}\boldsymbol{V}^\top\Omega^{*\frac{1}{2}}\|_F^2 - 2\|\boldsymbol{V}\boldsymbol{V}^\top\Omega^{*\frac{1}{2}}\|_F^2 \\
&= \|\Omega^{*\frac{1}{2}}\|_F^2 - \|\boldsymbol{V}\boldsymbol{V}^\top\Omega^{*\frac{1}{2}}\|_F^2
\end{aligned}$$

Thus we get $\displaystyle\operatorname*{arg\,min}_{\boldsymbol{V}\in\mathbb{R}^{V\times d},\boldsymbol{V}^\top\boldsymbol{V}=I_d} -\langle\boldsymbol{V}\boldsymbol{V}^\top,\Omega^*\rangle = \operatorname*{arg\,min}_{\boldsymbol{V}\in\mathbb{R}^{V\times d},\boldsymbol{V}^\top\boldsymbol{V}=I_d} \|\Omega^{*\frac{1}{2}} - \boldsymbol{V}\boldsymbol{V}^\top\Omega^{*\frac{1}{2}}\|_F^2$.

Note that $\boldsymbol{V}\boldsymbol{V}^\top\Omega^{*\frac{1}{2}}$ has columns that are columns of $\Omega^{*\frac{1}{2}}$ projected on the space spanned by columns $\boldsymbol{V}$. It is folklore that the best such subspace $\boldsymbol{V}^*$ is the subspace spanned by the top $d$ eigenvectors of $\Omega^{*\frac{1}{2}}$, which is the same as top $d$ eigenvectors of $\Omega^*$, thus giving us $\boldsymbol{V}^*\boldsymbol{V}^{*\top} = \boldsymbol{U}_d\boldsymbol{U}_d^\top$. Thus we get $\boldsymbol{V}^* = \boldsymbol{U}_d\boldsymbol{M}$ for $\boldsymbol{M} = \boldsymbol{U}_d^\top\boldsymbol{V}^*$.

This tells us that the optimal solution $\Phi^*$ will have SVD of the form $\Phi^* = \boldsymbol{N}^*\boldsymbol{T}^*\boldsymbol{V}^{*\top}$, thus giving us $\Phi^* = \boldsymbol{B}\boldsymbol{U}_d^\top$ for matrix $\boldsymbol{B} = \boldsymbol{N}^*\boldsymbol{T}^*\boldsymbol{M}^\top \in \mathbb{R}^{d\times d}$. This directly gives $f^* = f_{\Phi^*}^* = (\Phi^*\Phi^{*\top})^{-1}\Phi^*\boldsymbol{p}_{\cdot|s}^* = \boldsymbol{N}^*\boldsymbol{T}^{-1}\boldsymbol{V}^{*\top}\boldsymbol{p}_{\cdot|s}^* = \boldsymbol{C}\boldsymbol{U}_d^\top\boldsymbol{p}_{\cdot|s}^*$ for $\boldsymbol{C} = \boldsymbol{N}^*\boldsymbol{T}^{*-1}\boldsymbol{M}^\top$.

$\square$

### E.6 Proofs for supporting lemmas

**Lemma E.2.** *For a language model $\{\boldsymbol{p}_{\cdot|s}\}$, if $\mathcal{T}$ is $(\tau, B)$-natural,*

$$\ell_{\mathcal{T}}(\{\boldsymbol{p}_{\cdot|s}\}) \leq \tau + \sup_{\boldsymbol{v}\in\mathbb{R}^V,\|\boldsymbol{v}\|_\infty\leq B}\sqrt{\frac{\boldsymbol{v}^\top\Sigma_{p_L}(\Delta_{\{\boldsymbol{p}_{\cdot|s}\}})\boldsymbol{v}}{\gamma(p_{\mathcal{T}};\{\boldsymbol{p}_{\cdot|s}\})}}$$

*If $\mathcal{T}$ is $(\tau, B)$-natural w.r.t. $\Phi \in \mathbb{R}^{d\times V}$,*

$$\ell_{\mathcal{T}}(\{\Phi\boldsymbol{p}_{\cdot|s}\}) \leq \tau + \sup_{\substack{\boldsymbol{v}=\Phi^\top\lambda\in\mathbb{R}^V, \\ \|\boldsymbol{v}\|_\infty\leq B}}\sqrt{\frac{\boldsymbol{v}^\top\Sigma_{p_L}(\Delta_{\{\boldsymbol{p}_{\cdot|s}\}})\boldsymbol{v}}{\gamma_\Phi(p_{\mathcal{T}};\{\boldsymbol{p}_{\cdot|s}\})}}$$

*where $\gamma(\cdot)$ and $\gamma_\Phi(\cdot)$ are from Definition B.1.*

*Proof.* We note the following upper bounds on $\ell_{\mathcal{T}}(\{\boldsymbol{p}_{\cdot|s}\})$ and $\ell_{\mathcal{T}}(\{\Phi\boldsymbol{p}_{\cdot|s}\})$.

$$\ell_{\mathcal{T}}(\{\boldsymbol{p}_{\cdot|s}\}) = \inf_{\boldsymbol{v}\in\mathbb{R}^V}\left\{\ell_{\mathcal{T}}(\{\boldsymbol{p}_{\cdot|s}\},\boldsymbol{v})\right\} \leq \inf_{\substack{\boldsymbol{v}\in\mathbb{R}^V, \\ \|\boldsymbol{v}\|_\infty\leq B}}\left\{\ell_{\mathcal{T}}(\{\boldsymbol{p}_{\cdot|s}\},\boldsymbol{v})\right\} \tag{26}$$

$$\ell_{\mathcal{T}}(\{\Phi\boldsymbol{p}_{\cdot|s}\}) = \inf_{\boldsymbol{v}=\Phi^\top\lambda\in\mathbb{R}^V}\left\{\ell_{\mathcal{T}}(\{\boldsymbol{p}_{\cdot|s}\},\boldsymbol{v})\right\} \leq \inf_{\substack{\boldsymbol{v}=\Phi^\top\lambda\in\mathbb{R}^V,b\in\mathbb{R}, \\ \|\boldsymbol{v}\|_\infty\leq B}}\left\{\ell_{\mathcal{T}}(\{\boldsymbol{p}_{\cdot|s}\},\boldsymbol{v})\right\} \tag{27}$$

When $\mathcal{T}$ is $(\tau, B)$-natural, by Definition 3.1 we know that $\displaystyle\inf_{\substack{\boldsymbol{v}\in\mathbb{R}^V \\ \|\boldsymbol{v}\|_\infty\leq B}}\left[\ell_{\mathcal{T}}(\{\boldsymbol{p}_{\cdot|s}^*\},\boldsymbol{v})\right] \leq \tau$. We now upper bound $\ell_{\mathcal{T}}(\{\boldsymbol{p}_{\cdot|s}\},\boldsymbol{v})$ using Lemma E.8. Taking infimum w.r.t. $\boldsymbol{v}\in\mathbb{R}^V,\|\boldsymbol{v}\|_\infty\leq B$ from the inequality in Lemma E.8.

$$\ell_{\mathcal{T}}(\{\boldsymbol{p}_{\cdot|s}\},\boldsymbol{v}) \leq \ell_{\mathcal{T}}(\{\boldsymbol{p}_{\cdot|s}^*\},\boldsymbol{v}) + \sqrt{\boldsymbol{v}^\top\Sigma_{p_{\mathcal{T}}}(\Delta_{\{\boldsymbol{p}_{\cdot|s}\}})\boldsymbol{v}}$$

$$\inf_{\substack{\boldsymbol{v}\in\mathbb{R}^V \\ \|\boldsymbol{v}\|_\infty\leq B}}\ell_{\mathcal{T}}(\{\boldsymbol{p}_{\cdot|s}\},\boldsymbol{v}) \leq \inf_{\substack{\boldsymbol{v}\in\mathbb{R}^V \\ \|\boldsymbol{v}\|_\infty\leq B}}\ell_{\mathcal{T}}(\{\boldsymbol{p}_{\cdot|s}^*\},\boldsymbol{v}) + \sup_{\boldsymbol{v}\in\mathbb{R}^V,\|\boldsymbol{v}\|_\infty\leq B}\sqrt{\boldsymbol{v}^\top\Sigma_{p_{\mathcal{T}}}(\Delta_{\{\boldsymbol{p}_{\cdot|s}\}})\boldsymbol{v}}$$

This, combined with Equation (26), gives us

$$\ell_{\mathcal{T}}(\{\boldsymbol{p}_{\cdot|s}\}) \leq \tau + \sup_{\boldsymbol{v}\in\mathbb{R}^V,\|\boldsymbol{v}\|_\infty\leq B}\sqrt{\boldsymbol{v}^\top\Sigma_{p_{\mathcal{T}}}(\Delta_{\{\boldsymbol{p}_{\cdot|s}\}})\boldsymbol{v}} \tag{28}$$

Using Lemma E.10 and the definition of $\gamma(p_{\mathcal{T}}; \{\boldsymbol{p}_{\cdot|s}\})$ in Equation (7), we get that

$$\boldsymbol{v}^\top \Sigma_{p_{\mathcal{T}}}(\Delta_{\{\boldsymbol{p}_{\cdot|s}\}})\boldsymbol{v} \le \left\| \Sigma_{p_L}(\Delta_{\{\boldsymbol{p}_{\cdot|s}\}})^{-\frac{1}{2}} \Sigma_{p_{\mathcal{T}}}(\Delta_{\{\boldsymbol{p}_{\cdot|s}\}}) \Sigma_{p_L}(\Delta_{\{\boldsymbol{p}_{\cdot|s}\}})^{-\frac{1}{2}} \right\|_2 \left( \boldsymbol{v}^\top \Sigma_{p_L}(\Delta_{\{\boldsymbol{p}_{\cdot|s}\}})\boldsymbol{v} \right)$$
$$= \frac{\boldsymbol{v}^\top \Sigma_{p_L}(\Delta_{\{\boldsymbol{p}_{\cdot|s}\}})\boldsymbol{v}}{\gamma(p_{\mathcal{T}}; \{\boldsymbol{p}_{\cdot|s}\})} \tag{29}$$

We have thus successfully transferred the bound from the distribution $p_{\mathcal{T}}$ to $p_L$. Combining this with Equation (28) completes the proof of the first part of the lemma.

We now prove the second part of the lemma where we only assume that $\mathcal{T}$ is $(\tau, B)$-natural w.r.t. $\Phi$. Here we instead take the infimum over classifiers in the span of $\Phi$ in Lemma E.8 to get

$$\inf_{\substack{\boldsymbol{v}=\Phi^\top \lambda \in \mathbb{R}^V, b \in \mathbb{R}, \\ \|\boldsymbol{v}\|_\infty \le B}} \left\{ \ell_{\mathcal{T}}(\{\boldsymbol{p}_{\cdot|s}\}, \boldsymbol{v}) \right\} \le \inf_{\substack{\boldsymbol{v}=\Phi^\top \lambda \in \mathbb{R}^V, b \in \mathbb{R}, \\ \|\boldsymbol{v}\|_\infty \le B}} \left\{ \ell_{\mathcal{T}}(\{\boldsymbol{p}^*_{\cdot|s}\}, \boldsymbol{v}) \right\} +$$
$$\sup_{\substack{\boldsymbol{v}=\Phi^\top \lambda \in \mathbb{R}^V, \\ \|\boldsymbol{v}\|_\infty \le B}} \sqrt{\boldsymbol{v}^\top \Sigma_{p_{\mathcal{T}}}(\Delta_{\{\boldsymbol{p}_{\cdot|s}\}})\boldsymbol{v}} \tag{30}$$

This, combined with definition of $(\tau, B)$-natural task w.r.t. $\Phi$ and Equation (27) gives us

$$\ell_{\mathcal{T}}(\{\Phi\boldsymbol{p}_{\cdot|s}\}) \le \tau + \sup_{\substack{\boldsymbol{v}=\Phi^\top \lambda \in \mathbb{R}^V, \\ \|\boldsymbol{v}\|_\infty \le B}} \sqrt{\boldsymbol{v}^\top \Sigma_{p_{\mathcal{T}}}(\Delta_{\{\boldsymbol{p}_{\cdot|s}\}})\boldsymbol{v}} \tag{31}$$

For the last term, for any $\boldsymbol{v} = \Phi^\top \lambda, \lambda \in \mathbb{R}^d$ we notice that

$$\boldsymbol{v}^\top \Sigma_{p_{\mathcal{T}}}(\Delta_{\{\boldsymbol{p}_{\cdot|s}\}})\boldsymbol{v} = \lambda^\top \Phi \Sigma_{p_{\mathcal{T}}}(\Delta_{\{\boldsymbol{p}_{\cdot|s}\}})\Phi^\top \lambda = \lambda^\top \Sigma_{p_{\mathcal{T}}}(\Phi\Delta_{\{\boldsymbol{p}_{\cdot|s}\}})\lambda$$
$$\le^{(a)} \left\| \Sigma_{p_L}(\Phi\Delta_{\{\boldsymbol{p}_{\cdot|s}\}})^{-\frac{1}{2}} \Sigma_{p_{\mathcal{T}}}(\Phi\Delta_{\{\boldsymbol{p}_{\cdot|s}\}}) \Sigma_{p_L}(\Phi\Delta_{\{\boldsymbol{p}_{\cdot|s}\}})^{-\frac{1}{2}} \right\|_2 \left( \lambda^\top \Sigma_{p_L}(\Phi\Delta_{\{\boldsymbol{p}_{\cdot|s}\}})\lambda \right)$$
$$= \frac{\lambda^\top \Sigma_{p_L}(\Phi\Delta_{\{\boldsymbol{p}_{\cdot|s}\}})\lambda}{\gamma_\Phi(p_{\mathcal{T}}; \{\boldsymbol{p}_{\cdot|s}\})} = \frac{\boldsymbol{v}^\top \Sigma_{p_L}(\Delta_{\{\boldsymbol{p}_{\cdot|s}\}})\boldsymbol{v}}{\gamma_\Phi(p_{\mathcal{T}}; \{\boldsymbol{p}_{\cdot|s}\})}$$

This combined with Equation (31), we get

$$\ell_{\mathcal{T}}(\{\Phi\boldsymbol{p}_{\cdot|s}\}) \le \tau + \inf_{\substack{\boldsymbol{v}=\Phi^\top \lambda \in \mathbb{R}^V, \\ \|\boldsymbol{v}\|_\infty \le B}} \sqrt{\frac{\boldsymbol{v}^\top \Sigma_{p_L}(\Delta_{\{\boldsymbol{p}_{\cdot|s}\}})\boldsymbol{v}}{\gamma_\Phi(p_{\mathcal{T}}; \{\boldsymbol{p}_{\cdot|s}\})}}$$

$\square$

**Lemma E.3** (Pinsker's inequality). *For discrete distributions $q, q^* \in \Delta_V$, let $\boldsymbol{q}, \boldsymbol{q}^* \in \mathbb{R}^V$ be the corresponding vector of probabilities. Then we have*

$$\max_{\|\boldsymbol{v}\|_\infty \le 1} |\boldsymbol{v}^\top(\boldsymbol{q} - \boldsymbol{q}^*)| \le \sqrt{2D_{\mathrm{KL}}(q^*, q)}$$

*Proof.* This basically follows from Pinsker's inequality which upper bounds the total variation distance between distributions by their KL-divergence

$$\max_{\|\boldsymbol{v}\|_\infty \le 1} |\boldsymbol{v}^\top(\boldsymbol{q} - \boldsymbol{q}^*)| = \|\boldsymbol{q} - \boldsymbol{q}^*\|_1 = 2\,\mathrm{TV}(q^*, q) \le \sqrt{2D_{\mathrm{KL}}(q^*, q)}$$

$\square$

We remind the reader that for an embedding matrix $\Phi \in \mathbb{R}^{d \times V}$, $p_{\theta,\Phi} := \mathrm{softmax}(\Phi^\top \theta)$

**Lemma E.4** (Softmax variant of Pinsker's inequality). *Consider a matrix $\Phi \in \mathbb{R}^{d \times V}$ with $d \le V$. For any discrete distribution $q^* \in \Delta_V$ and softmax distribution $p_{\theta,\Phi} = \mathrm{softmax}(\Phi^\top \theta) \in \Delta_V$ for $\theta \in \mathbb{R}^d$, let $\boldsymbol{q}^*, \boldsymbol{p}_{\theta,\Phi} \in \mathbb{R}^V$ be the corresponding vector of probabilities. Then we have*

$$\max_{\substack{\boldsymbol{v}=\Phi^\top \lambda, \\ \|\boldsymbol{v}\|_\infty \le 1}} |\boldsymbol{v}^\top(\boldsymbol{p}_{\theta,\Phi} - \boldsymbol{q}^*)| \le \sqrt{2\left( D_{\mathrm{KL}}(p_{\theta,\Phi}, q^*) - \inf_{\theta^* \in \mathbb{R}^d} D_{\mathrm{KL}}(p_{\theta^*,\Phi}, q^*) \right)} \tag{32}$$

*Pinsker's inequality (Lemma E.3), on the other hand, gives*

$$\max_{\|\boldsymbol{v}\|_\infty \leq 1} |\boldsymbol{v}^\top (\boldsymbol{p}_{\theta,\Phi} - \boldsymbol{q}^*)| \leq \sqrt{2 D_{\mathrm{KL}}(p_{\theta,\Phi}, q^*)}$$

*Proof.* Define the loss $\rho(\theta) \coloneqq D_{\mathrm{KL}}(p_{\theta,\Phi}, q^*)$. The statement in Equation (32) to prove reduces to

$$\max_{\|\Phi^\top \lambda\|_\infty \leq 1} |\lambda^\top (\Phi \boldsymbol{p}_{\theta,\Phi} - \Phi \boldsymbol{q}^*)| \leq \sqrt{2 \left( \rho(\theta) - \inf_{\theta^* \in \mathbb{R}^d} \rho(\theta^*) \right)} \tag{33}$$

To prove this, we compute the gradient and hessian of $\rho(\theta)$ w.r.t. $\theta$. We can simplify $\rho(\theta)$ as follows

$$\rho(\theta) = D_{\mathrm{KL}}(p_{\theta,\Phi}, q^*) = \mathop{\mathbb{E}}_{w \sim q^*} [-\log(p_{\theta,\Phi}(w))] = \mathop{\mathbb{E}}_{w \sim q^*} \left[ -\log \left( \frac{e^{\theta^\top \phi_w}}{\sum_{w'} e^{\theta^\top \phi_{w'}}} \right) \right]$$

$$= -\theta^\top \Phi \boldsymbol{q}^* + \log \left( \sum_{w'} e^{\theta^\top \phi_{w'}} \right) = -\theta^\top \Phi \boldsymbol{q}^* + \log(Z_\theta)$$

The gradient is

$$\nabla \rho(\theta) = \nabla \left[ -\theta^\top \Phi \boldsymbol{q}^* + \log(Z_\theta) \right] = -\Phi \boldsymbol{q}^* + \frac{\nabla Z_\theta}{Z_\theta}$$

$$= -\Phi \boldsymbol{q}^* + \frac{\nabla \sum_w e^{\theta^\top \phi_w}}{Z_\theta} = -\Phi \boldsymbol{q}^* + \frac{\sum_w e^{\theta^\top \phi_w} \phi_w}{Z_\theta}$$

$$= -\Phi \boldsymbol{q}^* + \Phi p_{\theta,\Phi}$$

Similarly the Hessian can be computed

$$\nabla^2 \rho(\theta) = \nabla(\nabla \rho(\theta)) = \nabla[-\Phi \boldsymbol{q}^* + \Phi p_{\theta,\Phi}] = \nabla \sum_{w \in \mathcal{W}} p_{\theta,\Phi}(w) \phi_w = \sum_{w \in \mathcal{W}} \nabla \frac{e^{\theta^\top \phi_w}}{Z_\theta} \phi_w$$

$$= \sum_{w \in \mathcal{W}} \frac{e^{\theta^\top \phi_w}}{Z_\theta} \phi_w \phi_w^\top - \frac{e^{\theta^\top \phi_w}}{Z_\theta^2} \phi_w \left( \sum_{w'} e^{\theta^\top \phi_{w'}} \phi_{w'} \right)^\top$$

$$= \mathop{\mathbb{E}}_{w \sim p_{\theta,\Phi}} [\phi_w \phi_w^\top] - \left( \mathop{\mathbb{E}}_{w \sim p_{\theta,\Phi}} [\phi_w] \right) \left( \mathop{\mathbb{E}}_{w \sim p_{\theta,\Phi}} [\phi_w] \right)^\top = \mathrm{Cov}_{w \sim p_{\theta,\Phi}}[\phi_w]$$

Where $\mathrm{Cov}_{w \sim p_{\theta,\Phi}}[\phi_w]$ denotes the covariance of the word embeddings $\phi_w$ when measured w.r.t. the distribution $p_{\theta,\Phi}$. This directly gives us that $\nabla^2 \rho(\theta) \succeq 0$, since the covariance is always psd, and thus $\rho$ is convex in $\theta$.

We return to the statement in Equation (33) that we need to prove. With the expression for gradient of $\rho$ at hand, we can rewrite Equation (33) as trying to prove

$$|\lambda^\top \nabla \rho(\theta)| \leq \|\Phi^\top \lambda\|_\infty \sqrt{2 \left( \rho(\theta) - \inf_{\theta^* \in \mathbb{R}^d} \rho(\theta^*) \right)} \tag{34}$$

Furthermore, using the definition of the Hessian, it is not hard to see for some $\lambda, \tilde{\theta} \in \mathbb{R}^d$ that $\lambda^\top \nabla^2 \rho(\tilde{\theta}) \lambda = \mathrm{Cov}_{w \sim p_{\tilde{\theta},\Phi}}[\lambda^\top \phi_w] \leq \mathop{\mathbb{E}}_{w \sim p_{\tilde{\theta},\Phi}} [(\lambda^\top \phi_w)^2] \leq \|\Phi^\top \lambda\|_\infty^2$. Thus we can evoke

Lemma E.5 with $\ell = \rho$ and $L = \|\Phi^\top \lambda\|_\infty^2$ to prove Equation (34) and thus completing the proof. Intuitively Lemma E.5 exploits the smoothness of the function to argue that small suboptimality (i.e. being close to optimal solution in function value) is sufficient to guarantee small norm of the gradient, a property that is well-known in the optimization literature. We now present this lemma $\qquad \square$

**Lemma E.5.** *If a function $\ell : \mathbb{R}^d \to \mathbb{R}$ and $\lambda \in \mathbb{R}^d$ satisfy $\lambda^\top \nabla^2 \ell(\tilde{\theta}) \lambda \leq L, \forall \tilde{\theta} \in \mathbb{R}^d$ (L-smoothness in the direction of $\lambda$) and if $\ell^* = \inf_{\theta \in \mathbb{R}^d} \ell(\theta)$, then $|\lambda^\top \nabla \ell(\theta)|^2 \leq 2L(\ell(\theta) - \ell^*)$*

*Proof.* This is a variant of a classical result used in optimization and we prove it here for completeness. For any $\eta \in \mathbb{R}$ we have

$$
\begin{aligned}
\ell(\theta) - \ell^* &\geq^{(a)} \ell(\theta) - \ell(\theta - \eta\lambda) \\
&\geq^{(b)} \ell(\theta) - \left( \ell(\theta) + \langle \nabla\ell(\theta), -\eta\lambda \rangle + \frac{\eta^2}{2}\lambda^\top\nabla^2\ell(\tilde{\theta})\lambda \right) \\
&\geq^{(c)} \eta(\lambda^\top\nabla\ell(\theta)) - \frac{\eta^2 L}{2}
\end{aligned}
$$

where $(a)$ follows from the definition of infimum and $(b)$ follows from Taylor's expansion for some $\tilde{\theta} \in [\theta - \eta\lambda, \theta]$ and $(c)$ follows from the smoothness condition in the statement of the lemma. Picking $\eta = \frac{\lambda^\top\nabla\ell(\theta)}{L}$ gives us $\ell(\theta) - \ell^* \geq \frac{1}{2L}|\lambda^\top\nabla\ell(\theta)|^2$, thus completing the proof. $\qquad\square$

**Lemma E.6.** *For a language model $\{p_{\cdot|s}\}$ and classifier $v \in \mathbb{R}^V$,*

$$
v^\top\Sigma_{p_L}(\Delta_{\{p_{\cdot|s}\}})v \leq 2\|v\|_\infty^2 \left( \ell_{xent}(\{p_{\cdot|s}\}) - \ell_{xent}^* \right)
$$

*where $\Sigma_{p_L}(g) = \underset{s \sim p_L}{\mathbb{E}}[g(s)g(s)^\top]$ and $\Delta_{\{p_{\cdot|s}\}}(s) = p_{\cdot|s} - p_{\cdot|s}^*$ are defined in Section B*

*Proof.* We first note that

$$
\ell_{\text{xent}}(\{p_{\cdot|s}\}) - \ell_{\text{xent}}(\{p_{\cdot|s}^*\}) = \underset{s \sim p_L}{\mathbb{E}}\underset{w \sim p_{\cdot|s}^*}{\mathbb{E}}\left[ \log\left( \frac{p_{\cdot|s}^*(w)}{p_{\cdot|s}(w)} \right) \right] = \underset{s \sim p_L}{\mathbb{E}}\left[ D_{\text{KL}}(p_{\cdot|s}^*, p_{\cdot|s}) \right] \quad (35)
$$

We bound $v^\top\Sigma_{p_L}(\Delta_{\{p_{\cdot|s}\}})v$ below

$$
\begin{aligned}
v^\top\Sigma_{p_L}(\Delta_{\{p_{\cdot|s}\}})v &= \underset{s \sim p_L}{\mathbb{E}}\left[ \left( v^\top(p_{\cdot|s} - p_{\cdot|s}^*) \right)^2 \right] \\
&\leq^{(a)} \|v\|_\infty^2 \underset{s \sim p_L}{\mathbb{E}}\left[ 2D_{\text{KL}}(p_{\cdot|s}^*, p_{\cdot|s}) \right] \\
&=^{(b)} 2\|v\|_\infty^2 \left( \ell_{\text{xent}}(\{p_{\cdot|s}\}) - \ell_{\text{xent}}(\{p_{\cdot|s}^*\}) \right)
\end{aligned}
$$

where $(a)$ follows from Lemma E.3 (Pinsker's inequality), $(b)$ uses Equation (35). $\qquad\square$

**Lemma E.7.** *For a fixed $\Phi$, a softmax language model with features $f$ and $\lambda \in \mathbb{R}^d$,*

$$
\lambda^\top\Sigma_{p_L}(\Phi\Delta_{\{p_{f(s)}\}})\lambda \leq 2\|\Phi^\top\lambda\|_\infty^2 (\ell_{xent}(f, \Phi) - \ell_{xent}^*(\Phi))
$$

*where $\Sigma_{p_L}(\Phi\Delta_{\{p_{f(s)}\}}) = \underset{s \sim p_L}{\mathbb{E}}\left[ (\Phi p_{f(s)} - \Phi p_{\cdot|s}^*)(\Phi p_{f(s)} - \Phi p_{\cdot|s}^*)^\top \right]$ as defined in Section B.*

*Proof.* We start by nothing that

$$
\begin{aligned}
\lambda^\top\Sigma_{p_L}(\Phi\Delta_{\{p_{f(s)}\}})\lambda &= \lambda^\top\underset{s \sim p_L}{\mathbb{E}}\left[ (\Phi p_{f(s)} - \Phi p_{\cdot|s}^*)(\Phi p_{f(s)} - \Phi p_{\cdot|s}^*)^\top \right]\lambda \\
&= \underset{s \sim p_L}{\mathbb{E}}[|\lambda^\top(\Phi p_{f(s)} - \Phi p_{\cdot|s}^*)|^2] = \underset{s \sim p_L}{\mathbb{E}}[|(\Phi^\top\lambda)^\top(p_{f(s)} - p_{\cdot|s}^*)|^2]
\end{aligned}
$$

We will use the variant of Pinsker's inequality from Lemma E.4 to bound each term on the right hand side. Notice that $\ell_{\text{xent}}(f, \Phi) - \ell_{\text{xent}}^*(\Phi) = \underset{s \sim p_L}{\mathbb{E}}[\ell_{\text{xent},s}(f(s), \Phi) - \underset{\theta \in \mathbb{R}^d}{\inf}\ell_{\text{xent},s}(\theta, \Phi)]$.

$$
\begin{aligned}
\lambda^\top\Sigma_{p_L}(\Phi\Delta_{\{p_{f(s)}\}})\lambda &= \underset{s \sim p_L}{\mathbb{E}}[|(\Phi^\top\lambda)^\top(p_{f(s)} - p_{\cdot|s}^*)|^2] \\
&\leq^{(a)} 2\|\Phi^\top\lambda\|_\infty^2 \underset{s \sim p_L}{\mathbb{E}}\left[ D_{\text{KL}}(p_{\cdot|s}^*, p_{f(s),\Phi}) - \underset{\theta \in \mathbb{R}^d}{\inf} D_{\text{KL}}(p_{\cdot|s}^*, p_{\theta,\Phi}) \right] \\
&\leq 2\|\Phi^\top\lambda\|_\infty^2 \underset{s \sim p_L}{\mathbb{E}}\left[ \ell_{\text{xent},s}(f(s), \Phi) - \underset{\theta \in \mathbb{R}^d}{\inf}\ell_{\text{xent},s}(\theta, \Phi) \right] \\
&\leq 2\|\Phi^\top\lambda\|_\infty^2 (\ell_{\text{xent}}(f, \Phi) - \ell_{\text{xent}}^*(\Phi))
\end{aligned}
$$

where $(a)$ follows from Lemma E.4. This completes the proof. $\qquad\square$

### E.6.1 CLASSIFICATION LOSS TO COVARIANCE OF ERROR

**Lemma E.8.** *For any task $\mathcal{T}$ and classifier $\boldsymbol{v} \in \mathbb{R}^V$ and predicted probabilities $\{\boldsymbol{p}_{\cdot|s}\}$*

$$\ell_{\mathcal{T}}(\{\boldsymbol{p}_{\cdot|s}\}, \boldsymbol{v}) \leq \ell_{\mathcal{T}}(\{\boldsymbol{p}^*_{\cdot|s}\}, \boldsymbol{v}) + \sqrt{\mathop{\mathbb{E}}_{s \sim p_{\mathcal{T}}} \left[ (\boldsymbol{v}^\top (\boldsymbol{p}_{\cdot|s} - \boldsymbol{p}^*_{\cdot|s}))^2 \right]}$$

$$= \ell_{\mathcal{T}}(\{\boldsymbol{p}^*_{\cdot|s}\}, \boldsymbol{v}) + \sqrt{\boldsymbol{v}^\top \Sigma_{p_{\mathcal{T}}}(\Delta_{\{\boldsymbol{p}_{\cdot|s}\}}) \boldsymbol{v}}$$

*where $\Sigma_{p_{\mathcal{T}}}(g) = \mathop{\mathbb{E}}_{s \sim p_{\mathcal{T}}}[g(s)g(s)^\top]$ and $\Delta_{\{\boldsymbol{p}_{\cdot|s}\}}(s) = \boldsymbol{p}_{\cdot|s} - \boldsymbol{p}^*_{\cdot|s}$ are defined in Section B.*

*Proof.* The following sequence of inequalities proves it

$$\ell_{\mathcal{T}}(\{\boldsymbol{p}_{\cdot|s}\}, \boldsymbol{v}) = \mathop{\mathbb{E}}_{(s,y) \sim p_{\mathcal{T}}} \left[ \ell(\boldsymbol{v}^\top \boldsymbol{p}_{\cdot|s}, y) \right] \leq^{(a)} \mathop{\mathbb{E}}_{(s,y) \sim p_{\mathcal{T}}} \left[ \ell(\boldsymbol{v}^\top \boldsymbol{p}^*_{\cdot|s}, y) + |\boldsymbol{v}^\top (\boldsymbol{p}^*_{\cdot|s} - \boldsymbol{p}_{\cdot|s})| \right]$$

$$\leq^{(b)} \mathop{\mathbb{E}}_{(s,y) \sim p_{\mathcal{T}}} \left[ \ell(\boldsymbol{v}^\top \boldsymbol{p}^*_{\cdot|s}, y) \right] + \sqrt{\mathop{\mathbb{E}}_{s \sim p_{\mathcal{T}}} \left[ \left| \boldsymbol{v}^\top (\boldsymbol{p}^*_{\cdot|s} - \boldsymbol{p}_{\cdot|s}) \right|^2 \right]}$$

$$= \ell_{\mathcal{T}}(\{\boldsymbol{p}^*_{\cdot|s}\}, \boldsymbol{v}) + \sqrt{\boldsymbol{v}^\top \left( \mathop{\mathbb{E}}_{s \sim p_{\mathcal{T}}} \left[ (\boldsymbol{p}^*_{\cdot|s} - \boldsymbol{p}_{\cdot|s})(\boldsymbol{p}^*_{\cdot|s} - \boldsymbol{p}_{\cdot|s})^\top \right] \right) \boldsymbol{v}}$$

$$= \ell_{\mathcal{T}}(\{\boldsymbol{p}^*_{\cdot|s}\}, \boldsymbol{v}) + \sqrt{\boldsymbol{v}^\top \Sigma_{p_{\mathcal{T}}}(\Delta_{\{\boldsymbol{p}_{\cdot|s}\}}) \boldsymbol{v}}$$

where $(a)$ follows from 1-lipschitzness of $\ell$, $(b)$ follows from Jensen's inequality. $\qquad \square$

### E.6.2 HANDLING DISTRIBUTION SHIFT

**Lemma E.9.** *For any $g : \mathcal{S} \to \mathbb{R}^D$ and $p_{\mathcal{T}} \in \Delta_{\mathcal{S}}$, we have $\|\Sigma_{p_L}(g)^{-\frac{1}{2}} \Sigma_{p_{\mathcal{T}}}(g) \Sigma_{p_L}(g)^{-\frac{1}{2}}\|_2 \leq \gamma(p_{\mathcal{T}})^{-1}$*

*Proof.* By definition of $\gamma(p_{\mathcal{T}})$, we have that

$$\Sigma_{p_L}(g) = \mathop{\mathbb{E}}_{s \sim p_L}[g(s)g(s)^\top] = \sum_{s \in \mathcal{S}} p_L(s)g(s)g(s)^\top$$

$$\succcurlyeq \gamma(p_{\mathcal{T}}) \sum_{s \in \mathcal{S}} p_{\mathcal{T}}(s)g(s)g(s)^\top = \gamma(p_{\mathcal{T}}) \mathop{\mathbb{E}}_{s \sim p_{\mathcal{T}}}[g(s)g(s)^\top] = \gamma(p_{\mathcal{T}})\Sigma_{p_{\mathcal{T}}}(g)$$

Thus $\frac{1}{\gamma(p_{\mathcal{T}})}\Sigma_{p_L}(g) \succcurlyeq \Sigma_{p_{\mathcal{T}}}(g)$ and hence $\frac{1}{\gamma(p_{\mathcal{T}})}\Sigma_{p_L}(g)^{-\frac{1}{2}}\Sigma_{p_L}(g)\Sigma_{p_L}(g)^{-\frac{1}{2}} \succcurlyeq \Sigma_{p_L}(g)^{-\frac{1}{2}}\Sigma_{p_{\mathcal{T}}}(g)\Sigma_{p_L}(g)^{-\frac{1}{2}}$, which is equivalent to $\frac{1}{\gamma(p_{\mathcal{T}})}I_D \succcurlyeq \Sigma_{p_L}(g)^{-\frac{1}{2}}\Sigma_{p_{\mathcal{T}}}(g)\Sigma_{p_L}(g)^{-\frac{1}{2}}$. This finishes the proof. $\qquad \square$

**Lemma E.10.** *For matrices $\boldsymbol{X}, \boldsymbol{Y} \in \mathbb{R}^{D \times D}$ s.t. $\boldsymbol{X}, \boldsymbol{Y} \succcurlyeq 0$ and $\boldsymbol{Y}$ is full rank, we have that $\max_{\boldsymbol{a} \in \mathbb{R}^D, 0 < \|\boldsymbol{a}\| \leq \lambda} \frac{\boldsymbol{a}^\top \boldsymbol{X} \boldsymbol{a}}{\boldsymbol{a}^\top \boldsymbol{Y} \boldsymbol{a}} = \|\boldsymbol{Y}^{-\frac{1}{2}}\boldsymbol{X}\boldsymbol{Y}^{-\frac{1}{2}}\|_2$ for any norm $\| \cdot \|$.*

*Proof.* Note that $\frac{\boldsymbol{a}^\top \boldsymbol{X} \boldsymbol{a}}{\boldsymbol{a}^\top \boldsymbol{Y} \boldsymbol{a}}$ is independent of the scaling of $\boldsymbol{a}$. The following sequence of inequalities completes the proof

$$\max_{\boldsymbol{a} \in \mathbb{R}^D, 0 < \|\boldsymbol{a}\| \leq \lambda} \frac{\boldsymbol{a}^\top \boldsymbol{X} \boldsymbol{a}}{\boldsymbol{a}^\top \boldsymbol{Y} \boldsymbol{a}} = \max_{\boldsymbol{a} \in \mathbb{R}^D} \frac{\boldsymbol{a}^\top \boldsymbol{X} \boldsymbol{a}}{\boldsymbol{a}^\top \boldsymbol{Y} \boldsymbol{a}} = \max_{\boldsymbol{a} \in \mathbb{R}^D} \frac{\boldsymbol{a}^\top \boldsymbol{X} \boldsymbol{a}}{(\boldsymbol{Y}^{\frac{1}{2}}\boldsymbol{a})^\top (\boldsymbol{Y}^{\frac{1}{2}}\boldsymbol{a})}$$

$$= \max_{\boldsymbol{a} \in \mathbb{R}^D, \|\boldsymbol{Y}^{\frac{1}{2}}\boldsymbol{a}\|_2 = 1} \boldsymbol{a}^\top \boldsymbol{X} \boldsymbol{a} = \max_{\boldsymbol{b} \in \mathbb{R}^D, \|\boldsymbol{b}\|_2 = 1} (\boldsymbol{Y}^{-\frac{1}{2}}\boldsymbol{b})^\top \boldsymbol{X} (\boldsymbol{Y}^{-\frac{1}{2}}\boldsymbol{b})$$

$$= \max_{\boldsymbol{b} \in \mathbb{R}^D, \|\boldsymbol{b}\|_2 = 1} \boldsymbol{b}^\top \boldsymbol{Y}^{-\frac{1}{2}}\boldsymbol{X}\boldsymbol{Y}^{-\frac{1}{2}}\boldsymbol{b} = \|\boldsymbol{Y}^{-\frac{1}{2}}\boldsymbol{X}\boldsymbol{Y}^{-\frac{1}{2}}\|_2$$

$\qquad \square$

# F    EXPERIMENT DETAILS

For all experiments[9], we use the 117M parameter "small" GPT-2 model proposed in Radford et al. (2019) and implemented in HuggingFace (Wolf et al., 2019). Linear classification experiments (except for fine-tuning baseline in Table 1) are performed on *fixed* output features from GPT-2.

We note that the binary SST-2 dataset used in all experiments is comprised of complete sentences, and there are 6,920 train examples and 1,821 test examples. In particular, this dataset is smaller than the version included with the GLUE benchmark (Wang et al., 2018). This smaller version of SST-2 better fits the sentence completion hypothesis we propose.

## F.1    SOLVING DOWNSTREAM TASKS USING $f$ AND $\Phi p_f$

The features $f$ from GPT-2 for any input sequence $(w_1, \ldots, w_N)$ is the output embedding of the final token $w_N$ at the final layer, where $N$ is the input length and can be different for different inputs. This is also the embedding that is directly multiplied by the word embeddings to get the softmax distribution for language modeling, as in the theoretical setting. To use a prompt, the same prompt is added at the end of all inputs and the features are extracted for this modified input.

We use the `LogisticRegressionCV` class from the scikit-learn package to fit linear classifiers to all fixed features (i.e., no finetuning). We use the liblinear solver and one-vs-rest loss function unless it catastrophically fails (e.g., close to random performance) on a particular multi-class task. In that case, we use the stochastic average gradient (SAG) algorithm with multinomial loss. We use 5-fold cross validation for all experiments and test values for the regularization parameter $C$ between $1e{-}6$ and $1e4$ for small datasets (i.e., fewer than 10K examples) and between $1e{-}3$ and $1e3$ for larger datasets.

**Details about word subsets:**    For all of the results presented in Table 1, we use a pre-trained GPT-2 model. For SST, we use the prompt "This movie is " when indicated. For AG News, we use the prompt "This article is about " when indicated.

We compute the conditional probability of selecting a subset of words to complete the sentence. For AG News, this subset is: 'world', 'politics', 'sports', 'business', 'science', 'financial', 'market', 'foreign', 'technology', 'international', 'stock', 'company', 'tech', 'technologies'. For SST, this subset is: ':)', ':(', 'great', 'charming', 'flawed', 'classic', 'interesting', 'boring', 'sad', 'happy', 'terrible', 'fantastic', 'exciting', 'strong'. For AG News, the class words we use are: 'foreign', 'sports', 'financial', 'scientific'. For SST, the class words we use are ':)' and ':('.

We account for BPE tokenization by using the encoding of the word directly and the encoding of the word with a space prepended. We then filter to use only words that encode to a single BPE token.

**Tests on additional datasets:**    We also test the performance of pre-trained GPT-2 embeddings $f$ and the conditional mean embeddings $\Phi p_f$ on the DBPedia (Auer et al., 2007), Yahoo Answers (Zhang et al., 2015), TREC (Li & Roth, 2002), IMDb (Maas et al., 2011), Customer Review (CR) (Hu & Liu, 2004), and MPQA polarity (Wilson & Wiebe, 2003) datasets in Table 2. We limited the training set size to 250K for larger datasets (i.e., DBPedia and Yahoo Answers). For CR and MPQA, we follow Zhang et al. (2015) and average the performance across 10 random 90-10 train-test splits of the dataset.

We find that $\Phi p_f$ consistently has comparable performance to $f$ across non-sentiment and sentiment downstream classification tasks. We include baseline results of bag of $n$-grams (BonG) for most tasks and the mLSTM model (Radford et al., 2017) for sentiment tasks. BonG performs quite well on the larger datasets, but not as well on smaller datasets, due to the high dimensionality of features.

For sentiment tasks, adding a prompt almost always boosts performance. We also demonstrate that much of the performance can be recovered by only looking at "positive" and "negative" or ":)" and ":(" as class words. Using these 2-dimensional features is even more sample-efficient than the standard 768-dimensional ones.

---

[9]Link to code: `https://github.com/sadhikamalladi/mathematical-exploration-downstream-tasks`.

Table 2: GPT-2 performance without fine-tuning on downstream task test sets with $k$ classes. We provide the performance of bag of $n$-grams (BonG) as an approximate baseline for these tasks. AG News, DBPedia and Yahoo performances were reported in Zhang et al. (2015), and the other tasks were reported in Khodak et al. (2018). We also include results from mLSTM (Sentiment Neuron) (Radford et al., 2017) for the sentiment-related classification tasks (SST, IMDb, CR, and MPQA) with numbers reported from Khodak et al. (2018). Furthermore, we include results for BERT (Devlin et al., 2018) features without fine-tuning, where we use the output features for the first position of an input for linear classification. An asterisk indicates we add a standard sentiment prompt "The sentiment is" to each input, but for AG News we used the prompt "This article is about". We also tested the performance of the conditional probability distribution over "positive" and "negative" as well as ":)" and ":(" on the sentiment-related tasks with and without the prompt.

| Task | $k$ | $f(s)$ | $\Phi p_f(s)$ | $\boldsymbol{p}_{\cdot\|s}$: pos,neg | $\boldsymbol{p}_{\cdot\|s}$: :),:( | BonG | mLSTM | BERT |
|---|---|---|---|---|---|---|---|---|
| *Non-sentiment* | | | | | | | | |
| AG News | 4 | 90.7 | 84.6 | - | - | 92.4 ($n = 5$) | - | 88.9 |
| AG News* | 4 | 91.1 | 88.2 | - | - | - | - | 89.9 |
| DBPedia | 14 | 97.2 | 88.2 | - | - | 98.6 ($n = 5$) | - | 98.7 |
| Yahoo | 10 | 69.2 | 56.7 | - | - | 68.5 ($n = 5$) | - | 65.0 |
| TREC | 6 | 93.6 | 87.8 | - | - | 89.8 ($n = 3$) | - | 90.6 |
| *Sentiment* | | | | | | | | |
| SST | 2 | 87.5 | 83.3 | 74.9 | 78.7 | 80.9 ($n = 2$) | 91.8 | 85.8 |
| SST* | 2 | 89.4 | 87.3 | 80.8 | 79.1 | - | - | 84.1 |
| SST fine | 5 | 49.2 | 43.5 | 37.5 | 39.2 | 42.3 ($n = 3$) | 52.9 | 43.5 |
| SST fine* | 5 | 49.4 | 48.0 | 41.5 | 40.2 | - | - | 43.3 |
| IMDb | 2 | 88.1 | 82.7 | 73.8 | 76.2 | 89.8 ($n = 3$) | 92.3 | 82.2 |
| IMDb* | - | 88.4 | 85.3 | 81.8 | 80.9 | - | - | 84.0 |
| CR | 2 | 86.8 | 84.6 | 74.9 | 80.0 | 78.3 ($n = 3$) | 91.4 | 85.5 |
| CR* | - | 87.9 | 87.1 | 82.5 | 79.4 | - | - | 84.6 |
| MPQA | 2 | 86.0 | 79.2 | 75.6 | 70.7 | 85.6 ($n = 3$) | 88.5 | 87.3 |
| MPQA* | - | 87.8 | 86.1 | 80.3 | 71.4 | - | - | 88.1 |

We also include results using the pre-trained BERT base cased model (Devlin et al., 2018; Wolf et al., 2019), using the embedding at the first token as input to the downstream task. We also tried using the mean embedding and last token embedding and found that the first token embedding is often the best. Moreover, the first token embedding is what is extracted in the traditional usage of BERT on downstream tasks, though we note that it is rare to use BERT without fine-tuning.

## F.2 FINETUNING EXPERIMENTS

As a strong baseline, we finetune the GPT-2 features along with learning a linear classifier for the SST and AG News classification tasks and report accuracy numbers in Table 1. We use a maximum sequence length of 128 BPE tokens for downstream inputs of SST-2 and a maximum length of 400 BPE tokens for AG News inputs. We use the end of sentence token as the padding token. The datasets are described below.

1. AG News has 108K train examples, 12K dev examples, 7600 test examples. We split the train set for AG News into train and dev (90-10) and use the same test set as the non-finetuning experiments.

2. The sentence version of SST-2 has 6,920 train examples (same as non-finetuning), and 810 examples for dev and test each (split the original test set in half).

3. Fine-grained SST-2 has 8,544 train examples (same as non-finetuning), and 1,105 examples each for the dev and test data (split the original test set in half).

To select the best hyperparameter configuration, we run a grid search over learning rate and batch size. We train each model for 10 epochs. For all datasets, we test learning rates $5\mathrm{e}{-}5$, $1\mathrm{e}{-}4$, and

Table 3: Comparing Quad features to cross-entropy features for GPT-2 trained on the IMDb un-labeled corpus (Maas et al., 2011). In this experiment we fix $\Phi$ to be the word embeddings from prertained GPT-2 model for the cross-entropy objective. For the Quad objective, we initialize $\Phi$ to be the SVD of the pre-trained embeddings. An asterisk indicates that we added the prompt "This movie is " to each input.

| Task | $f(s)$ (xent) | $\Phi p_f(s)$ (xent) | $f(s)$ (Quad) |
|------|------|------|------|
| SST | 82.1% | 79.9% | 77.3% |
| SST* | 83.1% | 81.1% | 80.7% |

Table 4: Comparing Quad features to cross-entropy features for GPT-2 trained on the Amazon corpus. An asterisk indicates that we added the prompt "This movie is " to each input. Note that the validation loss was still decreasing at the time of measurement.

| Task | $f(s)$ (xent) | $\Phi p_f(s)$ (xent) | $f(s)$ (Quad, learned $\Phi$) |
|------|------|------|------|
| SST | 89.4% | 89.7% | 79.2% |
| SST* | 89.7% | 89.2% | 84.3% |

$3e-4$. For both version of SST-2, we try batch sizes 8, 16, and 32, and for AG News, we try batch sizes 8, 12, and 16. We note that the longer sequence length of AG News inputs required us to use parallelization across multiple GPUs to simulate larger batch sizes, which made batch size 32 prohibitively expensive to test.

We take the hyperparameter configuration that achieves the best performance on the dev set and then perform fine-tuning using those settings with three different random seeds: 8, 33, and 42. We then report the average performance on the test set in Table 1.

We perform the hyperparameter grid search over the standard datasets and then perform fine-tuning using the best settings on the dataset with task-specific prompts added. For SST-2, we use the prompt "This movie is ", and for AG News we use "This article is about ".

### F.3 TESTING QUAD OBJECTIVE

We test two models with the same parametrizations, one trained using our Quad objective and another trained with the standard cross-entropy objective using the unlabeled IMDb corpus (Maas et al., 2011) and the Amazon product review corpus (McAuley et al., 2015). We slightly modify the standard architecture of GPT-2 to generate Tables 3 and 4. First we add a single linear layer (that is trained) on top of the output features of the standard Transformer architecture. Furthermore, instead of tying the input and output word (token) embeddings, we learn them separately so that $f$ and $\Phi$ are independent functions; this is more in line with out theoretical setup. We fix the input embeddings and the positional embeddings to be the parameters from the pre-trained GPT-2.

For Quad, we initialize $\Phi$, the output embeddings, using the singular vectors of the pre-trained word embeddings $\Phi$. For the cross-entropy models, we initialize $\Phi$ to be the full pre-trained word embeddings $\Phi$, because we found that initializing with the singular vectors harmed performance. Given our parameterization, initializing with the singular vectors is as expressive as initializing with the pretrained embeddings $\Phi$ themselves; however it potentially lends a better optimization landscape and speeds up training for our new objective Quad. As described in Section 5.2, we minimize the following objective

$$\ell_{quad}(f, \Phi) = \mathop{\mathbb{E}}_{(s,w)}\left[-f(s)^\top \phi_w + \frac{1}{2}\|\Phi^\top f(s)\|^2\right] \tag{36}$$

where $(s, w)$ are sampled from the text corpus. The implementation of the Quad loss is the same as the standard cross-entropy loss, the main difference being the second term: it is $\frac{1}{2}\|\Phi^\top f(s)\|^2$ for Quad instead of the log-partition function $\log\left(\sum_{w'} e^{f(s)^\top \phi_{w'}}\right)$ in the cross-entropy objective.

True log partition versus learned quadratic function

Figure 2: Fit of the learned quadratic function to the log partition function on various datasets for features computed by the full, pre-trained GPT-2. We also plot the $y = x$ line for reference. These plots are meant to verify Assumption 4.1.

Because IMDb is a smaller dataset, we fix $\Phi$ at its initialization and only train $f$ to generate Table 3. When training on the Amazon dataset, we initialized $\Phi$ the same way as we did for the IMDb dataset, but we allowed $f$ and $\Phi$ to both be trained, since more data was available. To train the models, we use the standard learning rate schedule as in in Radford et al. (2019). To learn a model on IMDb, we use a context size of 512 BPE tokens, and for the Amazon reviews dataset (McAuley et al., 2015), we use the standard context length of 1,024 BPE tokens.

We observe that training using Quad, in both cases, yields comparable performance to the language model on the SST task, but always slightly worse. According to the theory, features $f(s)$ from Quad should learn $p^*_{\cdot|s}$ on a subspace, just like $\Phi p_f$ from cross-entropy models, thus making the comparison between these two important. Furthermore, adding a prompt consistently improves performance for both objectives. While Quad did not beat the cross-entropy in either case, its good performs at least demonstrates that insights from the theoretical analysis can translate to practical algorithms. We leave exploring the gap in performance between Quad and cross-entropy and a more extensive evaluation of Quad for future work.

## F.4   Learning the quadratic approximation of the log-partition function

In Assumption 4.1, we assert that there is a quadratic fit for the log partition function, which allows us to show in Lemma 4.3 that a linear relation holds between $f$ and $\Phi p_f$. We validate these theoretical findings by fitting a quadratic function to the log partition function for a subset of embeddings from the IMDb, SST, and AG News datasets (Figure 1). Here, we describe how we learned $\boldsymbol{A}, \boldsymbol{b}$ and $c$. To ensure $\boldsymbol{A}$ is symmetric and positive semi-definite as required, we parametrize $\boldsymbol{A} = \boldsymbol{U}\boldsymbol{U}^T$. As defined earlier, the partition function $Z_\theta = \sum_{w'} e^{\theta^\top \phi_{w'}}$ and $\Phi p_\theta = \sum_{w'} \frac{e^{\theta^\top \phi_{w'}}}{Z_\theta} \phi_{w'}$ for any $\theta \in \mathbb{R}^d$. We minimize the following objective function:

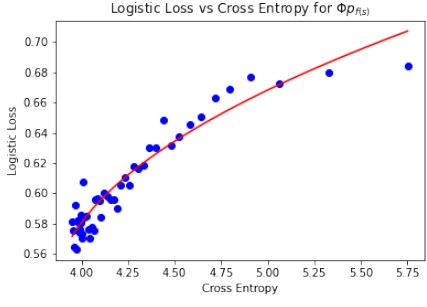 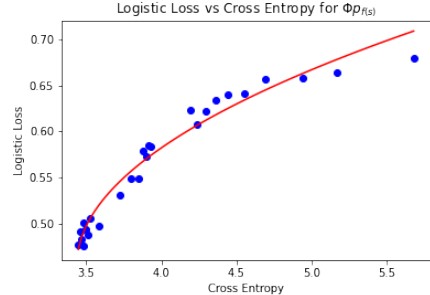

(a) Trained on IMDb (Maas et al., 2011)       (b) Trained on Amazon (McAuley et al., 2015)

Figure 3: Logistic loss of conditional mean features on the SST-2 task for various checkpoints of a GPT-2 architecture trained on IMDb and Amazon. The reported cross-entropy is measured on the validation set. The red trend shows the fit of a square-root function, which is what the upper bound in Theorem 4.2 looks like.

$$\mathcal{L}(\boldsymbol{U}, \boldsymbol{b}, c) = \mathbb{E}_{\theta} \left[ \lambda_1 \left( \log(Z_\theta) - \frac{1}{2} \theta^\top \boldsymbol{U}\boldsymbol{U}^\top \theta - \theta^\top \boldsymbol{b} - c \right)^2 + \lambda_2 \left\| \Phi p_\theta - \boldsymbol{U}\boldsymbol{U}^\top \theta - \boldsymbol{b} \right\|^2 \right] \quad (37)$$

In practice, we train only on the regression loss (i.e., $\lambda_1 = 0$, $\lambda_2 = 1$) for the most promising results. Note that the regression term is trying to learn a linear relationship between between $\theta$ and $\Phi p_\theta$ that Lemma 4.3 aims to prove. This ends up learning a matrix $\boldsymbol{A} = \boldsymbol{U}\boldsymbol{U}^\top$ and vector $\boldsymbol{b}$ that also satisfy the quadratic form of $\log(Z_\theta)$ from Assumption 4.1.

We use 20,000 examples from a mix of IMDb, SST, and AG News embeddings as the training set. Thus we sample $\theta$ by sampling $s$ from the aforementioned datasets and set $\theta = f(s)$, $f$ being the feature map from pretrained GPT-2. We use the Adam (Kingma & Ba, 2014) optimizer with learning rate 1e−3 for $\boldsymbol{U}$ and learning rate 1e−4 for $\boldsymbol{b}$ and $c$. We decay the learning rate every 50 steps by a factor of 0.1. We use the $\boldsymbol{U}$ obtained after 8 epochs of training. We further demonstrate the quality of the learned fit by plotting the true log partition and estimated log partition function for embeddings from other datasets in Figure 2.

### F.5 EXPERIMENTALLY CHECKING THEOREM 4.2

Theorem 4.2 can be informally summarized as stating that an $\epsilon$ suboptimality in the cross-entropy of a $d$-dimensional language model propagates to a $\sqrt{\epsilon}$ increase in the logistic loss. We note that the $\tau, B,$ and $\gamma(p_\mathcal{T})$ factors are fixed for a given pre-training corpus and downstream task, so we can empirically test if this square root relationship holds in practice. In particular, Theorem 4.2 says

$$\ell_\mathcal{T}(\Phi p_f) \leq \tau + \sqrt{2B^2 \left( \gamma(p_\mathcal{T}) \right)^{-1} \left( \ell_{\text{xent}}(f, \Phi) - \ell^*_{\text{xent}} \right)} \quad (38)$$

Of these, $\tau, B, \gamma(p_\mathcal{T})^{-1}$ and $\ell^*_{\text{xent}}$ are independent of the language model $(f, \Phi)$ and only depend on the task $\mathcal{T}$ and language modeling distribution. Thus we can rewrite this as $\ell_\mathcal{T}(\Phi p_f) \leq c + a\sqrt{\ell_{\text{xent}}(f, \Phi) - b}$ for suitable constants $a, b, c \in \mathbb{R}$. The left hand side, $\ell_\mathcal{T}(\Phi p_f)$, is the logistic loss of conditional mean features from language model $(f, \Phi)$ on task $\mathcal{T}$ and $\ell_{\text{xent}}(f, \Phi)$ is the cross-entropy loss of the language model, both of which can be measured in practice.

We train a 117M parameter GPT-2 model from scratch on the IMDb and Amazon corpora, described in Section F.3. We maintain checkpoints during training, and for each checkpoint, we measure the cross-entropy of the model on the validation set as well as the performance of the conditional mean features $\Phi p_f$ on SST-2. Plotting these values together yields Figure 3.

We furthermore fit a square root trend, shown in red, to these points. We learn $a, b, c$ such that $y \approx a\sqrt{x - b} + c$, where $y = \ell_\mathcal{T}(\Phi p_f)$ is the logistic loss and $x = \ell_{\text{xent}}(f, \Phi)$ is the cross-entropy loss. For this, we perform a grid search over 100 evenly spaced valid values of $b$, and for each $b$,

we perform linear regression on $\sqrt{x - b}$ to find $a$ and $c$. We choose the $a, b, c$ that maximizes the $r$-value of the regression. While Theorem 4.2 only provides an upper bound on the logistic loss, this experiment shows that some square-root trend is observable in practice.

