# OpenReview forum: "A Mathematical Exploration of Why Language Models Help Solve Downstream Tasks"
_ICLR.cc/2021/Conference — ICLR 2021 Poster_

### Official Review · AnonReviewer1 · 2020-10-25
**A well-written paper that mathematizes an empirical fact**

**Rating:** 7
**Confidence:** 4

**Review:**

**Summary**
This work relates a pre-training performance with a downstream performance for tasks that _can_ be reformulated as next word prediction tasks. The authors show that for such tasks, if the pre-training objective is $\epsilon$-optimal, then the downstream objective of a linear classifier is $\mathcal{O}(\sqrt{\epsilon})$-optimal.

**Strengths**
- To the best of my knowledge, this is the first work that _mathematically_ justifies the connection between the pre-training objective and the downstream performance.
- The proof technique (pre-training performance $\to$ covariance of pre-training errors $\to$ covariance of downstream errors $\to$ downstream performance) is itself interesting. If the paper is accepted I am looking forward to seeing a high-level proof sketch in the main part as is done in Section 2.1 of [Arora et al. (2015)](https://arxiv.org/abs/1502.03520). A three-line explanation at the end of Section 4.1 seems a bit scarce to me.
- The paper is well written: it gives an appropriate context, presents the main theoretical results, and verifies _some_ of the claims experimentally.

**Major concern**
If I understand correctly (and please correct me if I am wrong), in Theorem B.1, the ratio between the downstream error $\ell_\mathcal{T}(\\{p_{\cdot\mid s}\\}) - \tau$ and the pre-training error $\ell_\text{xent}(\\{p_{\cdot\mid s}\\})-\ell_\text{xent}^\ast$ is _hidden_ in the $\gamma(p_{\mathcal{T}}; \\{p_{\cdot\mid s}\\})$ coefficient. Let me elaborate on this:
1. In Lemma D.1 (with the help of Lemma D.9) you show that $\frac{1}{\gamma(p_{\mathcal{T}}; \\{p_{\cdot\mid s}\\})}$ is an upper bound for the ratio $\frac{\boldsymbol{v}^\top\Sigma_{p_{\mathcal{T}}}\boldsymbol{v}}{\boldsymbol{v}^\top\Sigma_{p_{L}}\boldsymbol{v}}$
2. The latter ratio seems proportional to the ratio $\frac{\ell_\mathcal{T}(\\{p_{\cdot\mid s}\\}) - \tau}{\ell_\text{xent}(\\{p_{\cdot\mid s}\\})-\ell_\text{xent}^\ast}$. I am not sure on this---my intuition is based on your Lemma D.2 and the fact that for a $p_{\cdot\mid s}$ with full support a non-precise reverse version of [Pinsker's inequality](https://en.wikipedia.org/wiki/Pinsker%27s_inequality#Inverse_problem) holds.

In a nutshell, aren't you showing that
$$\text{downstream error}=\mathcal{O}\left(\sqrt{\text{pre-training error}\cdot\frac{\text{downstream error}}{\text{pre-training error}}}\right)\qquad ?$$

**Issues**
- Why don't you verify the main claim---$\epsilon$-optimality in pre-training propagates as $\mathcal{O}(\sqrt{\epsilon})$-optimality on downstream---empirically?For this, you may want to vary the language modeling performance (e.g. by pruning the language model) and then verifying that the downstream loss increase is indeed $\mathcal{O}(\sqrt{\text{pre-training loss increase}})$. I believe such an experiment will definitely make the submission stronger.
- I don't see why your theory does not generalize to a _masked_ language modeling (MLM). Why do we need to treat $s$ as the left-context only? Given the success of MLM as a powerful pre-training objective, please consider formulating your claims in a more general way.

**Minor issues**
- At the beginning of Section 2.3, $p_{\mathcal{T}}$ is introduced as a distribution over $\mathcal{S}\times\\{\pm1\\}$, but later (e.g. in formula (5)), it is used as a distribution over $\mathcal{S}$ only. Please clarify/fix this.
- What is the "margin of task $\mathcal{T}$" mentioned on p.5? Is it a margin of an SVM classifier that solves $\mathcal{T}$?

**Limitations**
- The authors admit that their work is limited to a particular type of downstream tasks. Indeed, it is not clear how one can reformulate e.g. linguistic tasks (like POS-tagging or dependency parsing) as a next word prediction task.

**Update (after the author's response)**: During the rebuttal, the authors clarified my major concern, as well as provided additional experiments that verify the main claim of the paper. I am totally satisfied with the author's response, hence I am changing the score 6 $\to$ 7

---

> ### Author Response · Authors · 2020-11-20
> **Response to Reviewer 1**
>
> Thank you for comments. We address the concerns you raise below.
>
> **[Ratio between errors is hidden in the coefficient]**
> This is not exactly true and the coefficient is instead a measure of how the different the distribution of contexts $p_{L}$ and $p_{\mathcal{T}}$ are, as measured by the learned LM. There are a few quick ways to see this:
>
> - $\gamma(p_{\mathcal{T}}, \{p_{\cdot|s}\})$ is always lower bounded by $\gamma(p_{\mathcal{T}})$ which only depends on $p_{L}$ and $p_{\mathcal{T}}$ and has nothing to do with the absolute value of the error made by the LM.
>
> - When $p_{L} = p_{\mathcal{T}}$, $\gamma(p_{\mathcal{T}}, \{p_{\cdot|s}\}) = 1$ always, again independent of the scale of errors
>
> Here is a proof sketch that clarifies more. If  $\boldsymbol{v}^*$ is the optimal classifier for $\{p^*_{\cdot|s}\}$ that gets error less than $\epsilon$
>
> $$\frac{\ell_{\mathcal{T}}(\{p_{\cdot|s}\}) - \tau}{\sqrt{\ell_\text{xent}(\{p_{\cdot| s}\})-\ell_\text{xent}^*}}     \le    \frac{\ell_{\mathcal{T}}(\{p_{\cdot|s}\}, {\boldsymbol{v}^*}) - \tau}{\sqrt{\ell_\text{xent}(\{p_{\cdot| s}\})-\ell_\text{xent}^*}}   =     \frac{\ell_{\mathcal{T}}(\{p_{\cdot|s}\}, {\boldsymbol{v}^*}) - \tau}{\sqrt{{\boldsymbol{v}^*}^\top\Sigma_{p_{\mathcal{T}}}{\boldsymbol{v}^*}}} \sqrt{\frac{{\boldsymbol{v}^*}^\top\Sigma_{p_{\mathcal{T}}}{\boldsymbol{v}^*}}{{\boldsymbol{v}^*}^\top\Sigma_{p_{L}}{\boldsymbol{v}^*}}} \sqrt{\frac{{\boldsymbol{v}^*}^\top\Sigma_{p_{L}}{\boldsymbol{v}^*}}{\ell_\text{xent}(\{p_{\cdot| s}\})-\ell_\text{xent}^*}}$$
>
> The first term in the product is upper bounded by 1 using 1-lipschitzness of $\ell$ and optimality of $\boldsymbol{v}^*$ (around Equation 14) and the third term is upper bounded by $\sqrt{2\|\boldsymbol{v}^*\|^2\_{\infty}}$ (Lemma D.2) which is proved using Pinsker's inequality. The crux of the argument is to connect the cross-entropy loss in LM to the logistic/hinge loss in classification, and that is captured in these two terms. In fact, for the softmax case we prove a stronger $d$-dimensional variant of Pinsker's inequality in Lemma D.6. The middle term is upper bounded by the coefficient  $1/\sqrt{\gamma(p_{\mathcal{T}}, \{p_{\cdot|s}\})}$ (Lemma D.9) to handle the distribution shift as described above.
>
>
> **[Can we empirically verify the $\mathcal{O}(\sqrt{\epsilon})$ dependence?]**
> Note that $\mathcal{O}(\sqrt{\epsilon})$ is only an *upper bound* on the classification error and thus may not hold in practice. The reasoning is similar to why Pinsker's inequality will not be tight for all distributions, but only for some worst case ones. Regardless, we test the dependency for various checkpoints in the training process of the language model and find a weak $\sqrt{\epsilon}$-like dependency. We add plots that demonstrate this dependence and details for this experiment in Appendix E.4 (revised pdf).
>
>
> **[Why does theory not directly apply to masked language modeling]**
> Our analysis can directly work for masked language modeling (BERT) if only one token is masked per sequence. However in practice multiple ($\approx 12$\%) tokens are masked. Thus BERT receives only masked sentences during pretraining while downstream tasks only use unmasked input sentences. Handling this mismatch is not as straightforward and we are separately investigating if we can extend our analysis to BERT-style models.
>
> **[What is margin of task $\mathcal{T}$]**
> We use margin here loosely, but it is related to the margin of an SVM classifier that solves $\mathcal{T}$. The discussion right after that sentence aims to clarify this point, that $B$ is how large the norm needs to be to make $\boldsymbol{v}^\top p^*_{\cdot|s}=\Omega(1)$, which is related to SVM notion of margin. One difference is that $B$ is measured using infinity norm as opposed to Euclidean norm in SVMs.

---

> > ### Comment · AnonReviewer1 · 2020-11-20
> > **Satisfied with the author's response**
> >
> > Thank you for the clarifications! I am satisfied with the response. One little issue: did you forget to update the PDF (as I cannot find Appendix E.4)?

---

> > > ### Author Response · Authors · 2020-11-20
> > > **We see the updated version**
> > >
> > > Oh that is weird because we see the updated version, with Appendix E.4 starting at the end of page 24 and the plots on page 25. Does it say "(modified: 20 Nov 2020)" right below the title on this page?

---

> > > > ### Comment · AnonReviewer1 · 2020-11-21
> > > > **Now I see the updates**
> > > >
> > > > Yes, now I see the updated version with Appendix E.4. This is great! I'm changing my score from 6 to 7.

---

> > > > > ### Author Response · Authors · 2020-11-23
> > > > > **Glad that is resolved**
> > > > >
> > > > > Thank you for the update!

---

### Official Review · AnonReviewer3 · 2020-10-30
**Insightful theoretical analysis on the benefits of language model pre-training.**

**Rating:** 8
**Confidence:** 4

**Review:**

This paper studies why language model pre-training has been such an effective technique in improving downstream performance across a wide range of NLP tasks recently.  In particular, it considers language models which compute a probability distribution over the next word in a text, given the previous context.  Then, taking inspiration from recent work that shows that many downstream tasks can be reframed as sentence completion tasks, it defines a “natural task” as one on which a sparse linear model over the output of the “true” language model (next word probability distribution, conditioned on context) attains strong performance.  Theoretically, it shows that language models which are close to the “true” language model are guaranteed to attain strong performance on natural tasks.  Empirically, it demonstrates that several NLP tasks are “natural”.


**Strengths**
- The paper is generally quite clearly written and the claims are well-validated.
- The definitions, models, and assumptions in the paper are intuitive and clear (e.g., natural task).
- The analysis which builds on these definitions/models/assumptions provides meaningful theoretical insight into why language model pre-training may be so beneficial for downstream training.  It provides a nice theoretical framework for thinking about the connection between language models and downstream tasks, which future work could build on.
- The empirical validation is thoughtful and relatively thorough.  Even though the results don’t show that the proposed loss function and proposed “conditional mean features” give improvements over baselines, the empirical results show that the basic assumptions and definitions in the theoretical analysis are relatively realistic.  For example, Figure 4 validates Assumption 4.1 (log-partition function is roughly quadratic in theta), and Table 1 shows many real tasks are approximately “natural”.  Furthermore, when there is a gap between the empirical results and the theoretical results (e.g., validation of Lemma 4.3 at the end of Section 4), the paper makes these limitations clear, which I appreciated very much as a reader (paper does not over-claim its contributions).

**Weaknesses**
* Unclear if there are real practical applications to the insights from this paper.  Neither the proposed “Quad” loss function, nor the theoretically inspired “conditional mean features”, perform better than the baselines.
* The current analysis doesn’t apply directly to BERT, which is trained to predict masked words in a sentence, instead of the next word.  Furthermore, BERT doesn’t predict these masked words using a linear softmax model over a contextual embedding for the whole sentence, which is the assumed structure for the softmax language models considered in the analysis.  (This limitation is acknowledged in the conclusion, which is good).
* The paper doesn’t explain why learning a linear model directly on the context embeddings f(s) performs better than using the contextual mean embeddings.
  * One idea I had here: Could you define a natural task as one for which there exists a sparse linear model over the *logits* of p*( . | s) which performs well, instead of a model directly over p*( . | s)?  Due to the very “flat” portions of the softmax function, there can be meaningful differences between the logits corresponding to 2 different words, but the LM probabilities for those words are extremely similar (and thus, harder for a linear model to distinguish).  With this definition, a linear model of the logits is also a linear model over the context embeddings f(s) directly.
* There are some points in the paper that could be made clearer.
  * I think it should be discussed earlier (in intro/related work) why the paper focuses on language models which do next word prediction via linear softmax models over fixed dimensional context embeddings, and that BERT is out of scope.
  * I think there should be more discussion about the implications of Proposition 2.2.  As I understand it, this result shows that any part of p_f(s) orthogonal to row-span(Phi) doesn’t affect the cross-entropy of the language model (first order optimality condition would still be satisfied).  However, this doesn’t necessarily imply that p_f(s) will be in span(Phi) for all contexts s.  In particular,  the architecture of the embedding model f likely constrains f in such a way that makes it impossible for p_f(s) to be in span(Phi) for all contexts s.  Furthermore, at the end of section 3 it should be better explained why the assumption that p_f(s) is in span(Phi) for all s implies that Definition 3.2 should only consider sparse models v which are in this span as well (decompose v = v_in + v_out (component of v in the span, and orthogonal to the span), v^T p = (v_in + v_out)^T p = v_in^T p).
  * I found the discussion in Section 4.1 pretty confusing.  In particular the part that argued why B = O(1/alpha).

Overall, I really enjoyed reading this paper, and found it to be quite insightful.  It provided me with a much more thoughtful explanation for why language model pre-training improves downstream task performance, beyond simply “it helps learn good general representations of language using large amounts of unlabeled text data” (my previous reasoning).  As a result, I recommend acceptance for this paper.

NIT:
- Grammar last sentence of Section 1.1 (“…analyze the efficiency *of* …”)
- Proposition 2.2: Maybe write “\forall s \in S” instead of “\forall s ~ p_L”.
- Section 3: ...append a prompt like “This movie is” (the final quotation mark is on the next line).
- Equation (5).  Use “sup” instead of “max”.
- Discussion in Section 4.1
- I think Figure 4 should be explained in more detail (in caption and/or text).
- Using capital and lower case tau in Theorem 4.2 is confusing notation.
- Similarly, using bold and not-bold B in Theorem 5.2 is confusing notation.
- After definition 5.1, what does Omega[w] = Omega[w’] mean?
- In Table 1, can you explain more explicitly (in caption and text) what “subset” and “class words” means? Also, can you add a column where a dense linear model over p_f(s) is used?

---

> ### Author Response · Authors · 2020-11-20
> **Response to Reviewer 3**
>
> Thank you for the careful reading and the thorough review! We discuss the weaknesses and answer questions below.
>
>
> **[Unclear if there are real practical applications to the insights from this paper. Neither the proposed "Quad" loss function, nor the theoretically inspired "conditional mean features", perform better than the baselines.]**
> Indeed we were hoping that conditional mean features or the Quad objective would do better than the standard setup, but that wasn't to be (it was already a pleasant surprise that they performed as well as they did!). On the bright side, this confirms that the story is not complete and there are still gaps to be filled. We hope that our work builds a foundation for systematically tackling other setups (e.g. BERT) and hopefully inspires more successful practical applications.
> Another insight from our analysis: if we know $\Phi$ such that tasks are likely to be natural w.r.t. it, then using those word embeddings during language modeling training can guarantee good performance on downstream tasks. Given that $\Phi$ is normally trained end-to-end, we believe this finding can inspire a more intentional design of the word embeddings.
>
> **[The current analysis doesn't apply directly to BERT]**
> As you rightly point out, the extension to BERT like models is not straightforward. This is because BERT receives sentences with around 12\% of the tokens masked during pretraining, while downstream tasks use unmasked input sentences. We are currently working on handling this mismatch to show guarantees for BERT-like masked language models. The current analysis can extend to a variant of BERT where just 1 token is masked during pretraining. We will add some more discussion about BERT.
>
> **[No explanation for why $f(s)$ is better than $g_{f,\Phi}$]**
> Defining natural tasks to be linear functions of the logits ($\log(p_{\cdot|s}^*)$) is indeed a natural idea to explain the success of $f$ and we considered it. However this has a fundamental issue: we do not yet have a closed form expression for the optimal $d$-dimensional logits ($f^*(s)$) for all distributions $p_{\cdot|s}^*$. We do not even know if $f^*(s)$ is a linear function of $\log(p_{\cdot|s}^*)$ (this is likely not true). In contrast, first order optimality condition of the cross-entropy objective directly gives us  $p_{f^*(s)}$ is a linear function of $p_{\cdot|s}^*$. Thus studying the logits directly requires some more thought and we are currently working on it.
>
> **[Discussion and clarification about Proposition 2.2]**
> We will add some discussion about this. Note that our result does not imply that $p_{f^*(s)}$ lies in the subspace of $\Phi$. In fact for many $p_{\cdot|s}^*$, there might be a unique optimal $d$-dimensional softmax solution $p_{f^*(s)}$ that satisfies Proposition 2.2, and that does not lie on the subspace. This is not because of the model architecture, but due to the inherent limitation of using $d$ dimensions.
>
> **[Discussion in Section 4.1 is confusing]**
> We give intuitive interpretations for the various mathematical components of the bound. $B$ denotes the $\ell_{\infty}$ norm required for a classifier $\boldsymbol{v}$ to achieve small error ($\tau$, say 0.1). Both hinge loss and logistic loss can be small only if the absolute value of prediction prediction $|\boldsymbol{v}^\top p_{\cdot|s}| = \Omega(1)$ on average. If you imagine the classifier to have value $B$ on the indicative positive words and $-B$ on negative words, then $\boldsymbol{v}^\top p_{\cdot|s} = B(p_{\cdot|s}[\text{+ve words}] - p_{\cdot|s}[\text{-ve words}])$, thus you need $B = \Omega(\frac{1}{p_{\cdot|s}[\text{+ve words}] - p_{\cdot|s}[\text{-ve words}]}) = \Omega(\frac{1}{p_{\cdot|s}[\text{+ve words}]}) = \Omega(1/\alpha)$. Thus $B$ is smaller when the task depends on more frequent set of words.
>
> **[Notation]**
> Thank you for pointing out typos and suggestions for notations. We will add more details in the caption of Table 1. We can also add a column for linear classifier over the entire vector $p_{f(s)}\in\mathbb{R}^V$. Note that for smaller datasets like SST, this might not be sample efficient. $\Omega[w]$ refers to the row of $\Omega$ corresponding to word $w$.

---

### Official Review · AnonReviewer4 · 2020-11-02
**Interesting analysis framework and results but some conceptual concerns**

**Rating:** 6
**Confidence:** 2

**Review:**

Summary: This paper presents an explanation of why pretraining on language modeling (LM) helps performance on downstream text classification tasks. The explanation relies on formulating classification tasks as next word prediction tasks (i.e. language modeling). They use their theoretical results to design the Quad objective and experiment with it on SST and AG news, finding that it performs close but slightly worse than standard cross entropy training of classifiers.

Overall, this work contributes an interesting framework for analysis. However, I have one large conceptual concern about the framework. Central to the proposed explanation is the ability to formulate text classification tasks as next word prediction, possibly with a prompt appended to the input (e.g. for sentiment analysis of movie reviews, “This movie is “). In a trivial sense, this is always possible: We can simply append the task definition to the end of an input as a question (e.g. “Is the sentiment of the review positive?”) and check the probabilities of “yes”/”no”. Then predicting the answer to this prompt is equivalent to performing the task, and a perfect LM is of course able to perform the task perfectly. This formulation makes Sections 3 and 4.1 feel trivially true. Though, to the authors’ credit, they do have to do additional work to extend an LM that is eps-optimal in next word cross entropy (i.e. on average) to optimality on the specific task formulation.

However, the authors don’t mention this trivial reformulation strategy and instead base their argument on the existence of heuristic words (e.g. for sentiment analysis, the probability of “:)” or “:(“ after a review). This strategy introduces the potential for spurious correlations: The heuristics might be strongly correlated with the task in general, but might be off due to other factors like sarcasm. Additionally, relying on these single-word heuristics seems a bit off to me, as many text classification tasks don’t readily admit single words that encapsulate the task definition or label semantics. There’s an argument, then, that the theory described here doesn’t apply to these tasks (i.e. they’re not (t, B)-natural), but what’s frustrating about this argument is that this theory doesn’t provide us a way to distinguish which tasks fall in the category of single-word predictable or how to find such words other than trial and error.

It is very likely I am misunderstanding something about this paper. I am not sure what it means for a task to “lie in the row span of word embeddings”.

---

> ### Author Response · Authors · 2020-11-20
> **Response to Reviewer 4**
>
>
> We thank the reviewer for their comments and explain some potential misunderstandings below. We would like to emphasize is that our main contribution in this work is a precise mathematical formalization of existing intuitions in a way that lets us show meaningful results along with some novel insights.
>
> **[There are alternate sentence completion formulations]**
> The sentence completion formulation mainly provides a starting point from which we develop a substantial amount of theory to formally explain why language modeling is a useful pre-training task for solving downstream tasks. We selected prompts that felt intuitive, but your suggested prompts are also entirely compatible with the theory we develop. The goal of the sentence completion intuition is to establish why $p_{\cdot|s}^*$ can be useful for solving downstream tasks, so it is not as important what specific prompt and heuristic words we use.
>
>
> **[Sections 3 and 4.1 feel trivially true given the sentence completion formulations]**
> While reformulating tasks in similar ways (e.g. with task definition or task specific prompts) has been proposed in recent prior work (cited in Section 1.1), the main goal of Section 3 is to mathematically formalize (Definitions 3.1 and 3.2) this intuition into the assumption that these tasks can be solved as a *linear function* of the condition probability vector $p^*_{\cdot|s}$. This particular formalization lets us show the result in Section 4.1 (Theorem 4.1) with guarantees even for suboptimal language models, as you note. It captures the intuition that stronger language models are better for downstream tasks (through $\epsilon$) and also that it is better for a task to depend on a more frequent set of words (through $B$). While not complicated to prove, the analysis coupled with the aforementioned formalization is the main contribution and we would argue that this is not trivial.
>
> Of equal importance are our contributions in Section 4.2 and beyond, that show a similar but stronger result for $d$-dimensional softmax parametrized language models with the following new observations:
>
> - $\epsilon$-suboptimality w.r.t. the best $d$-dimensional softmax model suffices, rather than comparing to the absolute optimal
>
> - A new mathematical object, conditional mean features, can provably do well on linear classification tasks, as also verified empirically. These features are weakly linear in the original features (see Section 4.3), hence providing some explanation for the success of $f(s)$ on downstream tasks.
>
> **[Theory doesn't provide us a way to distinguish which tasks fall in the category of single-word predictable or how to find such words]**
> Classification tasks that can be intuitively reformulated as sentence completion, possibly by adding a prompt that makes sense, are good candidates to be natural tasks. A more direct way to test is to use the output probabilities from a good language model $\{p_{\cdot|s}\}$ and evaluate the performance of linear classification using $p_{\cdot|s}\in\mathbb{R}^V$ as features. If it does well, then the task is natural almost by definition. For softmax language models, even good performance of the conditional mean features ($\Phi p_{f(s)}\in\mathbb{R}^d$) suffices to know that the task is natural. One way to find words that help solve the task is to look at coordinates of linear classifier learned on $p_{\cdot|s}$ that have high absolute value. But finding these words is not really necessary and the purpose of doing so was mainly to motivate the definition of natural tasks.
>
> **[What does "Lie in the row span of word embeddings" mean]**
> By row span we mean that the classifier $\boldsymbol{v}\in\mathbb{R}^V$ that is used on top of the features $p_{\cdot | s}\in\mathbb{R}^V$ is a linear combination of the $d$ rows of $\Phi$, in other words $\boldsymbol{v} = \Phi^\top \lambda$ for some $\lambda\in\mathbb{R}^d$. We discuss this after Definition 3.2.

---

### Official Review · AnonReviewer5 · 2020-11-09
**An interesting mathematical framework towards understanding language model representations**

**Rating:** 6
**Confidence:** 3

**Review:**

Summary of review:

There have been lots of interests to understand why self-supervised learning approaches such as the next word prediction task learn a useful representation for downstream tasks. This paper provides a mathematical framework to understand this question. One novel finding of this paper is that the distribution of the next word, conditional on the context, can provide a strong discriminative signal for the downstream task. In particular, using a carefully selected subset of "prompt" words, the authors observe that learning a linear predictor over the next word distributions of these words achieves performance close to a pre-trained GPT-2 model.

Setting and Main Result:

This paper focuses on classification tasks, and the bulk of the work goes into how to model the next word distributions as features or representations. For this purpose, the authors introduced the definition of a "natural" task. Informally , a task is defined as natural if, just by using the next word distributions as features, the downstream task can be solved with a small loss.

Result 1: Under the above assumption over the downstream task, this paper provides a bound on the empirical loss of the downstream prediction task. This bound consists of two parts:
- The first part measures how "natural" the task is, that is, how well can the task be solved using the next word distributions as features.
- The second part measures the difference between the "empirical" next word distributions and the "optimal" next word distributions.

Result 2: The authors further extend this result to word embedding features, which are obtained by a weighted average of word embedding vectors based on the next word distributions.

There are several follow-up results built on these two results, such as a new loss objective for predicting the downstream task, but to the best of my understanding, these two results are the main claims of this paper.

A key parameter that occurs in obtaining the above results is a worst-case coefficient that bounds the distributional shift between language model distributions of the training dataset and that of the downstream task. Intuitively, this parameter arises from translating the "natural" task assumption, which only guarantees transfer on average to the downstream task.

Pros:

- A new framework for understanding why learning how to predict the next word helps the downstream task. This paper finds that the next word distributions of a subset of "prompt" words contain discriminative signals and are good features. This seems to be a novel finding and may help inspire future work in this important direction.

Cons:

- The main result (Thm 4.1) applies to next/conditional word distributions that are very close to the optimal distribution. It is unclear to me how the authors are going to justify this "assumption". Should we expect the empirical distribution to converge to the true distribution when there are an infinite amount of samples?

- Secondly, this main result depends on the worst-case coefficient, which is also unclear to me. For the transferability coefficient proposed in Section 5.1, is it possible to measure it in experiments? How large should we expect this coefficient to be?

Writing:

The writing is overall clear and easy to follow, although it took me quite some time to map out the definitions of various notations. Many of the notations look cumbersome and I suspect that there is still room for making the notations more accessible for new readers.


Detailed comments:

- P2, Sec 1.1: "analyze the efficiency language model features" -> analyze the efficiency of language model features

- P2, Sec 2: you started introducing these notations without explaining what they mean. For example, the $p^{\star}$ notation is also defined in Sec 2.1.

- P2, Sec 2: "where $p^{\star}_{\cdot | s}$ is used as a vector on the left and distribution on the right". What does this sentence mean?

- P3, Sec 2.2: "... achieve lower test perplexity than traditional n-gram models" Why is this true? Could you add a reference?

- P5, Sec 4.1: "The result suggests that small test cross-entropy (hence test perplexity)..." Same question as above.

- P6, Sec 4.3: "In fact, $f$ almost always performs better than ..." This part seems intriguing despite the linear relationship shown in figure 1. Could you discuss this more here?

- P8, Table 2: The results from using Quad look worse than the above two. Could you explain the significance of this result again?

- P24, Figure 2: What are the x and y axis, and what does each dot mean in this figure?

---

> ### Author Response · Authors · 2020-11-20
> **Response to Reviewer 5**
>
> We thank the reviewer for their feedback and clarify several points below.
>
>
> **[Should we expect the empirical distribution to converge to the true distribution when there are an infinite amount of samples?]**
> Since the absolute optimal solution is the true distribution, with sufficiently many samples and computational power, an unconstrained LM should be able to learn the true distribution, while an arbitrarily expressive $d$-dimensional softmax LM would learn $p^*_{\cdot|s}$ in a subspace. Hopefully this clarifies your question.
>
> **[Worst-case transferability coefficient is unclear; Can this be measured in experiments]**
> This coefficient captures the shift in distribution of contexts from language modeling ($p_L$) to a downstream task ($p_{\mathcal{T}}$). In fact, this shift is quite intuitive. Suppose $p_L$ is a general purpose corpus, like Wikipedia articles, and the downstream task of interest is to classify sentences by whether they are discussing physics or history.
> If the set of sentences about physics or history forms 10\% of the Wikipedia articles, then $\gamma(p_\mathcal{T}) = 1/10$. On the other hand if $p_L = p_{\mathcal{T}}$, then $\gamma(p_{\mathcal{T}}) = 1$ and we do not suffer blow up. The refined coefficient from Section 5.1 gives a better bound, since it is always upper bounded by $\gamma(p_{\mathcal{T}})$. Since both the coefficients depend on the true distribution $\{p^*_{\cdot|s}\}$, it is not possible to measure these exactly.
>
>
> **[Notation clarification]**
> $p_{\cdot|s}^*$ denotes both a distribution over words and also a vector in $\mathbb{R}^V$. $p_{\cdot|s}^*[w]$ is treating $p_{\cdot|s}^*$ as a vector and accessing the coordinate corresponding to $w$, while $p_{\cdot|s}^*(w)$ is treating it as a distribution and accessing the probability of the word $w$.
>
> **[Citation for softmax language models being better than $n$-gram models]**
> In Table 1 of [1], the authors demonstrate that using softmax parametrized language models can improve perplexity over $n$-gram models by more than 30 points. Softmax language models have since been the parametrization of choice.
>
> **["In fact, $f$ almost always performs better than $g_{f,\Phi}$ (despite a linear relationship)";  Details about Figure 2."]**
> Figures 1 and 2 show the quadratic fit for the log partition function based on Assumption 4.1, where each point in the scatter plot is a sentence/context from the given dataset. The linear relation between $f$ and $g_{f,\Phi}$ is not exact, as discussed right after Corollary 4.1, which is why $f$ could be superior. Lemma 4.3 does not hold exactly because Figure 1 only verifies the quadratic fit for sentence embeddings and not all vectors, as required by Assumption 4.1.
>
> **[Significance of Quad results]**
> The motivation for the Quad objective is to use our insights to design an objective with theoretical guarantees. Quad is more analyzable than the standard KL objective, as we can compute a closed form expression for both $f^*$ and $\Phi^*$. The good performance of this objective is yet another validation for the theoretical analysis, and the performance gap between Quad and KL models suggests that it is worthwhile to study factors beyond what we consider in our analysis (e.g., inductive bias, expressivity of transformer models).
>
> [1] Jozefowicz, R., Vinyals, O., Schuster, M., Shazeer, N. and Wu, Y., 2016. Exploring the limits of language modeling. arXiv preprint arXiv:1602.02410.

---

### Official Review · AnonReviewer2 · 2020-11-10
**A very interesting exploration indeed**

**Rating:** 7
**Confidence:** 3

**Review:**

Summary.

This work tries to understand why features from trained language models can be used to solve classification tasks effectively. A language model (LM) in the analysis is modeled as a feature map $f : S \rightarrow \mathbb{R}^d$, a word embedding $\Phi \in \mathbb{R}^{d \times V}$, and a trained language model is thought of as $\epsilon$-optimal in terms of its cross-entropy (from the true distribution over S). The work shows that for classification tasks approximately solvable by linear functions over the distributions of the next token the best linear classifier based on $(f, \Phi)$ will suffer error of $O(\sqrt{\epsilon})$. The authors also propose an additional assumption where the log partition function is quadratic in $f$ based on which some improvements can be obtained. Being inspired by this assumption, a new objective function Quad where the partition function is directly replaced by a quadratic of $f$ is proposed. Experiments seem to support key assertions in the theoretical analysis.

Strengths.

1. The authors’ approach to a well-posed question seem original to me. In particular, some proposed concepts such as the refined transferability coefficient, conditional mean features, substitutability matrix might be useful for future studies.
2. The article is precise, well-written and cautious in its tone. The accompanying experiments are informative and supportive of the main theoretical claims. I enjoy the overall journey the authors presented and would love to see more well-reasoned articles like this in ICLR.

Weaknesses.

1. The presentation can be improved by allocating more space to ideas in the Extensions section. This part seems more creative (perhaps a little less coherent but expected for “a mathematical exploration”) but is too compressed as it stands. (See Suggestion1)

Minor issues.

1. Consider to replace “partial sentences” with prefixes, which is more technically accurate.
2. The many notations involving $p$ has inconsistent meanings for their subscripts. I would suggest to consolidate them to reduce confusion. For example, consider to use notations of this form $p(w|s; \theta)$. Perhaps boldface for when it is viewed as a vector. Similarly for $\ell$. (See Suggestion2)
3. Below section 2.1, “trained to learn” -> “trained to fit”.

Suggestions.

1. I think a moderate revision reducing some (parallel) elaboration on “unconstrained language model” should provide the space needed for Extensions (and other novel ideas). After all, I found results on unconstrained LMs somewhat trivial and I suspect that you hope to use it only as an instructional tool. Perhaps a serial layout would save space and even improve the perceived emphasis.
2. I strongly suggest a review of the notations and to adopt a more consistent scheme. One trick I found useful is to follow the notational convention of a textbook or a classic paper. The current notation has too much overloading and variability.

Questions.

1. It seems to me that the central question lacks strong practical motivation. The NLP community seems to move to prefer a _natural_ answer (in the form of a generated sentence) instead of a label (from a classifier). As you have argued, many classification tasks can be framed as predicting the next token (perhaps in the presence of a prompt). What is your opinion?
2. How realistic is the $\epsilon$-optimality condition on cross-entropy for LMs? Can you comment on any associated sample complexity bounds?

// Post-rebuttal update:
Thank you for replying to my questions. I am still concerned about the sample complexity associated with $\epsilon$-optimality in cross-entropy (even in the trivial case, and perhaps impossible for some low-dim representations) as LMs are over a countably infinite extended alphabet.

---

> ### Author Response · Authors · 2020-11-20
> **Response to Reviewer 2**
>
> We thank the reviewer for recognizing the novel elements of our work and providing constructive feedback. We address suggestions and questions below.
>
> **[Redistribution of emphasis by allocating more space to the "Extensions" section. Results on unconstrained LMs are used only as an instructional tool]**
> We will try to elaborate more on the extensions section in the revised version using the extra page. The results for unconstrained LMs, although easier to prove, demonstrate how our formalization of the sentence completion view lead to formal statements about downstream task performance. Moreover, the bounds mathematically capture a lot of intuitions without the incurring the additional complications of $d$-dimensional features.
>
> **[Notations]**
> Thank you for the useful suggestions! We will try to incorporate them in the revision. The deviation from standard notations is, in part, to make equations more concise and less cumbersome to read. We will try to find a better trade-off between the two.
>
> **[What about other NLP tasks that want to predict a natural answer rather than label]**
> As correctly pointed out, our current results are only for classification tasks (and potentially other word prediction tasks), which are still relevant and included in benchmarks such as GLUE. We admit that extending the framework and analysis to a more diverse set of NLP tasks will require further thought. We are, however, hopeful that insights from this work can form a strong foundation for non-trivial extensions to those settings.
>
> **[How realistic is the $\epsilon$-optimality condition for LMs?]**
> We note that $\epsilon$-optimality here is measured w.r.t. cross-entropy, which is equivalent to log-perplexity. So if the true perplexity of language is 20, a language model with perplexity 30 will have $\epsilon = \log(30/20) \approx 0.4$.
> It is plausible that current language models that are approaching optimality, but since we do not know the true perplexity of language, we cannot measure $\epsilon$. As we continue to build stronger language models, we can get a lower bound on the $\epsilon$ of the previous ones. Note that for $d$-dimensional softmax models, $\epsilon$ denotes the sub-optimality w.r.t. the best $d$-dimensional model, which can be much less strict than suboptimality w.r.t. true language distribution and thus more likely to be small. The achieved suboptimality $\epsilon$ for any language modeling algorithm will depend on the expressivity of the architecture, optimization algorithm, and amount of data available.

---

### Decision · Program_Chairs · 2021-01-07
**Final Decision**

**Decision:**

Accept (Poster)

**Comment:**

The paper attempts to provide a theoretical explanation for benefit of language model pretraining on downstream classification task. In this regard, the authors provide a mathematical framework which seems to indicate that the distribution of the next word, conditional on the context, can provide a strong discriminative signal for the downstream task. The reviewers found the formulation insightful, interesting, and novel. Also reviewers enjoyed reading the well written paper and appreciated its cautious in its tone. As correctly pointed out by reviewers, the proposed framework might not directly align with techniques used in practice. Applicability of the framework to other pre-training approaches is limited.  Also, there are some unresolved concerns about $O(\sqrt{\epsilon})$ assumption still. Nevertheless, reviewers reached a consensus that the framework would be beneficial for the community and attract follow-up works. Thus, I recommend an acceptance to ICLR. Following reviewer suggestion, it is strongly recommended that extensions section be expanded in the revised version using the extra page.